# The High Line: Exact Risk and Learning Rate Curves of Stochastic Adaptive Learning Rate Algorithms

**Elizabeth Collins-Woodfin**
McGill University
elizabeth.collins-woodfin@mail.mcgill.ca

**Inbar Seroussi**
Tel-Aviv University
inbarser@tauex.tau.ac.il

**Begoña García Malaxechebarría**
University of Washington
begogar9@uw.edu

**Andrew W. Mackenzie**
McGill University
andrew.mackenzie@mail.mcgill.ca

**Elliot Paquette**
McGill University
elliot.paquette@mcgill.ca

**Courtney Paquette**[*]
McGill University & Google DeepMind
courtney.paquette@mcgill.ca

## Abstract

We develop a framework for analyzing the training and learning rate dynamics on a large class of high-dimensional optimization problems, which we call the high line, trained using one-pass stochastic gradient descent (SGD) with adaptive learning rates. We give exact expressions for the risk and learning rate curves in terms of a deterministic solution to a system of ODEs. We then investigate in detail two adaptive learning rates – an idealized exact line search and AdaGrad-Norm – on the least squares problem. When the data covariance matrix has strictly positive eigenvalues, this idealized exact line search strategy can exhibit arbitrarily slower convergence when compared to the optimal fixed learning rate with SGD. Moreover we exactly characterize the limiting learning rate (as time goes to infinity) for line search in the setting where the data covariance has only two distinct eigenvalues. For noiseless targets, we further demonstrate that the AdaGrad-Norm learning rate converges to a deterministic constant inversely proportional to the average eigenvalue of the data covariance matrix, and identify a phase transition when the covariance density of eigenvalues follows a power law distribution. We provide our code for evaluation at https://github.com/amackenzie1/highline2024.

## 1 Introduction

In deterministic optimization, adaptive stepsize strategies, such as line search (see [40], therein), AdaGrad-Norm [59], Polyak stepsize [48], and others were developed to provide stability and improve efficiency and adaptivity to unknown parameters. While the practical benefits for deterministic optimization problems are well-documented, much of our understanding of adaptive learning rate strategies for stochastic algorithms are still in their infancy.

There are many adaptive learning rate strategies used in machine learning with many design goals. Some are known to adapt to stochastic gradient descent (SGD) gradient noise while others are robust to hyper-parameters (e.g., [4, 63]). Theoretical results for adaptive algorithms tend to focus on guaranteeing minimax-optimal rates, but this theory is not engineered to provide realistic performance

---

[*]Corresponding author

38th Conference on Neural Information Processing Systems (NeurIPS 2024).

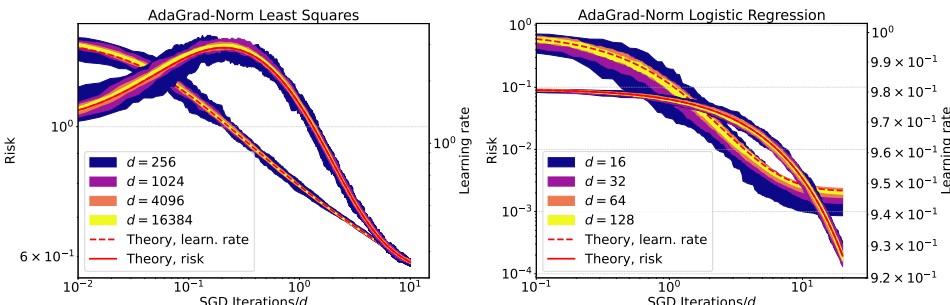

Figure 1: **Concentration of learning rate and risk for AdaGrad-Norm** on least squares with label noise $\omega = 1$ (left) and logistic regression with no noise (right). As dimension increases, both risk and learning rate concentrate around a deterministic limit (red) described by our ODE in Theorem 2.1. The initial risk increase (left) suggests the learning rate started too high, but AdaGrad-Norm adapts. Our ODEs predict this behavior. See Sec. H for simulation details.

comparisons; indeed many adaptive algorithms are minimax-optimal, and so more precise statements are needed to distinguish them. For instance, the exact learning rates (or rate schedules) to which these strategies converge are unknown, nor their dependence on the geometry of the problem. Moreover, we often do not know how these adaptive stepsizes compare with well-tuned constant or decaying fixed learning rate SGD, which can be viewed as a cost associated with selecting the adaptive strategy in comparison to tuning by hand.

In this work, we develop a framework for analyzing the exact dynamics of the risk and adaptive learning rate strategies for a wide class of optimization problems that we call *high-dimensional linear (high line) composite functions*. In this class, the objective function takes the form of an expected risk $\mathcal{R} : \mathbb{R}^d \to \mathbb{R}$ over high-dimensional data $(a, \epsilon) \sim \mathcal{D} \subset \mathbb{R}^d \times \mathbb{R}$ of a function $f : \mathbb{R}^3 \to \mathbb{R}$ composed with the linear functions $\langle X, a \rangle, \langle X^\star, a \rangle$. That is, we seek to solve

$$\min_{X \in \mathbb{R}^d} \left\{ \mathcal{R}(X) \stackrel{\text{def}}{=} \mathbb{E}_{a,\epsilon}[f(\langle a, X \rangle, \langle a, X^\star \rangle, \epsilon)] \quad \text{for} \quad (a, \epsilon) \sim \mathcal{D}, X^\star \in \mathbb{R}^d \right\}. \tag{1}$$

We suppose $a \sim \mathcal{N}(0, K)$ where $K \in \mathbb{R}^{d \times d}$ is the covariance matrix. We train (1) using (one-pass) stochastic gradient descent with adaptive learning rates, $\mathfrak{g}_k$ (SGD+AL). Our main goal is to give a framework for better[2] performance analysis of these adaptive methods. We then illustrate this framework by considering two adaptive learning rate algorithms on the least squares problem[3], the results of which appear in Table 1: exact line-search (idealistic) (Sec. 3) and AdaGrad-Norm (Sec. 4). We expect other losses and adaptive learning rates can be studied using this approach.

**Main contributions.**   *Performance analysis framework.* We provide an equivalence of $\mathcal{R}(X_k)$ and learning rate $\mathfrak{g}_k$ under SGD+AL to deterministic functions $\mathscr{R}(t)$ and $\gamma_t$ via solving a *deterministic* system of ODEs (see Section 2), which we then analyze to show how the covariance spectrum influences the optimization. See Figure 1. As the dimension $d$ of the problem grows, the learning curves of $\mathcal{R}(X_k)$ become closer to $\mathscr{R}(t)$ and the curves concentrate around $\mathscr{R}(t)$ with probability better than any inverse power of $d$ (See Theorem 2.1).

*Greed can be arbitrarily bad in the presence of strong anisotropy (that is, $Tr(K)/d \ll Tr(K^2)/d$).* Our analysis reveals that exact line search, which is to say optimally decreasing the risk at each step, can run arbitrarily slower than the best fixed learning rate for SGD on a least squares problem when $\lambda_{\min} \stackrel{\text{def}}{=} \lambda_{\min}(K) > C > 0$. The best fixed stepsize (least squares problem) is $(Tr(K)/d)^{-1}$ or the inverse of the average eigenvalue, see Polyak stepsize [48]. Line search, on the other hand, converges to a fixed stepsize of order $\lambda_{\min}/(Tr(K^2)/d)$. It can be that $\lambda_{\min}/(Tr(K^2)/d) \ll (Tr(K)/d)^{-1}$ making exact line search substantially underperform Polyak stepsize. We further explore this and, in the case where $d$-eigenvalues of $K$ take only two values $\lambda_1 > \lambda_2 > 0$, we give an exact expression as a function of $\lambda_1$ and $\lambda_2$ for the limiting behavior of $\gamma_t$ as $t \to \infty$ (See Fig. 5).

---

[2]More realistic, in that it deals with high-dimensional anisotropic loss geometries and more precise, in that it can distinguish minimax optimal algorithms as more-or-less performant.

[3]We extend some results to the general strongly convex setting.

Table 1: **Summary of adaptive learning rates results on the least squares problem.** We summarize our results for line search and AdaGrad-Norm under various assumptions on the covariance matrix $K$. We denote $\lambda_{\min}$ the smallest non-zero eigenvalue of $K$ and $\frac{\text{Tr}(K)}{d}$ the average eigenvalue. Power law$(\delta, \beta)$ assumes the eigenvalues of $K$, $\{\lambda_i\}_{i=1}^d$, follow a power law distribution, that is, for $0 < \beta < 1$, $\lambda_i \sim (1-\beta)\lambda^{-\beta}\mathbf{1}_{(0,1)}$ for all $1 \le i \le d$ and $\langle X_0 - X^\star, \omega_i \rangle^2 \sim \lambda_i^{-\delta}$ where $\{\omega_i\}_{i=1}^d$ are eigenvectors of $K$ (see Prop 4.4). For $^\star$ (see Prop. 4.2), requires a good initialization on $b, \eta$.

| Learning rate | K assumption | Limiting $\gamma_\infty$ | Convergence rate |
|---|---|---|---|
| AdaGrad-Norm$(b, \eta)$ (see Sec. 4) | $\lambda_{\min} > C$ | $\gamma_t \asymp \frac{\eta^2}{\frac{b}{\eta} + \frac{1}{4d}\text{Tr}(K)\|X_0 - X^\star\|^2}$ | $\log(\mathscr{R})^\star \asymp -\lambda_{\min}\gamma_\infty t$ |
| AdaGrad-Norm$(b, \eta)$ Power law (see Sec. 4) | $\beta + \delta < 1$ | $\gamma_t \asymp_{\delta,\beta} 1$ | $\mathscr{R}(t) \asymp_{\delta,\beta} t^{\beta+\delta-2}$ |
| | $\beta + \delta = 1$ | $\gamma_t \asymp_{\delta,\beta} \frac{1}{\log(t+1)}$ | $\mathscr{R}(t) \asymp_{\delta,\beta} \left(\frac{t}{\log(t+1)}\right)^{-1}$ |
| | $1 < \beta + \delta < 2$ | $\gamma_t \asymp_{\delta,\beta} t^{-1+\frac{1}{\beta+\delta}}$ | $\mathscr{R}(t) \asymp_{\delta,\beta} t^{-\frac{2}{\beta+\delta}+1}$ |
| Exact line search, idealized (see Sec. 3) | $\lambda_{\min} > C$ | $\gamma_t \asymp \frac{\lambda_{\min}}{\text{Tr}(K^2)/d}$ | $\log(\mathscr{R}) \asymp -\lambda_{\min}\gamma_\infty t$ |
| Polyak stepsize (see Sec. 3) | $\lambda_{\min} > C$ | $\gamma_t = \frac{1}{\text{Tr}(K)/d}$ | $\log(\mathscr{R}) \asymp -\lambda_{\min}\gamma_\infty t$ |

*AdaGrad-Norm selects the optimal step-size, provided it has a warm start.* In the absence of label noise and when the smallest eigenvalue of $K$ satisfies $\lambda_{\min} > C > 0$, the learning rate converges to a deterministic constant that depends on the average condition number (like in Polyak) and scales inversely with $\frac{\text{Tr}(K)}{d}\|X_0 - X^\star\|^2$. Therefore it attains automatically the optimal fixed stepsize in terms of the covariance *without* knowledge of $\text{Tr}(K)$, but pays a penalty in the constant, namely $\|X_0 - X^\star\|^2$. If one knew $\|X_0 - X^\star\|^2$ then by tuning the parameters of AdaGrad-Norm one might achieve performance consistent with Polyak; this also motivates more sophisticated adaptive algorithms such as DoG [29] and D-Adaptation [18], which adaptively compensate and/or estimate $\|X_0 - X^\star\|^2$.

*AdaGrad-Norm can use overly pessimistic decaying schedules on hard problems.* Consider power law behavior for the spectrum of $K$ and the signal $X^\star$. This is a natural setting as power law distributions have been observed in many datasets [60]. Here the learning rate and asymptotic convergence of $K$ undergo a *phase transition*. For power laws corresponding to easier optimization problems, the learning rate goes to a constant and the risk decays at $t^{-\alpha_1}$. For harder problems, the learning rate decays like $t^{-\eta_1}$ and the risk decays at a different sublinear rate $t^{-\alpha_2}$. See Table 1 and Sec. 4 for details.

**Notation.** Define $\mathbb{R}_+ = [0, \infty)$. We say an event holds *with overwhelming probability, w.o.p.,* if there is a function $\omega : \mathbb{N} \to \mathbb{R}$ with $\omega(d)/\log d \to \infty$ so that the event holds with probability at least $1 - e^{-\omega(d)}$. We let $\mathbf{1}_A(x)$ be the indicator function of the set $A$ where it is 1 if $x \in A$ and 0 otherwise. For a matrix $A \in \mathbb{R}^{m \times d}$, we use $\|A\|$ to denote the Frobenius norm and $\|A\|_{\text{op}}$ to denote the operator-2 norm. If unspecified, we assume that the norm is the Frobenius norm. For normed vector spaces $\mathcal{A}, \mathcal{B}$ with norms $\|\cdot\|_{\mathcal{A}}$ and $\|\cdot\|_{\mathcal{B}}$, respectively, and for $\alpha \ge 0$, we say a function $F : \mathcal{A} \to \mathcal{B}$ is *$\alpha$-pseudo-Lipschitz* with constant $L$ if for any $A, \hat{A} \in \mathcal{A}$, we have

$$\|F(A) - F(\hat{A})\|_{\mathcal{B}} \le L\|A - \hat{A}\|_{\mathcal{A}}(1 + \|A\|_{\mathcal{A}}^\alpha + \|\hat{A}\|_{\mathcal{A}}^\alpha).$$

We write $f(t) \asymp g(t)$ if there exist *absolute* constants $C, c > 0$ such that $c \cdot g(t) \le f(t) \le C \cdot g(t)$ for all $t$. If the constants depend on parameters, e.g., $\alpha$, then we write $\asymp_\alpha$.

**Related work.** Some notable adaptive learning rates in the literature are AdaGrad-Norm [32, 59, 61], RMSprop [28], stochastic line search, stochastic Polyak stepsize [35], and more recently DoG [29] and D-Adaptation [18]. In this work, we introduce a framework for analyzing these algorithms, and we strongly believe it can be used to analyze many more of these adaptive algorithms. We highlight below a nonexhaustive list of related work.

**AdaGrad-Norm.** AdaGrad, introduced by [19, 36], updates the learning rate at each iteration using the stochastic gradient information. The single stepsize version [32, 59, 61], that depends on the norm of the gradient, (see Table 2 for the updates), has been shown to be robust to input parameters [34]. Several works have shown worst-case convergence guarantees [21, 33, 57, 59]. A linear rate of $O(\exp(-\kappa T))$ is possible for $\mu$-strongly convex, $L$-smooth functions ($\kappa$ is the condition number $\mu/L$). In [62] (similar idea in [61]), the authors show for strongly convex, smooth stochastic objectives (with additional assumptions) that the AdaGrad-Norm learning rate exhibits a two stage behavior – a burn in phase and then when it reaches the smoothness constant it self-stablizes.

**Stochastic line search and Polyak stepsizes.** Recently there has been renewed interest in studying stochastic line search [20, 42, 54] and stochastic Polyak stepsize (and their variants) [7, 26, 27, 30, 35, 39, 41, 49]. Much of this research focuses on worst-case convergence guarantees for strongly convex and smooth functions (see e.g., [35]) and designing practical algorithms. In [53], the authors provide a bound on the learning rate for Armijo line search in the finite sum setting with a rate of $L_{\max}/$avg. $\mu$ where avg. $\mu$ is the avg. strong convexity and $L_{\max}$ is the max. Lipschitz constant of the individual functions. In this work, we consider a slightly different problem. We work with the population loss and we note that the analogue to $L_{\max}$ for us would require that the samples $a$ satisfy $\|aa^T\|_{\mathrm{op}} \leq L_{\max}$ for all $a$; this fails to hold for $a \sim \mathcal{N}(0, K)$. Moreover, $L_{\max}$ could be much worse than $\mathbb{E}[\|aa^T\|_{\mathrm{op}}]$.

**Deterministic dynamics of stochastic algorithms in high-dimensions.** The literature on deterministic dynamics for isotropic Gaussian data has a long history [9, 10, 50, 51]. These results have been rigorously proven and extended to other models under the isotropic Gaussian assumption [1, 2, 6, 16, 17, 23, 58]. Extensions to multi-pass SGD with small mini-batches [46] as well as momentum [31] have also been studied. Other high-dimensional limits leading to a different class of dynamics also exist [11–13, 22, 37]. Recently, significant contributions have been made in understanding the effects of a non-identity data covariance matrix on the training dynamics [5, 14, 15, 24, 25, 64]. The non-identity covariance modifies the optimization landscape and affects convergence properties, as discussed in [15]. This work extends the findings of [15] to stochastic adaptive algorithms, exploring the effect of non-identity covariance within these algorithms. Notably, Theorem 1.1 from [15] is restricted to deterministic learning rate schedules, limiting its applicability in many practical scenarios. In contrast, our Theorem 2.1 accommodates stochastic adaptive learning rates, aligning with widely used algorithms in practice.

## 1.1 Model Set-up

We suppose that a sequence of independent samples $\{(a_k, y_k)\}$ drawn from a distribution $\mathcal{D} \subset \mathbb{R}^d \times \mathbb{R}$ is provided where $y_k$ is the target. The target $y_k$ is a function of some random label noise $\epsilon_k \in \mathbb{R}$ and the input feature $a_k$ dotted with a ground truth signal $X^\star \in \mathbb{R}^d$, $\langle a_k, X^\star \rangle$. Therefore, the distribution of the data is only determined by the input feature and the noise, i.e., the pair $(a, \epsilon)$. In particular, we assume $(a, \epsilon)$ follows a distributional assumption.

**Assumption 1** (Data and label noise). *The samples $(a, \epsilon) \sim \mathcal{D}$ are normally distributed: $\epsilon \sim \mathcal{N}(0, \omega^2)$ where $\omega \in \mathbb{R}$, and $a \sim \mathcal{N}(0, K)$, with a covariance matrix $K \in \mathbb{R}^{d \times d}$ that is bounded in operator norm independent of $d$; i.e., $\|K\|_{op} \leq C$. Furthermore, $a$ and $\epsilon$ are independent.*

For $a, X, X^\star \in \mathbb{R}^d$, $\epsilon \in \mathbb{R}$, and a function $f : \mathbb{R}^3 \to \mathbb{R}$, we seek to minimize an expected risk function $\mathcal{R} : \mathbb{R}^d \to \mathbb{R}$, which we refer to as the *high-dimensional linear composite*[4], of the form

$$\mathcal{R}(X) \stackrel{\text{def}}{=} \mathbb{E}_{a,\epsilon}[\Psi(X; a, \epsilon)] \quad \text{for} \quad (a, \epsilon) \sim \mathcal{D}, \quad \text{and} \quad \Psi(X; a, \epsilon) = f(\langle a, X \rangle, \langle a, X^\star \rangle, \epsilon). \quad (2)$$

In what follows, we use the matrix $W = [X|X^\star] \in \mathbb{R}^{d \times 2}$ that concatenates $X$ and $X^\star$, and we shall let $B = B(W) = W^T K W$ be the covariance matrix of the Gaussian vector $(\langle a, X \rangle, \langle a, X^\star \rangle)$.

**Assumption 2** (Pseudo-lipschitz $f$). *The function $f : \mathbb{R}^3 \to \mathbb{R}$ is $\alpha$-pseudo-Lipschitz with $\alpha \leq 1$.*

By assumption, $\mathcal{R}(X)$ involves an expectation over the correlated Gaussians $\langle a, X \rangle$ and $\langle a, X^\star \rangle$. We can express this as $\mathcal{R}(X) \stackrel{\text{def}}{=} h(B)$ for some well-behaved function $h : \mathbb{R}^{2 \times 2} \to \mathbb{R}$.

---

[4]Note that $d$ need not be large to define this, but the structure allows us to consider $d$ as a tunable parameter. Moreover, as we increase $d$, the analysis we do will be more meaningful.

Table 2: **Two adaptive learning rates considered in detail.** The stochastic adaptive learning rate, $\mathfrak{g}_k$, is the learning rate directly used in the update for SGD whereas the deterministic, $\gamma_t$, is the deterministic equivalent of $\mathfrak{g}_k$ after scaling.

| Algorithm | | General update | Least squares |
|---|---|---|---|
| AdaGrad-Norm$(b, \eta)$ $b_0 = b \times d$ | $\mathfrak{g}_k$ | $b_k^2 = b_{k-1}^2 + \|\nabla\Psi(X_{k-1})\|^2;$ $\mathfrak{g}_{k-1} = d \times \frac{\eta}{|b_k|}$ | same |
| | $\gamma_t$ | $\dfrac{\eta}{\sqrt{b^2 + \frac{\mathrm{Tr}(K)}{d}\int_0^t I(\mathscr{B}(s))\,\mathrm{d}s}}$ | $\dfrac{\eta}{\sqrt{b^2 + \frac{2\mathrm{Tr}(K)}{d}\int_0^t \mathscr{R}(s)\,\mathrm{d}s}}$ |
| Exact line search (idealized) | $\mathfrak{g}_k$ | $\dfrac{\|\nabla\mathscr{R}(X_k)\|^2}{\frac{\mathrm{Tr}(\nabla^2\mathscr{R}(X_k)K)}{d}\mathbb{E}_{a,\epsilon}[(f'(\langle a,X\rangle;\langle a,X^\star\rangle,\epsilon))^2]}$ | $\dfrac{\|\nabla\mathscr{R}(X_k)\|^2}{\frac{2\mathrm{Tr}(K^2)}{d}\mathscr{R}(X_k)}$ |
| | $\gamma_t$ | $\underset{\gamma}{\arg\min}\,\mathrm{d}\mathscr{R}(t)$ | $\dfrac{\sum_{i=1}^d \lambda_i^2 \mathscr{D}_i^2(t)}{2\mathrm{Tr}(K^2)\mathscr{R}(t)}$ |

**Assumption 3** (Risk representation). *There exists a function $h : \mathbb{R}^{2\times 2} \to \mathbb{R}$ such that $h(B) = \mathcal{R}(X)$ is differentiable and satisfies*

$$\nabla_X \mathcal{R}(X) = \mathbb{E}_{a,\epsilon}\nabla_X\Psi(X; a, \epsilon).$$

*Furthermore, $h$ is continuously differentiable and its derivative $\nabla h$ is $\alpha$-pseudo-Lipschitz for some $0 \le \alpha \le 1$, with constant $L(\nabla h)$.*

The final assumption is the well-behavior of the Fisher information matrix of the gradients. The first coordinate of $f$ is special, as the optimizer must be able to differentiate it. Thus, we treat $f(x, x^\star, \epsilon)$ as a function of a single variable with two parameters: $f(x, x^\star, \epsilon) = f(x; x^\star, \epsilon)$ and denote the (almost everywhere) derivative with respect to the first variable as $f'$.

**Assumption 4** (Fisher matrix). *Define $I(B) \stackrel{\text{def}}{=} \mathbb{E}_{a,\epsilon}[(f'(\langle a, X\rangle; \langle a, X^\star\rangle, \epsilon))^2]$ where the function $I : \mathbb{R}^{2\times 2} \to \mathbb{R}$. Furthermore, $I$ is $\alpha$-pseudo-Lipschitz with constant $L(I)$ for some $\alpha \le 1$.*

A large class of natural regression problems fit within this framework, such as logistic regression and least squares (see [15, Appendix B]). We also note that Assumptions 3 and 4 are nearly satisfied for $L$-smooth objectives $f$ (see Lemma B.1), and a version of the main theorem holds under just this assumption (albeit with a weaker conclusion).

## 1.2 Algorithmic set-up

We apply *one-pass* or *streaming* SGD with an adaptive learning rate $\mathfrak{g}_k$ (SGD+AL) to solve $\min_{X\in\mathbb{R}^d} \mathcal{R}(X)$, (2). Let $X_0 \in \mathbb{R}^d$ be an initial vector (random or non-random). Then SGD+AL iterates by selecting a *new* data point $(a_{k+1}, \epsilon_{k+1})$ such that $a_{k+1} \sim \mathcal{N}(0, K)$ and $\epsilon_{k+1} \sim \mathcal{N}(0, \omega^2)$ and makes the update

$$X_{k+1} = X_k - \frac{\mathfrak{g}_k}{d}\cdot\nabla_X\Psi(X_k; a_{k+1}, \epsilon_{k+1}) = X_k - \frac{\mathfrak{g}_k}{d}f'(\langle a_{k+1}, X_k\rangle; \langle a_{k+1}, X^\star\rangle, \epsilon_{k+1})a_{k+1}, \quad (3)$$

where $\mathfrak{g}_k > 0$ is a learning rate (see assumptions below)[5]. To perform our analysis, we place the following assumption on the initialization $X_0$ and signal $X^\star$.

**Assumption 5** (Initialization and signal). *The initialization point $X_0$ and the signal $X^\star$ are bounded independent of $d$, that is, $\max\{\|X_0\|, \|X^\star\|\} \le C$ for some $C$ independent of $d$.*

**Adaptive learning rate.** Our analysis requires some mild assumptions on the learning rate. To this end, we define a learning rate function $\gamma : \mathbb{R}_+ \times D([0,\infty)) \times D([0,\infty)) \times D([0,\infty)) \to \mathbb{R}_+$ by[6]

$$\mathfrak{g}_k \stackrel{\text{def}}{=} \gamma(k, N_k(d\times\cdot), G_k(d\times\cdot), Q_k(d\times\cdot)), \text{ for } k\in\mathbb{N}, \text{ where for any } t\ge 0,$$

$$(N_k(t), G_k(t), Q_k(t)) \stackrel{\text{def}}{=} \mathbf{1}_{\{t<k\}}\left((W_t)^T W_t, \tfrac{1}{d}\|\nabla_X\Psi(X_t; a_{t+1}, \epsilon_{t+1})\|^2, \mathcal{R}(X_t)\right). \quad (4)$$

---

[5]Note that cases where $\frac{\mathrm{Tr}(K^2)}{d} = o(d)$ can lead to dynamics that converge to full-batch gradient flow. While our theorem specifically addresses the scenario where the intrinsic dimension, $\mathrm{Dim}(K) \stackrel{\text{def}}{=} \mathrm{Tr}(K)/\|K\|_{\mathrm{op}}$, satisfies $\mathrm{Dim}(K) = \Theta(d)$, other cases, such as $\mathrm{Dim}(K) = o(d)$, may require different learning rate scalings.

[6]$D([0,\infty))$ is the càdlàg function class on $[0,\infty)$.

In this definition, for functions taking integer arguments, we extend them to real-valued inputs by first taking the floor function of its argument. Note that the adaptive learning rates can depend on the whole history of stochastic iterates $(N_k)$, gradients $(G_k)$, and risk $(Q_k)$ via this definition.

We also define a conditional expectation version of $G_k$ where the filtration $\mathcal{F}_k = \sigma(X^\star, X_0, \ldots, X_k)$:

$$\mathcal{G}_k(t) \overset{\text{def}}{=} \mathbf{1}_{\{t<k\}}(\cdot)\frac{1}{d}\mathbb{E}[\|\nabla_X \Psi(X_t; a_{t+1}, \epsilon_{t+1})\|^2|\mathcal{F}_t] \quad \text{for } t \geq 0.$$

With this, we impose the following learning rate condition.

**Assumption 6** (Learning rate). *The learning rate function $\gamma : \mathbb{R}_+ \times D([0,\infty)) \times D([0,\infty)) \times D([0,\infty)) \to \mathbb{R}$ is $\alpha$-pseudo-Lipschitz with constant $L(\gamma)$ (independent of d) in $D([0,\infty)) \times D([0,\infty)) \times D([0,\infty))$. Moreover, for some constant $C = C(\gamma) > 0$ independent of d and $\delta > 0$,*

$$\mathbb{E}\left[|\gamma(k, f, G_k(d \times \cdot), q) - \gamma(k, f, \mathcal{G}_k(d \times \cdot), q)| \,|\, \mathcal{F}_k\right] \leq Cd^{-\delta}(1 + \|f\|_\infty^\alpha + \|q\|_\infty^\alpha) \quad \text{w.o.p.} \quad (5)$$

*Finally, $\gamma$ is bounded, i.e., there exists a constant $\hat{C} = \hat{C}(\gamma) > 0$ independent of d so that*

$$\gamma(k, f, g, q) \leq \hat{C}(1 + \|f\|_\infty^\alpha + \|q\|_\infty^\alpha + \|g\|_\infty^\alpha). \quad (6)$$

The inequality (5) ensures that the learning rate concentrates around the mean behavior of the stochastic gradients. Many well-known adaptive stepsizes satisfy (4) and Assumption 6 including AdaGrad-Norm, DoG, D-Adaptation, and RMSProp (see Table 2, Sec. A, and Sec. C.3).

## 2 Deterministic dynamics for SGD with adaptive learning rates

**Intuition for deriving dynamics:** The risk $\mathcal{R}(X)$ and Fisher matrix can be evaluated solely in terms of the covariance matrix $B$. Thus, to know the evolution of the risk over time, it would suffice to know the evolution of $B$. Alas, except in the isotropic case where $K$ is a multiple of the identity, the evolution of $B$ is not autonomous (i.e., its time evolution depends on other unknown variables). However, if we let $(\lambda_i, \omega_i)$ be the eigenvalues and corresponding orthonormal eigenvectors of $K$, we can consider projections $V_i(X_k) = d \cdot W_k^T \omega_i \omega_i^T W_k$, and it turns out that these behave autonomously.

**Example: Least Squares.** One canonical example of (2) is least squares, where we aim to recover the target $X^\star$ given noisy observations $\langle a, X^\star \rangle + \epsilon$. In this case, the *least squares problem* is

$$\min_{X \in \mathbb{R}^d} \left\{ \mathcal{R}(X) = \tfrac{1}{2}\mathbb{E}_{a,\epsilon}[(\langle a, X - X^\star \rangle - \epsilon)^2] = \tfrac{1}{2}\omega^2 + \tfrac{1}{2}(X - X^\star)^T K(X - X^\star) \right\}. \quad (7)$$

The pair of functions $h$ (Assumption 3) and $I$ (Assumption 4) can be evaluated simply:

$$h(B(W)) = \tfrac{1}{2}I(B(W)) = \tfrac{1}{2}(X - X^\star)^T K(X - X^\star) + \tfrac{1}{2}\omega^2.$$

The deterministic dynamics for the risk $\mathscr{R}(t)$ in this case can be simplified to:

$$\mathscr{R}(t) = \tfrac{1}{2}(X_0 - X^\star)^T Ke^{-2K\int_0^t \gamma_s \, ds}(X_0 - X^\star) + \tfrac{1}{2}\omega^2 + \tfrac{1}{d}\int_0^t \gamma_s^2\mathrm{Tr}(K^2e^{-2K\int_s^t \gamma_\tau \, d\tau})\mathscr{R}(s) \, ds.$$

This is a convolution Volterra equation with a convergence threshold of $\gamma_t < \frac{2d}{\mathrm{Tr}K}$ [14, 44, 46, 47].

In the noiseless label case (i.e., $\epsilon = 0$), the risk is given by $\mathscr{R}(t) = \frac{1}{2d}\sum_{i=1}^d \lambda_i \mathscr{D}_i^2(t)$. Using the ODEs in (9), we get the following deterministic equivalent ODE for the $\mathscr{D}_i^2$'s:

$$\frac{d}{dt}\mathscr{D}_i^2(t) = -2\gamma_t\lambda_i\mathscr{D}_i^2(t) + 2\gamma_t^2\lambda_i\mathscr{R}(t). \quad (8)$$

We will perform a deep analysis of the dynamics of the learning rate on least squares (7), which will generalize to settings where the outer function $f$ is strongly convex (see D.1).

**Deterministic dynamics.** To derive deterministic dynamics, we make the following change to continuous time by setting

$$k \text{ iterations of SGD} = \lfloor td \rfloor, \quad \text{where } t \in \mathbb{R} \text{ is the continuous time parameter.}$$

This time change is necessary, as when we scale the size of the problem, more time is needed to solve the underlying problem. This scaling law scales SGD so all training dynamics live on the same

space. One can solve a smaller $d$ problem and scale it to recover the training dynamics of the larger problem.[7]

We now introduce a coupled system of differential equations, which will allow us to model the behaviour of our learning algorithms. For the $i$th $(\lambda_i, \omega_i)$-eigenvalue/eigenvector of $K$, set

$$\mathscr{V}_i(t) \stackrel{\text{def}}{=} \begin{bmatrix} \mathscr{V}_{11,i}(t) & \mathscr{V}_{12,i}(t) \\ \mathscr{V}_{12,i}(t) & \mathscr{V}_{22,i}(t) \end{bmatrix} \text{ and averaging over } i, \ \mathscr{B}(t) \stackrel{\text{def}}{=} \frac{1}{d} \sum_{i=1}^{d} \lambda_i \mathscr{V}_i(t).$$

The $\mathscr{V}_i(t)$ and $\mathscr{B}(t)$ are deterministic continuous analogues of $V_i(X_{td})$ and $B(X_{td})$ respectively. Define the following continuous analogues

$$\nabla h(\mathscr{B}(t)) \stackrel{\text{def}}{=} \begin{bmatrix} H_{1,t} & H_{2,t} \\ H_{2,t} & H_{3,t} \end{bmatrix}, \ \mathscr{N}(t) \stackrel{\text{def}}{=} \frac{1}{d} \sum_{i=1}^{d} \mathscr{V}_i(t), \ \mathscr{R}(t) \stackrel{\text{def}}{=} h(\mathscr{B}(t)), \ \mathscr{I}(t) \stackrel{\text{def}}{=} I(\mathscr{B}(t)),$$

$$\text{and finally } \gamma_t \stackrel{\text{def}}{=} \gamma(t, 1_{\{\cdot \leq t\}} \mathscr{N}(\cdot), \tfrac{\text{Tr}(K)}{d} 1_{\{\cdot \leq t\}} \mathscr{I}(\cdot), 1_{\{\cdot \leq t\}} \mathscr{R}(\cdot)).$$

We now introduce a system of coupled ODEs for each $(\lambda_i, \omega_i)$-eigenvalue/eigenvector pair of $K$

$$\begin{aligned} \mathrm{d}\mathscr{V}_{11,i}(t) &= -2\lambda_i \gamma_t \left( \mathscr{V}_{11,i}(t) H_{1,t} + H_{1,t} \mathscr{V}_{11,i}(t) + \mathscr{V}_{12,i}(t) H_{2,t} + H_{2,t} \mathscr{V}_{12,i}(t) \right) + \lambda_i \gamma_t^2 \mathscr{I}(t), \\ \mathrm{d}\mathscr{V}_{12,i}(t) &= -2\lambda_i \gamma_t \left( H_{1,t} \mathscr{V}_{12,i}(t) + H_{2,t} \mathscr{V}_{22,i}(t) \right) \end{aligned}$$

(9)

with the initialization of $\mathscr{V}_i(0)$ given by $V_i(X_0)$. We finally state the deterministic dynamics for the risk and learning rate.

**Theorem 2.1.** *Under Assumptions 1, 2, 3, 4, 5, 6, then for any $\varepsilon \in (0, \frac{1}{2})$ and any $T > 0$*

$$\sup_{0 \leq t \leq T} \left\| \begin{pmatrix} \mathcal{R}(X_{\lfloor td \rfloor}) \\ \mathfrak{g}_{\lfloor td \rfloor} \end{pmatrix} - \begin{pmatrix} \mathscr{R}(t) \\ \gamma_t \end{pmatrix} \right\| < d^{-\varepsilon}, \quad \textit{w.o.p.} \tag{10}$$

*The same statements hold comparing $W_{td}^T W_{td}$ to $\mathscr{N}(t)$ and $W_{td}^T K W_{td}$ to $\mathscr{B}(t)$.*

In fact, we can derive deterministic dynamics for a large class of statistics which are linear combinations of $\mathscr{V}(t)$ and functions thereof (See Theorem B.1, and Corollary B.1).

One important corollary is a deterministic limit for the distance to optimality, $D^2(X_k) = \|X_k - X^\star\|^2$, which is a quadratic form of $W_k^T W_k$ and hence covered by Thm. 2.1. The equivalent deterministic dynamics are

$$\mathscr{D}^2(t) = \frac{1}{d} \sum_{i=1}^{d} \mathscr{D}_i^2(t) = \frac{1}{d} \sum_{i=1}^{d} (\mathscr{V}_{11,i}(t) - 2\mathscr{V}_{12,i}(t) + \mathscr{V}_{22,i}(t)), \tag{11}$$

where $\mathscr{D}_i^2(t)$ corresponds $D_i^2(X_k) \stackrel{\text{def}}{=} d \times (\langle X_k - X^\star, \omega_i \rangle)^2$.

## 3 Idealized Exact Line Search and Polyak Stepsize

In this section, we consider two classical idealized algorithms – *exact line search* and *Polyak stepsize*. In deterministic optimization, these learning rate strategies are chosen so that the function value (exact line search) or distance to optimality (Polyak) produces the largest decrease in function value (resp. distance to optimality) at the next iteration. For stochastic algorithms, we can ask this to hold for the deterministic equivalent to the risk $\mathscr{R}(t)$ (resp. distance to optimality, $\mathscr{D}(t)$) since we know that SGD is close to these deterministic equivalents. Thus, the question is: what choice of learning rate decreases the $\mathscr{R}(t)$ (*exact line search*) and/or $\mathscr{D}(t)$ (*Polyak stepsize*)? We will restrict to least squares in this section – see Appendix F.1 and F.2 for general functions as well as proofs for least squares. These are idealized algorithms because we can not implement them as they require distributional knowledge of $a$ or $X^\star$. Despite this, they provide a basis for more practical algorithms.

---

[7]Note that, holding time fixed, we perform $O(d)$ gradient updates for a problem of dimension $d$. For the problems considered here, this scaling leads to consistent dynamics, but there do exist related problems where a different scaling is more appropriate. For example, under random initialization, to capture the escape of phase retrieval from the high-dimensional saddle, $O(d \log d)$ iterations are needed; see for example [56].

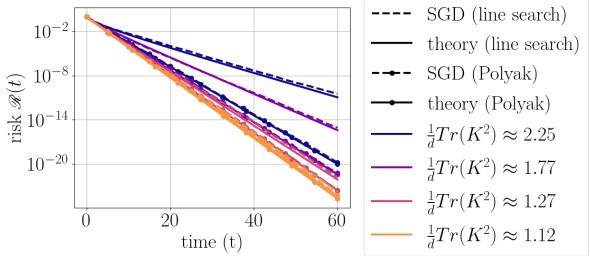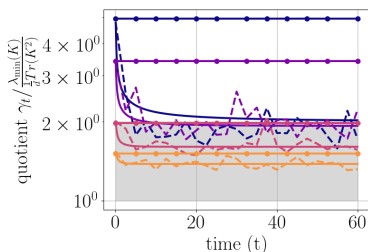

Figure 2: **Comparison for Exact Line Search and Polyak Stepsize** on a noiseless least squares problem. The left plot illustrates the convergence of the risk function, while the right plot depicts the convergence of the quotient $\gamma_t / \frac{\lambda_{\min}(K)}{\frac{1}{d}\operatorname{Tr}(K^2)}$ for Polyak stepsize and exact line search. Both plots highlight the implication of equation (13) in high-dimensional settings, where a broader spectrum of $K$ results in $\frac{\lambda_{\min}(K)}{\frac{1}{d}\operatorname{Tr}(K^2)} \ll \frac{1}{\frac{1}{d}\operatorname{Tr}(K)}$, indicating slower risk convergence and poorer performance of exact line search (unmarked) as it deviates from the Polyak stepsize (circle markers) . The gray shaded region demonstrates that equation (13) is satisfied. See Appendix H for simulation details.

**Polyak Stepsize.**    A natural threshold to consider is the largest learning rate such that $\mathrm{d}\mathscr{D}(t) < 0$, which we denote by $\bar{\gamma}_t^{\mathscr{D}}$. Using the least squares ODE (8), this is precisely

$$\bar{\gamma}_t^{\mathcal{D}} = \frac{(2\mathscr{R}(t)-\omega^2)}{\frac{\operatorname{Tr}(K)}{d}\mathscr{R}(t)} \quad \text{and} \quad \bar{\mathfrak{g}}_k^{\mathscr{D}} = \frac{(2\mathcal{R}(X_k)-\omega^2)}{\frac{\operatorname{Tr}(K)}{d}\mathcal{R}(X_k)}. \tag{12}$$

Without label noise, (12) simplifies to $\bar{\gamma}_t^{\mathcal{D}} = \bar{\mathfrak{g}}_k^{\mathscr{D}} = \frac{2}{\operatorname{Tr}(K)/d}$, the exact threshold for convergence of least squares.

A greedy stepsize strategy would maximize the decrease in the distance to optimality at each iteration, denoted by us as *Polyak stepsize*, $\gamma_t^{\text{Polyak}} \in \arg\min_\gamma \mathrm{d}\mathscr{D}(t)$. In the case of least squares, this is

$$\gamma_t^{\text{Polyak}} = \tfrac{1}{2}\bar{\gamma}_t^{\mathcal{D}} \quad \text{and} \quad \mathfrak{g}_k^{\text{Polyak}} = \tfrac{1}{2}\bar{\mathfrak{g}}_k^{\mathcal{D}}.$$

The latter yields the optimal fixed learning rate (up to absolute constant factors) for a noiseless target on a least squares problem [35, 43].[8]

**Exact Line Search.**    In the context of risk, using (8) and noting that $\mathscr{R}(t) = \frac{1}{2d}\sum_{i=1}^d \lambda_i \mathscr{D}_i^2(t)$, we can find $\gamma_t^{\text{line}} \in \arg\min \mathrm{d}\mathscr{R}(t)$; i.e., the greedy learning rate that decreases the risk the most in the next iteration. We call this *exact line search*. Expressions for the learning rates are given in Table 2, (c.f. Appendix F.1 for general losses). Because these come from ODEs, we can use ODE theory to give exact limiting values for the deterministic equivalent of $\mathfrak{g}_k^{\text{line}}$.

**Proposition 3.1.** *[Limiting learning rate; line search on noiseless least squares] Consider the noiseless ($\omega = 0$) least squares problem (7) . Then the learning rate is always lower bounded by*

$$\frac{\lambda_{\min}(K)}{\frac{1}{d}\operatorname{Tr}(K^2)} \leq \gamma_t^{\text{line}} \quad \textit{for all } t \geq 0.$$

*Moreover, suppose $K$ has only two distinct eigenvalues $\lambda_1 > \lambda_2 > 0$, i.e., $K$ has $d/2$ eigenvalues equal to $\lambda_1$ eigenvalues and $d/2$ eigenvalues equal to $\lambda_2$. Then*

$$\frac{\lambda_{\min}(K)}{\frac{1}{d}\operatorname{Tr}(K^2)} \leq \lim_{t\to\infty} \gamma_t^{\text{line}} \leq \frac{2\lambda_{\min}(K)}{\frac{1}{d}\operatorname{Tr}(K^2)}. \tag{13}$$

For a proof and explicit formula for $\lim_{t\to\infty} \gamma_t^{\text{line}}$, see Section F.2. Hence, being greedy for the risk in a sufficiently anisotropic setting will badly underperform Polyak stepsize (see Fig. 2).

---

[8]The Polyak stepsize we analyze in this paper differs slightly from the "classic" stepsize in the literature, that is, $\frac{\mathscr{R}(X_k)-\mathscr{R}(X^*)}{||\nabla\mathscr{R}(X_k)||^2}$. Rather than using this form, we skip an approximation step in the derivation [27] and use the exactly optimal form. Both variations of the Polyak stepsize can be analyzed under our assumptions; the choice was admittedly somewhat arbitrary. (Note that in the case of least squares, the two stepsizes coincide.)

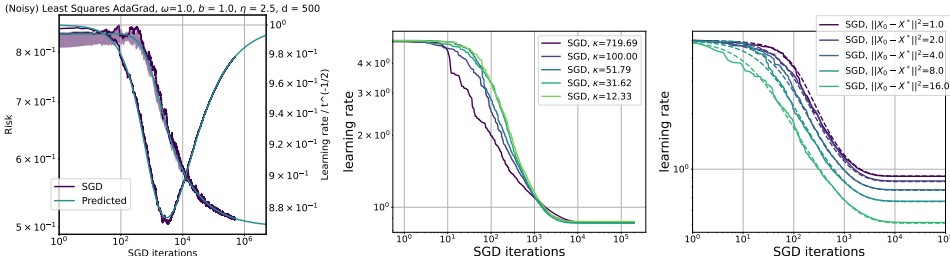

Figure 3: **Quantities effecting AdaGrad-Norm learning rate.** *(left):* Effect of noise ($\omega = 1.0$) on risk (left axis) and learning rate (right axis). Depicted is $\frac{\text{learning rate}}{\text{asymptotic}}$ so it approaches 1. *(Center, right)*: Noiseless least squares ($\omega = 0$). As predicted in Prop. 4.2, $\lim_{t\to\infty} \gamma_t$ depends on avg. eig. of $K$ ($\text{Tr}(K)/d$) and $\|X_0 - X^\star\|^2$ but not $\kappa = \lambda_{\max}/\lambda_{\min}$. See Appendix H for simulation details.

## 4 AdaGrad-Norm analysis

In this section, we analyze the behavior of AdaGrad-Norm learning rate in the least squares setting (see Sec. D for general strongly convex functions). In the presence of additive noise, the AdaGrad-Norm learning rate decays like $t^{-1/2}$, regardless of the data covariance $K$. In contrast, the model with no noise exhibits a learning rate that depends on the spectrum of $K$, as illustrated in Figure 3. The learning rate is bounded below by a constant when $\lambda_{\min}(K) > 0$ is fixed as $d \to \infty$, and we quantify this lower bound. If the limiting spectral measure of $K$ has unbounded density near 0 (e.g. power law spectrum), then the learning rate can approach zero and we quantify the rate of this convergence in the least squares setting as a function of spectral parameters.

For least squares with additive noise, the learning rate asymptotic $\gamma_t \asymp \eta/(b^2 + \frac{\omega^2}{d}\text{Tr}(K)t)^{(1/2)}$ is the fastest decay that AdaGrad-Norm can exhibit. In contrast, the propositions below concern the noiseless case where, for various covariance examples, the decay rate of $\gamma_t$ changes. This is tightly connected to whether the risk is integrable or not. In the simple case of identity covariance, we obtain a closed formula for the trajectory of the integral of the risk and therefore also the learning rate.

**Proposition 4.1.** *In the case of identity covariance ($K = I_d$), the risk solves the differential equation*

$$\frac{\mathrm{d}}{\mathrm{d}t}\mathscr{R}(t) = \frac{\eta^2 \mathscr{R}(t)}{b^2 + 2\int_0^t \mathscr{R}(s)\,\mathrm{d}s} - \frac{2\eta\mathscr{R}(t)}{\sqrt{b^2 + 2\int_0^t \mathscr{R}(s)\,\mathrm{d}s}}, \tag{14}$$

The solution $\int_0^t \mathscr{R}(s)\,\mathrm{d}s$ approaches (from below) a positive constant which yields a computable lower bound to which $\gamma_t$ will converge. Generalizing this to a broader class of covariance matrices, we get the next proposition, which captures the dependence of $\gamma_t$ on $\text{Tr}(K)$.

**Proposition 4.2.** *Suppose $\frac{1}{d}\text{Tr}(K) \le b/\eta$, and that $\int_0^\infty \mathscr{R}(s)\gamma_s\,\mathrm{d}s < \infty$ with $\gamma_s$ as in Table 2 (AdaGrad-Norm for least squares), then $\gamma_t \asymp \frac{1}{\frac{b}{\eta} + \frac{\eta^2}{4d}\text{Tr}(K)\mathscr{D}^2(0)}$.*

An analog of Proposition 4.2 for the strongly convex setting appears in Sec. D (see Prop. D.1). We now consider two cases in which, as $d \to \infty$, there are eigenvalues of $K$ arbitrarily close to 0.

**Proposition 4.3.** *Assume that, for some $C > 0$, the number of eigenvalues of $K$ below $C$ is $o(d)$, and that $\langle X^\star, \omega_i \rangle = O(d^{-1/2})$ for all $i$, (i.e. $X^\star$ is not concentrated in any eigenvector direction). Then, with the initialization $X_0 = 0$, there exists some $\tilde\gamma > 0$ such that $\gamma_t > \tilde\gamma$ for all $t > 0$.*

**Proposition 4.4.** *Let $K$ have a spectrum that converges as $d \to \infty$ to the power law measure $\rho(\lambda) = (1-\beta)\lambda^{-\beta}\mathbf{1}_{(0,1)}$, for some $\beta < 1$[9], and suppose that $\mathscr{D}_i^2(0) \sim \lambda_i^{-\delta}$ for $\delta \ge 0$. Then:*

- *For $1 > \beta + \delta$, there exists $\tilde\gamma$ such that $\gamma_t \ge \tilde\gamma$, and $\mathscr{R}(t) \asymp_{\delta,\beta} t^{\beta+\delta-2}$ for all $t \ge 1$.*

- *For $1 < \beta + \delta < 2$, $\gamma_t \asymp_{\delta,\beta} t_{\delta,\beta}^{-1+\frac{1}{\beta+\delta}}$, and $\mathscr{R}(t) \asymp_{\delta,\beta} t^{-\frac{2}{\beta+\delta}+1}$ for all $t \ge 1$.*

---

[9]Our result can be compared to existing findings for SGD under power-law distributions in [8, 52, 55]. While these works explore similar assumptions regarding the covariance matrix spectrum, they do not address the high-dimensional regime with diverging $\text{Tr}(K)$, focusing primarily on $\beta > 1$.

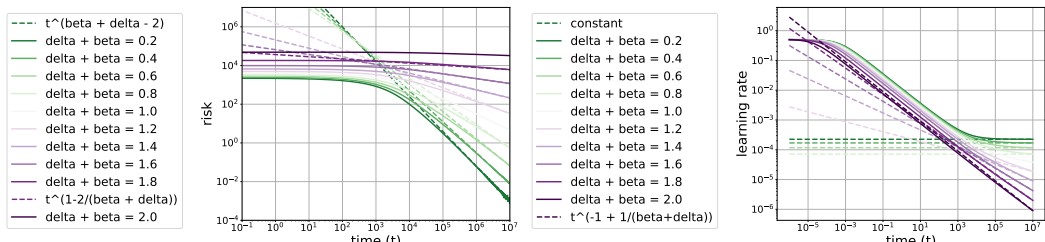

Figure 4: **Power law covariance in AdaGrad Norm** on a least squares problem. Ran exact predictions (ODE) for the risk and learning rate (solid lines). Dashed lines give the predictions from Prop. 4.4 which *match experimental results exactly*. **Phase transition as $\delta + \beta$ varies.** When $\delta + \beta < 1$ (green), the learning rate *(right)* is constant as $t \to \infty$. In contrast, when $2 > \delta + \beta > 1$ (purple), the learning rate decreases at a rate $t^{-1+1/(\beta+\delta)}$ with $\delta + \beta = 1$ (white) where the change occurs. Same phase transition occurs in the sublinear rate of the risk decay *(left)* (see Prop. 4.4).

- *For $1 = \beta + \delta$, $\gamma_t \asymp_{\delta,\beta} \frac{1}{\log(t+1)}$, and $\mathscr{R}(t) \asymp_{\delta,\beta} \left(\frac{t}{\log(t+1)}\right)^{-1}$ for all, $t \geq 1$.*

This proposition shows non-trivial decay of the learning rate is dictated by the residuals (distance to optimality at initialization) and the spectrum of $K$. We note that $\delta = 0$ corresponds to uniform contribution of each mode (e.g. $X_0$ normally distributed). As the eigenmodes of the residuals become more localized, the decay of the learning rate is closer to the behaviour in the presence of additive noise. Furthermore, the scaling behaviour of the loss is affected by the structure of the AdaGrad-Norm algorithm (see Fig. 4). Lastly, constant stepsize SGD yields $\mathscr{R}(t) \asymp t^{\beta+\delta-2}$, with no transition occurring at $\beta + \delta = 1$.

Proofs of the above propositions, in a slightly more general setting, are deferred to Sec. D.

## 5 Conclusions and Limitations

This work studies stochastic adaptive optimization algorithms when data size and parameter size are large, allowing for nonconvex and nonlinear risk functions, as well as data with general covariance structure. The theory shows a concentration of the risk, the learning rate and other key functions to a deterministic limit, which is described by a set of ODEs. The theory is then used to derive the asymptotic behavior of the AdaGrad-Norm and idealized exact line search on strongly convex and least square problems, revealing the influence of the covariance matrix structure on the optimization. A potential extension of this work would be to study other adaptive algorithms such as D-adaptation, DOG, and RMSprop which are covered by the theory. Studying the asymptotic behavior of the risk and the learning rate may improve our understanding of the performance and scalability of these algorithms on more realistic data. Another important application of the theory would be to analyze the ODEs presented here on nonconvex problems.

The current form of the theory is limited to Gaussian data, though many parts of the proof can be extended easily beyond Gaussian data. The main ODE comparison theorem is also only tuned for analyzing problem setups where the trace of the covariance is on the order of the ambient dimension; when the trace of the covariance is much smaller than ambient dimension, other stepsize scalings of SGD are needed. In addition, the analysis is limited to the streaming stochastic adaptive methods. We conjecture that a similar deterministic equivalent holds also for multi-pass algorithms at least for convex problems. This has already been shown in the least square problem for SGD with a fixed deterministic learning rate [43, 45]. Lastly, numerical simulations on real datasets (e.g., CIFAR-5m) suggests that the predicted risk derived by our theory matches the empirical risk of multipass SGD beyond Gaussian data (see for example Figure 6).

## Acknowledgments and Disclosure of Funding

E. Collins-Woodfin was supported by Fonds de recherche du Québec – Nature et technologies (FRQNT) postdoctoral training scholarship and Centre de recherches mathématiques (CRM) Applied math postdoctoral fellowship. Research of B. García Malaxechebarría was in part funded by NSF DMS 2023166 (NSF TRIPODS II). Research by E. Paquette was supported by a Discovery Grant from the Natural Science and Engineering Council (NSERC). C. Paquette is a Canadian Institute for Advanced Research (CIFAR) AI chair, Quebec AI Institute (MILA) and a Sloan Research Fellow in Computer Science (2024). C. Paquette was supported by a Discovery Grant from the Natural Science and Engineering Research Council (NSERC) of Canada, NSERC CREATE grant Interdisciplinary Math and Artificial Intelligence Program (INTER-MATH-AI), Google research grant, and Fonds de recherche du Québec – Nature et technologies (FRQNT) New University Researcher's Start-Up Program. Additional revenues related to this work: C. Paquette has 20% part-time employment at Google DeepMind.

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

# The High Line: Exact Risk and Learning Rate Curves of Stochastic Adaptive Learning Rate Algorithms

## Supplementary material

**Broader Impact Statement.** The work presented in this paper is foundational research and it is not tied to any particular application. The set-up is on a simple well-studied high-dimensional linear composites (e.g., least squares, logistic regression, phase retrieval) with synthetic data and solved using known algorithms, e.g., AdaGrad-Norm. We present deterministic dynamics for the training loss and adaptive stepsizes. The results are theoretical and we do not anticipate any direct ethical and societal issues. We believe the results will be used by machine learning practitioners and we encourage them to use it to build a more just, prosperous world.

## A    SGD adaptive learning rate algorithms and stepsizes

In this section, we write down the explicit update rules for 2 different adaptive stochastic gradient descent algorithms.

**Example: AdaGrad-Norm.**    We begin with AdaGrad-Norm (see Algorithm 1). Note by unraveling the recursion, we have that

$$\mathfrak{g}_k = \frac{\eta}{\sqrt{b^2 + \frac{1}{d^2} \sum_{j=0}^{k} \|\nabla_X \Psi(X_j; a_{j+1}, \epsilon_{j+1})\|^2}}, \tag{15}$$

with the deterministic equivalent (see Section 2 and also C.3) for this learning rate being

$$\gamma_t = \frac{\eta}{\sqrt{b^2 + \frac{\text{Tr}(K)}{d} \int_0^t I(\mathscr{B}(s)) \, \mathrm{d}s}}. \tag{16}$$

In the case of the least squares problem, the quantity $I(\mathscr{B}(t))$ is explicit and

$$\gamma_t = \frac{\eta}{\sqrt{b^2 + \frac{2\text{Tr}(K)}{d} \int_0^t \mathscr{R}(s) \, \mathrm{d}s}}. \tag{17}$$

---

**Algorithm 1** AdaGrad-Norm

---

**Require:** Initialize $\eta > 0$, $X_0 \in \mathbb{R}^d$, $b \in \mathbb{R}$ and set $b_0 = b \times d$
  **for** $k = 1, 2, \ldots,$ **do**
    Generate new sample $a_k \sim \mathcal{N}(0, K)$, $\epsilon_k \sim \mathcal{N}(0, \omega^2)$;
    $b_k^2 \leftarrow b_{k-1}^2 + \|\nabla_X \Psi(X_{k-1}; a_k, \epsilon_k)\|^2$;
    $\mathfrak{g}_{k-1} = d \times \frac{\eta}{|b_k|}$;                                        ▷ updating learning rate
    $X_k \leftarrow X_{k-1} - \frac{\mathfrak{g}_{k-1}}{d} \nabla_X \Psi(X_{k-1}; a_k, \epsilon_k)$;        ▷ updating step with stochastic gradient
  **end for**

---

**Example: RMSprop-Norm**    We consider the "normed" version of RMSprop, that is, where there is only one learning rate parameter.

We consider Algorithm 2 where we put a factor of the learning into the exponential moving average for RMSprop. The deterministic equivalent for $\mathfrak{g}_k$ for Alg. 2 (see Section 2) is

$$\gamma_t = \frac{\eta}{\sqrt{b^2 e^{-\alpha t} + \frac{\text{Tr}(K)}{d} \int_0^t e^{-\alpha(t-s)} I(\mathscr{B}(s)) \, \mathrm{d}s}}. \tag{18}$$

In the case of the least squares problem, the quantity $I(\mathscr{B}(t))$ is explicit and

$$\gamma_t = \frac{\eta}{\sqrt{b^2 e^{-\alpha t} + \frac{2\text{Tr}(K)}{d} \int_0^t e^{-\alpha(t-s)} \mathscr{R}(s) \, \mathrm{d}s}}. \tag{19}$$

**Algorithm 2** RMSprop-Norm, $\alpha$ Exponential Moving Average

---

**Require:** Initialize $\eta > 0$, $X_0 \in \mathbb{R}^d$, $b \in \mathbb{R}$ and set $b_0 = d \times b$, $\alpha > 0$ exponential moving avg.
$\quad \mathfrak{g}_{-1} = d \times \frac{\eta}{b_0}$;
$\quad$ **for** $k = 1, 2, \ldots,$ **do**
$\qquad$ Generate new sample $a_k \sim \mathcal{N}(0, K)$, $\epsilon_k \sim \mathcal{N}(0, \omega^2)$;
$\qquad b_k^2 \leftarrow \alpha \cdot b_{k-1}^2 + (1-\alpha)\|\nabla_X \Psi(X_{k-1}; a_k, \epsilon_k)\|^2$;
$\qquad \mathfrak{g}_{k-1} = d \times \frac{\eta}{|b_k|}$; $\qquad\qquad\qquad\qquad\qquad$ $\triangleright$ updating learning rate
$\qquad X_k \leftarrow X_{k-1} - \frac{\mathfrak{g}_{k-1}}{d} \nabla_X \Psi(X_{k-1}; a_k, \epsilon_k)$; $\qquad$ $\triangleright$ updating step with stochastic gradient
$\quad$ **end for**

---

# B  The Dynamical nexus

In this section, we prove the main theorem on concentration of the risk curves and learning rates. We shall set some notation. In what follows, we again use $W = [X|X^\star] \in \mathbb{R}^{d \times 2}$. We also use $W^+ = [W|X_0] = [X|X^\star|X_0]$.

We shall also use the shorthand $r = \langle a, W \rangle$, and $x = \langle a, X \rangle$ so that $f(\langle a, X \rangle, \langle a, X^\star \rangle; \epsilon) = f(\langle a, W \rangle; \epsilon) = f(r; \epsilon)$.

We shall let $B = B(X) = W^T K W$ be the covariance matrix of the Gaussian vector $r$. We also write $f'$ for the $\partial_x f$.

## B.1  Discussion of the assumptions on $f$

In this section we show how the assumptions we put on $h$ and $I$ are almost satisfied for $L$-smooth $f$. We say that $f$ is $L$-smooth if:
$$\|\nabla f(r_1, \epsilon_1) - \nabla f(r_2, \epsilon_2)\| \le L\sqrt{(\|r_1 - r_2\|^2 + \|\epsilon_1 - \epsilon_2\|^2)},$$
which we note implies $f$ is $\alpha$-pseudo Lipschitz with $\alpha = 1$.

**Lemma B.1.** *1. There exists a function $h : \mathbb{R}^{2 \times 2} \to \mathbb{R}$ such that $h(B(X)) = \mathcal{R}(X)$ is differentiable and satisfies*
$$\nabla_X \mathcal{R}(X) = \mathbb{E}_{a,\epsilon} \nabla_X \Psi(X; a, \epsilon).$$

*Furthermore, $h$ is continuously differentiable on $\{B : \det B \ne 0\}$ and its derivative $\nabla h$ satisfies an estimate*
$$\|\nabla h(B_1) - \nabla h(B_2)\| \le (\sqrt{2} + 1)L(f)\min\{\|B_1^{-1}\|_{op}, \|B_2^{-1}\|_{op}\}\|B_1 - B_2\|_F.$$

*2. The function $I(B) = \mathbb{E}_{a,\epsilon}[(f'(\langle a, X \rangle; \langle a, X^\star \rangle, \epsilon))^2]$ satisfies an estimate*
$$|I(B_1) - I(B_2)| \le L(f)\sqrt{I(B_1) + I(B_2)}\min\{\|B_1^{-1}\|_{op}, \|B_2^{-1}\|_{op}\}\|B_1 - B_2\|_F.$$

*Proof.* To derive the existence of $h$, note that
$$\mathcal{R}(X) = \mathbb{E}(\mathbb{E}(f(\langle a, X \rangle, \langle a, X^\star \rangle, \epsilon)|\epsilon))$$
is an expectation of a Gaussian vector $r = (\langle a, X \rangle, \langle a, X^\star \rangle)$. This vector can be expressed as an image of an iid Gaussian vector $z$ by representing $r = \sqrt{B}z$, and hence we have
$$h(B) \overset{\text{def}}{=} \mathbb{E}(\mathbb{E}(f(\sqrt{B}z, \epsilon)|\epsilon)).$$

As the function $f$ is absolutely continuous with a Lipschitz gradient, we can differentiate under the integral sign and conclude
$$\nabla_X \mathcal{R}(X) = \nabla_X \mathbb{E} f(\langle a, X \rangle, \langle a, X^\star \rangle, \epsilon) = \mathbb{E} \nabla_X f(\langle a, X \rangle, \langle a, X^\star \rangle, \epsilon).$$
For the differentiability of $h$, suppose for the moment that $f$ is $C^2$ with bounded second derivatives.[10] Setting $Q = \sqrt{B}$ the positive semi-definite square root of $B$, we have
$$\partial_{Q_{ij}} h(Q^2) = \mathbb{E}(\mathbb{E}(\partial_{Q_{ij}} f(Qz, \epsilon)|\epsilon)).$$

---

[10]This condition can be removed in a standard way: one creates an $f_\epsilon$ which is an approximation to $f$ formed by convolving with an isotropic Gaussian of variance $\epsilon$. This is $C^2$ and has bounded second derivatives (as $f$ was smooth). One then takes the limit as $\epsilon \to 0$.

Then using the chain rule, and setting $\partial_i f$ to be the $i$-th partial derivative of $f$,
$$\partial_{Q_{ij}} h(Q^2) = \mathbb{E}(\mathbb{E}(z_j \partial_i f(Qz, \epsilon)|\epsilon)) = \mathbb{E}(\mathbb{E}([Q_{ij}\partial_i + Q_{jj}\partial_j]\partial_i f(Qz, \epsilon)|\epsilon)),$$
where we have applied Stein's Lemma. We conclude when $\det Q \neq 0$ by the implicit function theorem that $h$ is differentiable and we have
$$\partial_{Q_{ij}} h(Q^2) = \sum \partial_{kl} h \partial_{Q_{ij}}(Q^2)_{kl} = \sum_l (\partial_{il} h) Q_{jl} + \sum_k (\partial_{kj} h) Q_{ik}.$$

As a matrix equation, this can be written as
$$(Dh)Q + Q(Dh) = JQ \quad \text{where} \quad J_{kl} = \mathbb{E}(\mathbb{E}((\partial_k \partial_l f)(Qz, \epsilon)|\epsilon)).$$
This is a linear equation in $Dh$. When $Q \succ 0$, we can define
$$A = \int_0^\infty e^{-tQ}(JQ)e^{-tQ}\,\mathrm{d}t,$$
and note
$$AQ + QA = -\int_0^\infty \frac{\mathrm{d}}{\mathrm{d}t}\left(e^{-tQ}(JQ)e^{-tQ}\right)\mathrm{d}t = JQ.$$
Moreover, the mapping $M \mapsto \int_0^\infty e^{-tQ} M e^{-tQ}\,\mathrm{d}t$ defines a two-sided inverse for $M \mapsto MQ + QM$, and so $Dh = A$. Note that by symmetry of $J$, $Q$, and $Dh$
$$JQ = (Dh)Q + Q(Dh) = QJ,$$
and therefore
$$(Dh)Q + Q(Dh) = \frac{1}{2}(JQ + QJ),$$
and so taking inverses on both sides, $Dh = J$.

Undoing Stein's Lemma, we have $Q(Dh) = (Dh)Q = M$, where $M_{ij} = \mathbb{E}(\mathbb{E}(z_j \partial_i f(Qz, \epsilon)|\epsilon))$. From $L$-smoothness of $f$
$$\|M(Q_1) - M(Q_2)\| \leq L\,\mathbb{E}(\|z\|\|Q_1 z - Q_2 z\|) \leq \sqrt{2}L\|Q_1 - Q_2\|_F.$$
Hence
$$\begin{aligned}\|Dh(Q_1^2) - Dh(Q_2^2)\| &= \|Q_1^{-1}M(Q_1) - Q_2^{-1}M(Q_2)\| \\ &\leq \|Q_1^{-1}\|_{op}\|M(Q_1) - Q_1 Q_2^{-1}M(Q_2)\| \\ &\leq \|Q_1^{-1}\|_{op}\left(\|M(Q_1) - M(Q_2)\| + \|(Q_2 - Q_1)Q_2^{-1}M(Q_2)\|\right).\end{aligned}$$
Note $Q_2^{-1}M(Q_2) = (Dh)(Q_2^2)$ is bounded by $L(f)$, and so we arrive at
$$\begin{aligned}\|Dh(Q_1^2) - Dh(Q_2^2)\| &\leq (\sqrt{2}+1)L(f)\|Q_1^{-1}\|_{op}\|Q_1 - Q_2\|_F \\ &\leq (\sqrt{2}+1)L(f)\|Q_1^{-2}\|_{op}\|Q_1^2 - Q_2^2\|_F.\end{aligned}$$
We note the bound is symmetric in $Q_1$ and $Q_2$, and by density of $C^2$ in space of $C^{1,lip}$, this holds for $L$-smooth $f$. This concludes the estimates for the derivative of $h$.

For the Fisher matrix, $I(B)$, from $L$-smoothness, we have again with $Q = \sqrt{B}$,
$$I(Q^2) = \mathbb{E}(\mathbb{E}((\partial_1 f(Qz, \epsilon))^2|\epsilon)).$$
Then
$$|I(Q_1^2) - I(Q_2^2)| \leq \left|\mathbb{E}(\mathbb{E}((\partial_1 f(Q_1 z, \epsilon))^2 - (\partial_1 f(Q_2 z, \epsilon))^2|\epsilon))\right|.$$
Applying Cauchy-Schwarz and using the $L$-smoothness of $f$,
$$|I(Q_1^2) - I(Q_2^2)| \leq \sqrt{I(Q_1^2) + I(Q_2^2)} \times L(f)\|Q_1 - Q_2\|_F.$$
$\square$

This lemma shows that an $L$-smooth function nearly satisfies Assumption 3 and 4 provided that $\|B^{-1}\|_{op}$ is bounded. Therefore, our concentration result Theorem B.1 and its Corollaries will hold provided we add a stopping time. Fix $M > 0$ and let
$$\hbar_M(B) \overset{\text{def}}{=} \inf\{t > 0 \,:\, \|B^{-1}\|_{op} > M\}.$$
Then the concentration of the risk under SGD to a deterministic function, Theorem B.1, holds with $t$ replaced with $t \wedge \hbar_M(B) \wedge \hbar_M(\mathscr{B})$. The corollaries of Theorem B.1 also follow under this added stopping time.

In the next section, we prove this concentration theorem, Theorem B.1.

## B.2 Integro-differential equation for $\mathscr{S}(t,z)$

A goal of this paper is to show that quadratic statistics $\varphi : \mathbb{R}^d \to \mathbb{R}$ applied to SGD converge to a deterministic function. This argument hinges on understanding the deterministic dynamics of one important statistic, defined as

$$S(W,z) = W^\top R(z;K)W,$$

applied to $W_{\lfloor td \rfloor}$ (SGD updates). Here $W = [X | X^\star]$ and $R(z;K) = (K - zI_d)^{-1}$ for $z \in \mathbb{C}$ is the resolvent of the matrix $K$. The statistic $S(W,z)$ is valuable because it encodes many other important quantities including $W^\top q(K)W$ for all polynomials $q$. We show that $S(W_{\lfloor td \rfloor}, z)$, is close to a deterministic function $(t,z) \mapsto \mathscr{S}(t,z)$ which satisfies an integro-differential equation.

To introduce the integro-differential equation, recall by Assumptions 3 and 4

$$\mathcal{R}(X) = h \circ B(W) \quad \text{and} \quad \mathbb{E}_{a,\epsilon}[f'(a^\top W)^2] = I \circ B(W) \quad \text{with} \quad B(W) = W^\top K W,$$

and $\alpha$-pseudo-Lipschitz functions $h : \mathbb{R}^{2 \times 2} \to \mathbb{R}$ differentiable and $I : \mathbb{R}^{2 \times 2} \to \mathbb{R}$. It will be useful, throughout the remaining paper, to express $\nabla h$ explicitly as a $2 \times 2$ matrix, that is,

$$\nabla h \cong \left[ \begin{array}{c|c} \nabla h_{11} & \nabla h_{12} \\ \hline \nabla h_{21} & \nabla h_{22} \end{array} \right].$$

With these recollections, the integro-differential equation is defined below.

---

**Integro-Differential Equation for $\mathscr{S}(t,z)$.** For any contour $\Omega \subset \mathbb{C}$ enclosing the eigenvalues of $K$, we have an expression for the derivative of $\mathscr{S}$:

$$\mathrm{d}\mathscr{S}(t,\cdot) = \mathcal{F}(z, \mathscr{S}(t,\cdot)) \, \mathrm{d}t \tag{20}$$

where $\mathcal{F}(z, \mathscr{S}(t,\cdot)) \stackrel{\text{def}}{=} -2\gamma_t \left( \left( \frac{-1}{2\pi i} \oint_\Omega \mathscr{S}(t,z) \, \mathrm{d}z \right) H(\mathscr{B}(t)) \right.$

$$+ H^T(\mathscr{B}(t)) \left( \frac{-1}{2\pi i} \oint_\Omega \mathscr{S}(t,z) \, \mathrm{d}z \right) \bigg)$$

$$+ \frac{\gamma_t^2}{d} \left[ \begin{array}{c|c} \mathrm{Tr}(KR(z;K))I(\mathscr{B}(t)) & 0 \\ \hline 0 & 0 \end{array} \right] \tag{21}$$

$$- \gamma_t (\mathscr{S}(t,z)(2zH(\mathscr{B}(t))) + (2zH^T(\mathscr{B}(t)))\mathscr{S}(t,z)).$$

Here $\mathscr{B}(t) = \frac{-1}{2\pi i} \oint_\Omega z\mathscr{S}(t,z) \, \mathrm{d}z$, $\quad H(\mathscr{B}) = \left[ \begin{array}{c|c} \nabla h_{11}(\mathscr{B}) & 0 \\ \hline \nabla h_{21}(\mathscr{B}) & 0 \end{array} \right]$,

$\gamma_t$ is defined in (9), and the initialization is $\quad \mathscr{S}(0,z) = W_0^\top R(z;K)W_0.$ (22)

The functions $h : \mathbb{R}^{2 \times 2} \to \mathbb{R}$ and $I : \mathbb{R}^{2 \times 2} \to \mathbb{R}$ are defined in Assumption 3 and Assumption 4, respectively.

---

We first note that there is an actual solution to the integro-differential equation. This solution is the same as the ODEs defined in the introduction (see (9)) and proved in [15, Lemma 4.1].

**Lemma B.2** (Equivalence to coupled ODEs.). *The unique solution of (21) with initial condition (22) is given by*

$$\mathscr{S}(t,z) = \frac{1}{d} \sum_{i=1}^d \frac{1}{\lambda_i - z} \mathscr{V}_i(t).$$

In this section, we will be working with approximate solutions to the integro-differential equation (20) (see below for specifics). For working with these solutions, we introduce some notation. We shall always work on a fixed contour $\Omega$ surrounding the spectrum of $K$, given by $\Omega \stackrel{\text{def}}{=} \{z : |z| = \max\{1, 2\|K\|_{\mathrm{op}}\}\}$. We note that this contour is always distance at least $\frac{1}{2}$ from the spectrum of $K$. We define a norm, $\| \cdot \|_\Omega$, on a continuous function $A : \mathbb{C} \to \mathbb{R}$ as

$$\|A\|_\Omega = \max_{z \in \Omega} \|A(z)\|. \tag{23}$$

**Definition B.1** (($\varepsilon, M, T$)-approximate solution to the integro-differential equation)**.** For constants $M, T, \varepsilon > 0$, we call a continuous function $\mathcal{S} : [0, \infty) \times \mathbb{C} \to \mathbb{R}^{2 \times 2}$ an $(\varepsilon, M, T)$-*approximate solution* of (20) if with

$$\hat{\tau}_M(\mathcal{S}) \overset{\text{def}}{=} \inf \left\{ t \geq 0 : \|\mathcal{S}(t, \cdot)\|_\Omega > M \right\},$$

then

$$\sup_{0 \leq t \leq (\hat{\tau}_M \wedge T)} \left\| \mathcal{S}(t, \cdot) - S(0, \cdot) - \int_0^t \mathcal{F}(\cdot, \mathcal{S}(s, \cdot)) \, ds \right\|_\Omega \leq \varepsilon$$

and $\mathcal{S}(0, \cdot) = W_0^\top R(\cdot, K) W_0$, where $W_0 = [X_0 | X^\star]$ is the initialization of SGD.

We suppress the $\mathcal{S}$ in the notation for $\hat{\tau}_M$, that is, $\hat{\tau}_M = \hat{\tau}_M(\mathcal{S})$, when the function $\mathcal{S}$ is clear from the context.

We are now ready to state and prove one of our main results.

**Theorem B.1** (Concentration of SGD and deterministic function $\mathscr{S}(t, z)$)**.** *Suppose the risk function* $\mathcal{R}(X)$ *(2) satisfies Assumptions 2, 3, and 4. Suppose the learning rate satisfies Assumption 6, and the initialization* $X_0$ *and hidden parameters* $X^\star$ *satisfy Assumption 5. Moreover the data* $a \sim \mathcal{N}(0, K)$ *and label noise* $\epsilon$ *satisfy Assumption 1. Let* $\{W_{\lfloor td \rfloor}\}$ *be generated from the iterates of SGD. Then there is an* $\varepsilon > 0$ *so that for any* $T, M > 0$ *and* $d$ *sufficiently large, with overwhelming probability*

$$\sup_{0 \leq t \leq T \wedge \hat{\tau}_M(S(W, \cdot)) \wedge \hat{\tau}_M(\mathscr{S})} \| S(W_{\lfloor td \rfloor}, \cdot) - \mathscr{S}(t, \cdot) \|_\Omega \leq d^{-\varepsilon}, \tag{24}$$

*where the deterministic function* $\mathscr{S}(t, z)$ *solves the integro-differential equation (20).*

*Proof.* By Proposition C.1, for any $M$ and $T$, we can find a $\tilde{\varepsilon} > 0$ such that the function $S(W_{td}, z)$ is an $(d^{-\tilde{\varepsilon}}, M, T)$-approximate solution. (For the deterministic function $\mathscr{S}$, it is an $(0, M, T)$-approximate solution by definition.) We now apply the stability result, [15, Prop. 4.1], to conclude that there exists a $\varepsilon > 0$ such that

$$\sup_{0 \leq t \leq T \wedge \hat{\tau}_M} \| \mathscr{S}(t, z) - S(W_{td}, z) \|_\Omega \leq d^{-\varepsilon}, \quad w.o.p, \tag{25}$$

where $\hat{\tau}_M$ is shorthand for $\hat{\tau}_M(S(W, \cdot)) \wedge \hat{\tau}_M(\mathscr{S})$. The result immediately follows. $\qquad\square$

**Corollary B.1.** *Suppose the assumptions of Theorem B.1 hold. Let* $f$ *be an* $\alpha$-*pseudo-Lipschitz function with* $\alpha \leq 1$ *and let* $q$ *be a polynomial. Set*

$$\varphi(X) \overset{\text{def}}{=} f(W^T q(K) W), \quad \phi(t) \overset{\text{def}}{=} f\left( \frac{-1}{2\pi i} \oint_\Omega q(z) \mathscr{S}(t, z) \, dz \right), \quad \text{where } \mathscr{S}(t, z) \text{ solves (20)}.$$

*Then there is an* $\varepsilon > 0$ *such that for* $d$ *sufficiently large, with overwhelming probability,*

$$\sup_{0 \leq t \leq T} |\varphi(X_{td}) - \phi(t)| \leq d^{-\varepsilon}.$$

*Proof.* This is basically equivalent to [15, Corollary 4.2]. The only difference is that [15, Corollary 4.2] requires the boundedness of $\mathcal{N}$; however, since our function $f$ is $\alpha$-pseudo-Lipschitz with $\alpha \leq 1$, this boundedness follows from [15, Proposition 1.2], and the rest of the proof is identical to the one in [15]. $\qquad\square$

**Remark B.1.** *The learning rate* $\mathfrak{g}_k$, *technically, is not a function of* $W^T q(K) W$. *However, Assumption 6 ensures that the learning rate concentrates around a function* $W^T q(K) W$. *Therefore, Corollary B.1 applies to the learning rate.*

## C  SGD-AL is an approximate solution

We introduce a rescaling of time to relate the $k$-th iteration of SGD to the continuous time parameter $t$ in the differential equation through the relationship $k = \lfloor td \rfloor$. Thus, when $t = 1$, SGD has done exactly $d$ updates. Since the parameter $t$ is continuous and the iteration counter $k$ (integer) discrete,

to simplify the discussion below, we *extend* $k$ to continuous values through the floor operation, $X_k \stackrel{\text{def}}{=} X_{\lfloor k \rfloor}$. Using the continuous parameter $t$, the iterates are related by $X_{td} = X_{\lfloor td \rfloor}$.

The paper [15] provides a net argument showing that we do not need to work with every $z$ on the contour $\Omega$ defining the integro-differential equation, but only polynomially many in $d$. Recall that $\Omega = \{z \,:\, |z| = \max\{2\|K\|_{\text{op}}, 1\}\}$. For a fixed $\xi > 0$, we say that $\Omega_\xi$ is a $d^{-\xi}$-*mesh of* $\Omega$ if $\Omega_\xi \subset \Omega$ and for every $z \in \Omega$ there exists a $\bar z \in \Omega_\xi$ such that $|z - \bar z| < d^{-\xi}$. We can achieve this with $\Omega_\xi$ having cardinality, $|\Omega_\xi| = C(|\Omega|)d^\xi$.

**Lemma C.1** (Net argument, [15], Lemma 5.1). *Fix $T, M > 0$ and let $\xi > 0$. Suppose $\Omega_\xi$ is a $d^{-\xi}$ mesh of $\Omega$ with $|\Omega_\xi| = C \cdot d^\xi$ and positive $C > 0$. Let the function $S(t, z) = S(W_{td}, z)$ satisfy*

$$\sup_{0 \le t \le (\hat\tau_M \wedge T)} \|S(t, \cdot) - S(0, \cdot) - \int_0^t \mathcal{F}(\cdot, S(s, \cdot)) \, \mathrm{d}s\|_{\Omega_\xi} \le \varepsilon \tag{26}$$

*with $\hat\tau_M = \inf\{t \ge 0 \,:\, \|S(t, \cdot)\|_\Omega > M\}$. Then $S$ is a $(\varepsilon + C(M, T, \|K\|_{\text{op}})d^{-\xi}, M, T)$-approximate solution to the integro-differential equation, that is,*

$$\sup_{0 \le t \le (\hat\tau_M \wedge T)} \|S(t, \cdot) - S(0, \cdot) - \int_0^t \mathcal{F}(\cdot, S(s, \cdot)) \, \mathrm{d}s\|_\Omega \le \varepsilon + C \cdot d^{-\xi},$$

*where $C = C(M, T, \|K\|_{\text{op}}, L(I), L(h))$ is a positive constant.*

(We prove in Section C.1 that $S(t, z)$ does indeed satisfy inequality (26).) We also cite the following lemma, which relates two stopping times used throughout this paper.

**Lemma C.2** (Stopping time, [15], Lemma 4.2). *For a constant $C$ depending on $\|K\|_{\text{op}}$, we have*

$$C \le \frac{\|S(W_{td}, \cdot)\|_\Omega}{\|W_{td}\|^2} \le 2.$$

**Remark C.1.** *Fix $M > 0$ and define the stopping time on $\|W_{td}\|$, $\vartheta = \vartheta_M$, by*

$$\vartheta_M(W_{td}) \stackrel{\text{def}}{=} \inf\{t \ge 0 : \|W_{td}\|^2 > M\}.$$

*Due to the previous lemma, any stopping time $\hat\tau_M$ defined on $\|S(t, \cdot)\|_\Omega$ corresponds to a stopping time $\vartheta$ on $\|W_{td}\|$, that is, for $c = C^{-1}$, $\hat\tau_M \le \vartheta_{cM}$.*

## C.1 SGD-AL is an approximated solution

**Proposition C.1** (SGD-AL is an approximate solution). *Fix a $T, M > 0$ and $0 < \varepsilon < \delta/8$, where $\delta$ is defined in Assumption 6. Then $S(W_{td}, z)$ is a $(d^{-\varepsilon}, M, T)$-approximate solution w.o.p., that is,*

$$\sup_{0 \le t \le (T \wedge \tau_M)} \|S(W_{td}, z) - S(W_0, z) - \int_0^t \mathcal{F}(z, S(W_{sd}, z)) \, \mathrm{d}s\|_\Omega \le d^{-\varepsilon}. \tag{27}$$

Again, the proof is very similar to [15, Prop. 5.2]. The one difference is that the martingales and error terms are slightly more involved, because of the non-deterministic stepsize we are using. The remainder of this section, along with section C.2, fills in the details of bounding these lower-order terms, so that the proof can proceed as in [15].

### C.1.1 Shorthand notation

In the following sections, we will be using various versions of the stepsize $\gamma$. In order to simplify notation, we set

$$\gamma(G_k) = \gamma(k, N_k(d \times \cdot), G_k(d \times \cdot), Q_k(d \times \cdot)),$$
$$\gamma(\mathcal{G}_k) = \gamma(k, N_k(d \times \cdot), \mathcal{G}_k(d \times \cdot), Q_k(d \times \cdot)),$$
$$\gamma(B_k) = \gamma(k, N_k(d \times \cdot), \text{Tr}(K)I(B_k(d \times \cdot))/d, Q_k(d \times \cdot)).$$

Further, setting $\Delta_k \stackrel{\text{def}}{=} f'(r_k)a_{k+1}$, define

$I_1(k) \stackrel{\text{def}}{=} \Delta_k^\top \nabla^2 \varphi(X_k) \Delta_k/d, \ I_2(k) \stackrel{\text{def}}{=} \text{Tr}(\nabla^2 \varphi(X_k)K)\mathbb{E}\left[f'(r_k)^2 \,|\, \mathcal{F}_k\right]/d, \ I_3(k) \stackrel{\text{def}}{=} \nabla\varphi(X_k)^\top \Delta_k.$

The normalization here (dividing by $d$) is chosen so that the $I$ terms are all $O(1)$; this is formally shown in Lemma C.5.

### C.1.2 SGD-AL under the statistic

We follow the approach in [15, Section 5.3] to rewrite the SGD adaptive learning rate update rule as an integral equation. Considering a quadratic function $\varphi : \mathbb{R}^d \to \mathbb{R}$ and performing Taylor expansion, we obtain

$$\varphi(X_{k+1}) = \varphi(X_k) - \frac{\gamma(G_k)}{d} \nabla\varphi(X_k)^\top \Delta_k + \frac{\gamma(G_k)^2}{2d^2} \Delta_k^\top \nabla^2 \varphi(X_k)\Delta_k. \tag{28}$$

We will now relate this equation to its expectation by performing a Doob decomposition, involving the following martingale increments and error terms:

$$\Delta\mathcal{M}_k^{\mathrm{grad}}(\varphi) \overset{\mathrm{def}}{=} \frac{1}{d} \left( -\gamma(G_k)I_3(k) + \mathbb{E}\left[ \gamma(G_k)I_3(k) \,\middle|\, \mathcal{F}_k \right] \right), \tag{29}$$

$$\Delta\mathcal{M}_k^{\mathrm{Hess}}(\varphi) \overset{\mathrm{def}}{=} \frac{1}{2d} \left( \gamma(G_k)^2 I_1(k) - \mathbb{E}\left[ \gamma(G_k)^2 I_1(k) \,\middle|\, \mathcal{F}_k \right] \right), \tag{30}$$

$$\mathbb{E}\left[\mathcal{E}_k^{\mathrm{Hess}}(\varphi) \,\middle|\, \mathcal{F}_k\right] \overset{\mathrm{def}}{=} \frac{1}{2d} \left( \mathbb{E}\left[ \gamma(G_k)^2 I_1(k) \,\middle|\, \mathcal{F}_k \right] - \gamma(B_k)^2 I_2(k) \right), \tag{31}$$

$$\mathbb{E}\left[\mathcal{E}_k^{\mathrm{grad}}(\varphi) \,\middle|\, \mathcal{F}_k\right] \overset{\mathrm{def}}{=} \frac{1}{d} \left( -\mathbb{E}\left[ \gamma(G_k)I_3(k) \,\middle|\, \mathcal{F}_k \right] + \gamma(B_k)\nabla\varphi(X_k)^\top \nabla\mathcal{R}(X_k) \right). \tag{32}$$

We can then write

$$\varphi(X_{k+1}) = \varphi(X_k) - \frac{\gamma(B_k)}{d} \nabla\varphi(X_k)^\top \nabla\mathcal{R}(X_k) + \frac{\gamma(B_k)^2}{2d^2} \mathrm{Tr}(\nabla^2\varphi(X_k)K)\mathbb{E}\left[ f'(r_k)^2 \,\middle|\, \mathcal{F}_k \right]$$
$$+ \Delta\mathcal{M}_k^{\mathrm{grad}}(\varphi) + \Delta\mathcal{M}_k^{\mathrm{Hess}}(\varphi) + \mathbb{E}\left[\mathcal{E}_k^{\mathrm{Hess}}(\varphi) \,\middle|\, \mathcal{F}_k\right] + \mathbb{E}\left[\mathcal{E}_k^{\mathrm{grad}}(\varphi) \,\middle|\, \mathcal{F}_k\right].$$

Extending $X_k$ into continuous time by defining $X_t = X_{\lfloor t \rfloor}$, we sum up (integrate). For this, we introduce the forward difference

$$(\Delta\varphi)(X_j) \overset{\mathrm{def}}{=} \varphi(X_{j+1}) - \varphi(X_j),$$

giving us

$$\varphi(X_{td}) = \varphi(X_0) + \sum_{j=0}^{\lfloor td \rfloor - 1} (\Delta\varphi)(X_j) \overset{\mathrm{def}}{=} \varphi(X_0) + \int_0^t d \cdot (\Delta\varphi)(X_{sd}) \, \mathrm{d}s + \xi_{td},$$

where $|\xi_{td}| = \left| \int_{(\lfloor td \rfloor - 1)/d}^t d \cdot \Delta\varphi(X_{sd}) \, \mathrm{d}s \right| \leq \max_{0 \leq j \leq \lceil td \rceil} \{|\Delta\varphi(X_j)|\}$. With this, we obtain the Doob decomposition for SGD-AL:

$$\varphi(X_{td}) = \varphi(X_0) - \int_0^t \gamma(B_{sd})\nabla\varphi(X_{sd})^\top \nabla\mathcal{R}(X_{sd}) \, \mathrm{d}s \tag{33}$$

$$+ \frac{1}{2d} \int_0^t \gamma(B_{sd})^2 \, \mathrm{Tr}(K\nabla^2\varphi(X_{sd}))\mathbb{E}\left[ f'(r_{sd})^2 \,\middle|\, \mathcal{F}_{sd} \right] \, \mathrm{d}s$$

$$+ \sum_{j=0}^{\lfloor td \rfloor - 1} \mathcal{E}_j^{\mathrm{all}}(\varphi),$$

$$\text{with} \quad \mathcal{E}_j^{\mathrm{all}}(\varphi) = \Delta\mathcal{M}_j^{\mathrm{grad}}(\varphi) + \Delta\mathcal{M}_j^{\mathrm{Hess}}(\varphi) \tag{34}$$
$$+ \mathbb{E}\left[\mathcal{E}_j^{\mathrm{Hess}}(\varphi) \,\middle|\, \mathcal{F}_j\right] + \mathbb{E}\left[\mathcal{E}_j^{\mathrm{grad}}(\varphi) \,\middle|\, \mathcal{F}_j\right]$$
$$+ \xi_{td}(\varphi).$$

From here, we can proceed as in [15, Section 5.3] to show that SGD-AL is an $(\varepsilon, M, T)$-approximated solution.

### C.1.3 $S(W_{td}, z)$ is an approximate solution

*Proof of Proposition C.1.* The appropriate stepsize, as a function of $W_{td}$, is

$$\gamma_t = \gamma(td, N_{td}, \mathrm{Tr}(K)I(B_{td})/d, Q_{td}).$$

(Note that $N$, $I$ and $Q$ can all be found as functions of $S(W_{td}, \cdot)$ using contour integration.) It is shown in the proof of [15, Proposition 5.2] that given the analogue of (33) for deterministic stepsize, $S(W_{td}, \cdot)$ satisfies

$$S(W_{td}, z) = S(W_0, z) + \int_0^t \mathcal{F}(z, S(W_{sd}, z)) \, ds + \sum_{i=0}^{\lfloor td \rfloor - 1} \mathcal{E}_j^{\text{all}}(S).$$

The only terms of (33) that differ in our case are the martingale and error terms. Thus to show that $S(W_{td}, \cdot)$ is an approximate solution of the integro-differential equation (20) all we need is to bound the martingales and error terms contained in $\mathcal{E}_j^{\text{all}}$. Let $\Omega = \{z \,:\, |z| = \max\{1, 2\|K\|_{\text{op}}\}\}$, as previously. We thus have that for all $z \in \Omega$,

$$\sup_{0 \leq t \leq T \wedge \hat{\tau}_M} \left| S(W_{td}, z) - S(W_0, z) - \int_0^t \mathcal{F}(z, S(W_{sd}, z)) \, ds \right| \leq \sup_{0 \leq t \leq T \wedge \hat{\tau}_M} \|\mathcal{E}_{td}^{\text{all}}(S(\cdot, z))\|. \quad (35)$$

Next, fix a constant $\xi > 0$. Let $\Omega_\xi \subset \Omega$ such that there exists a $\bar{z} \in \Omega_\xi$ such that $|z - \bar{z}| \leq d^{-\xi}$ and the cardinality of $\Omega_\xi$, $|\Omega_\xi| = C d^\xi$ where $C > 0$ can depend on $\|K\|_{\text{op}}$. For all $z \in \Omega$, we note that $\hat{\tau}_M \leq \vartheta_{cM}$ (see Lemma C.2). Consequently, we evaluate the error with the stopped process $W_{td}^\vartheta \stackrel{\text{def}}{=} W_{d(t \wedge \vartheta)}$ instead of using $\hat{\tau}_M$. By Proposition C.2, the proof of which we have deferred to Section C.2, we have, for any $\hat{\delta} > 0$

$$\sup_{z \in \Omega_\xi} \sup_{0 \leq t \leq T \wedge \vartheta_{cM}} \|\mathcal{E}_{dt}^{\text{all}}(S(\cdot, z))\| \leq d^{-\delta/4 + \hat{\delta}} \quad \text{w.o.p.} \quad (36)$$

We deduce that

$$\sup_{0 \leq t \leq T \wedge \hat{\tau}_M} \|S(W_{td}, z) - S(W_0, z) - \int_0^t \mathcal{F}(z, S(W_{sd}, z)) \, ds\|_{\Omega_\xi} \leq d^{\hat{\delta} - \delta/4} \quad \text{w.o.p.}$$

An application of the net argument, Lemma C.1, finishes the proof after setting $\hat{\delta} = \delta/8$ and $\xi = \delta/8$. $\qquad \square$

## C.2 Error bounds

All the martingale and error terms (34) go to $0$ as $d$ grows. Formally,

**Proposition C.2.** *Let the function $f$ be defined as in Assumption 2. Let the statistic $S : [0, \infty) \times \mathbb{C} \to \mathbb{R}^{2 \times 2}$ be defined as*

$$S(t, z) = W_{\lfloor td \rfloor}^\top R(z; K) W_{\lfloor td \rfloor}, \quad (37)$$

*where $W = [X | X^\star]$. Then, for any $z \in \Omega$ and $T, M, \zeta > 0$, with overwhelming probability,*

$$\sup_{0 \leq t \leq T \wedge \vartheta} \left\| \mathcal{E}_{dt}^{all}(S(\cdot, z)) \right\| \leq d^{-\delta/4 + \zeta},$$

*where to suppress notation we use $\vartheta$ as shorthand for $\vartheta_{cM}$, and $c$ is the constant from Lemma C.2.*

*Proof.* This follows from combining Propositions C.3, C.4, C.5, C.6, and C.7. $\qquad \square$

The remainder of this subsection is devoted to proving these supporting propositions; throughout these proofs we will work with the stopping time $\vartheta$ as defined in the proposition above.

### C.2.1 Bounds on the lower order terms in the gradient and hessian

**Proposition C.3** (Hessian error term)**.** *Let $f$ and $S$ be defined as in Assumption 2 and (37). Then, for any $z \in \Omega$, $T > 0$ and $\zeta > 0$, with overwhelming probability,*

$$\sup_{0 \leq t \leq T \wedge \vartheta} \sum_{k=0}^{\lfloor td \rfloor - 1} \left\| \mathbb{E}\left[ \mathcal{E}_k^{Hess}(S(\cdot, z)) \mid \mathcal{F}_k \right] \right\| \leq d^{-\delta/4 + \zeta}.$$

*Proof.* For arbitrary $z \in \Omega$ and $k \leq (T \wedge \vartheta)d - 1$, set $\varphi(X) = S_{ij}(W, z)$ to be the $ij$-th entry of the matrix $S(W, z)$. Then

$$
\begin{aligned}
2d\, \mathbb{E}[\mathcal{E}_k^{\text{Hess}}(\varphi) \,|\, \mathcal{F}_k] &= \mathbb{E}\left[\gamma(G_k)^2 I_1(k) \,|\, \mathcal{F}_k\right] - \gamma(B_k)^2 I_2(k) \\
&= \mathbb{E}[(\gamma(G_k)^2 - \gamma(\mathcal{G}_k)^2)I_1(k) \,|\, \mathcal{F}_k] \\
&\quad + (\gamma(\mathcal{G}_k)^2 - \gamma(B_k)^2)\,\mathbb{E}[I_1(k) \,|\, \mathcal{F}_k] + \gamma(B_k)^2\, \mathbb{E}[(I_1(k) - I_2(k) \,|\, \mathcal{F}_k] \\
&= \mathcal{E}_1 + \mathcal{E}_2 + \mathcal{E}_3.
\end{aligned}
$$

We look at $|\mathcal{E}_1|$ first.

$$
\begin{aligned}
|\mathcal{E}_1| &= \left|\mathbb{E}\left[(\gamma(G_k)^2 - \gamma(\mathcal{G}_k)^2)I_1(k) \,|\, \mathcal{F}_k\right]\right| \\
&\leq \mathbb{E}\left[\left|(\gamma(G_k)^2 - \gamma(\mathcal{G}_k)^2)\right|^2 \,|\, \mathcal{F}_k\right]^{\frac{1}{2}} \cdot \mathbb{E}\left[\left|I_1(k)\right|^2 \mathcal{F}_k\right]^{\frac{1}{2}} \\
&\leq \mathbb{E}\left[\left|\gamma(G_k) + \gamma(\mathcal{G}_k)\right|^{\frac{7}{2}} \left|\gamma(G_k) - \gamma(\mathcal{G}_k)\right|^{\frac{1}{2}} \,|\, \mathcal{F}_k\right]^{\frac{1}{2}} \cdot \mathbb{E}\left[\left|I_1(k)\right|^2 \mathcal{F}_k\right]^{\frac{1}{2}} \\
&\leq \mathbb{E}\left[\left|\gamma(G_k) + \gamma(\mathcal{G}_k)\right|^7 \,|\, \mathcal{F}_k\right]^{\frac{1}{4}} \cdot \mathbb{E}\left[\left|\gamma(G_k) - \gamma(\mathcal{G}_k)\right| \,|\, \mathcal{F}_k\right]^{\frac{1}{4}} \cdot \mathbb{E}\left[\left|I_1(k)\right|^2 \,|\, \mathcal{F}_k\right]^{\frac{1}{2}}.
\end{aligned}
$$

For the first term, we use (6). We have

$$
\mathbb{E}\left[\left|\gamma(G_k) + \gamma(\mathcal{G}_k)\right|^7 \,|\, \mathcal{F}_k\right] \leq \hat{C}(\gamma) \cdot \mathbb{E}\left[\left|2 + 2\|N_k\|_\infty^\alpha + 2\|Q_k\|_\infty^\alpha + \|G_k\|_\infty^\alpha + \|\mathcal{G}_k\|_\infty^\alpha\right|^7 \,|\, \mathcal{F}_k\right].
$$

All the terms inside the expectation, apart from $\|G_k\|_\infty^\alpha$, are deterministic with respect to $\mathcal{F}_k$ and bounded by a constant independent of $d$ (see Lemma C.6). Since we know from Lemma C.6 that for any $\varepsilon > 0$, all moments of $\|G_k\|_\infty$ are bounded by $d^\varepsilon$ w.o.p., we conclude

$$
\mathbb{E}\left[\left|\gamma(G_k) + \gamma(\mathcal{G}_k)\right|^7 \,|\, \mathcal{F}_k\right] \leq d^\varepsilon \quad \text{w.o.p.}
$$

For the second term, we use (5). Again, since $\|N_k\|_\infty$ and $\|Q_k\|_\infty$ are bounded due to our stopping time, we have

$$
\mathbb{E}\left[\left|\gamma(G_k) - \gamma(\mathcal{G}_k)\right| \,|\, \mathcal{F}_k\right]^{\frac{1}{4}} \leq d^{-\delta/4}.
$$

The last term, $\mathbb{E}\left[\left|I_1(k)\right|^2 \,|\, \mathcal{F}_k\right]^{\frac{1}{2}}$, is also bounded by a constant (see Lemma C.5), and all together, we find that $|\mathcal{E}_1| \leq d^{\varepsilon - \delta/4}$ with overwhelming probability.

Now let us consider $|\mathcal{E}_2|$:

$$
|\mathcal{E}_2| = |(\gamma(\mathcal{G}_k)^2 - \gamma(B_k)^2)\,\mathbb{E}[I_1(k) \,|\, \mathcal{F}_k]| = |\gamma(\mathcal{G}_k) + \gamma(B_k)| \cdot |\gamma(\mathcal{G}_k) - \gamma(B_k)| \cdot |\mathbb{E}[I_1(k) \,|\, \mathcal{F}_k]|.
$$

The first term is bounded by (6), since $\mathcal{G}_k$ and $\text{Tr}(K)I(B_k)/d$ are bounded independent of $d$; the second term is bounded $Cd^{-1}$ by Lemma C.9, and the last term is bounded by a constant by Lemma C.5.

Finally, consider $|\mathcal{E}_3|$:

$$
|\mathcal{E}_3| = \gamma(B_k)^2 \cdot |\mathbb{E}[(I_1(k) - I_2(k) \,|\, \mathcal{F}_k]|.
$$

By (6), the first term is bounded by $\hat{C}(\gamma)^2(1 + \|N_k\|_\infty^\alpha + \|Q_k\|_\infty^\alpha + \|\text{Tr}(K)I(B_k)/d\|_\infty^\alpha)^2$. All of these terms are bounded by a constant independent of $d$ (because of the stopping time.) The second term satisfies the assumptions of Lemma C.8 with $H = \nabla^2\varphi(X_k)$, and is thus bounded by $Cd^{-1}$. All together,

$$
2d\, \mathbb{E}[\mathcal{E}_k^{\text{Hess}}(\varphi) \,|\, \mathcal{F}_k] \leq d^{-\delta/4 + \varepsilon}.
$$

Summing up to $k = Td$ and dividing through by $2d$, we obtain the desired bound. $\qquad\square$

**Proposition C.4** (Gradient error term). *Let $f$ and $S$ be defined as in Assumption 2 and (37). Then, for any $z \in \Omega$, $\zeta > 0$ and $T > 0$, with overwhelming probability,*

$$
\sup_{0 \leq t \leq T \wedge \vartheta} \sum_{k=0}^{\lfloor td \rfloor - 1} \left\|\mathbb{E}\left[\mathcal{E}_k^{grad}(S(\cdot, z)) \,|\, \mathcal{F}_k\right]\right\| \leq d^{-\delta/4 + \zeta}.
$$

*Proof.* We have

$$d\mathbb{E}\left[\mathcal{E}_k^{\mathrm{grad}} \mid \mathcal{F}_k\right] = -\mathbb{E}\left[\gamma(G_k)\langle\nabla\varphi(X_k), \Delta_k\rangle \mid \mathcal{F}_k\right] + \gamma(B_k)\langle\nabla\varphi(X_k), \nabla R(X_k)\rangle$$
$$= -\mathbb{E}\left[(\gamma(G_k) - \gamma(\mathcal{G}_k))I_3(k) \mid \mathcal{F}_k\right] - (\gamma(\mathcal{G}_k) - \gamma(B_k))\mathbb{E}\left[I_3(k) \mid \mathcal{F}_k\right]$$
$$= \mathcal{E}_1 + \mathcal{E}_2.$$

We then have

$$|\mathcal{E}_1| \leq \mathbb{E}\left[|\gamma(G_k) - \gamma(\mathcal{G}_k)|^2 \mid \mathcal{F}_k\right]^{\frac{1}{2}} \cdot \mathbb{E}\left[|I_3(k)|^2 \mid \mathcal{F}_k\right]^{\frac{1}{2}}$$

$$\leq \mathbb{E}\left[|\gamma(G_k) + \gamma(\mathcal{G}_k)|^3 \mid \mathcal{F}_k\right]^{\frac{1}{4}} \cdot \mathbb{E}\left[|\gamma(G_k) - \gamma(\mathcal{G}_k)| \mid \mathcal{F}_k\right]^{\frac{1}{4}} \cdot \mathbb{E}\left[|I_3(k)|^2 \mid \mathcal{F}_k\right]^{\frac{1}{2}}.$$

Just as in the Hessian argument, (6) lets us bound $\mathbb{E}\left[|\gamma(G_k) + \gamma(\mathcal{G}_k)|^3 \mid \mathcal{F}_k\right]^{\frac{1}{4}}$ by $d^\varepsilon$ w.o.p., (5) lets us bound $\mathbb{E}\left[|\gamma(G_k) - \gamma(\mathcal{G}_k)| \mid \mathcal{F}_k\right]^{\frac{1}{4}}$ by $d^{-\delta/4}$ w.o.p., and Lemma C.5 lets us bound $\mathbb{E}\left[|I_3(k)|^2 \mid \mathcal{F}_k\right]^{\frac{1}{2}}$ by a constant, giving an overall bound of $|\mathcal{E}_1| \leq d^{-\delta/4+\varepsilon}$.

By the same argument as in the Hessian case, $|\mathcal{E}_2|$ is bounded by $Cd^{-1}$; in conclusion,

$$d\mathbb{E}\left[\mathcal{E}_k^{\mathrm{grad}} \mid \mathcal{F}_k\right] \leq d^{\varepsilon-\delta/4}.$$

Summing and dividing through by $d$, we obtain the desired result with $\zeta = \varepsilon$. $\qquad\square$

**Proposition C.5** (Gradient martingale)**.** *Let $f$ and $S$ be defined as in Assumption 2 and (37). Then, for any $z \in \Omega$, $\zeta > 0$ and $T > 0$, with overwhelming probability,*

$$\sup_{0 \leq t \leq T \wedge \vartheta} \left\|\mathcal{M}_{\lfloor dt \rfloor}^{\mathrm{grad}}(S(\cdot, z))\right\| \leq d^{-1/2+\zeta}.$$

*Proof.* For notational convenience, set $\Delta\mathcal{M}_k = \Delta\mathcal{M}_{d(k/d \wedge \vartheta)}^{\mathrm{grad}}$, and $F_k = -\gamma(G_k)I_3(k)/d$, so that

$$\Delta\mathcal{M}_k = F_k - \mathbb{E}[F_k \mid \mathcal{F}_k].$$

Set $F_k^\beta = \mathrm{Proj}_\beta(F_k)$, that is, ensuring $F_k$ stays in $[-\beta, \beta]$. Then $F_k^\beta - \mathbb{E}[F_k^\beta \mid \mathcal{F}_k]$ is in $[-2\beta, 2\beta]$, and so for the martingale $\mathcal{M}_k^\beta$ with increments $\Delta\mathcal{M}_k^\beta = F_k^\beta - \mathbb{E}[F_k^\beta \mid \mathcal{F}_k]$, Azuma's inequality tells us that

$$\mathbb{P}\left(|\mathcal{M}_k^\beta| \geq t\right) \leq 2\exp\left(\frac{-t^2}{2\sum_{i=0}^k (2\beta)^2}\right) \leq 2\exp\left(\frac{-t^2}{2Td(2\beta)^2}\right).$$

Set $\beta = d^{-1+\zeta/2}$ and $t = d^{-1/2+\zeta}$; this becomes

$$\mathbb{P}\left(|\mathcal{M}_k^\beta| \geq d^{-1/2+\zeta}\right) \leq 2\exp\left(\frac{-d^\zeta}{8T}\right).$$

However, $\mathcal{M}_k^\beta$ is not quite the martingale we started with: there is still an error term,

$$|\mathcal{M}_k - \mathcal{M}_k^\beta| = \left|\sum_{i=0}^k (F_k - \mathbb{E}[F_k \mid \mathcal{F}_k]) - (F_k^\beta - \mathbb{E}[F_k^\beta \mid \mathcal{F}_k])\right|$$

$$\leq \sum_{i=0}^k \left|F_k - F_k^\beta\right| + \left|\mathbb{E}[F_k - F_k^\beta \mid \mathcal{F}_k]\right|.$$

We bound this term in overwhelming probability. We have

$$\mathbb{P}\left(F_k - F_k^\beta \neq 0\right) = \mathbb{P}\left(|F_k| > \beta\right)$$

$$= \mathbb{P}\left(|\gamma(G_k)I_3(k)/d| > d^{-1+\zeta/2}\right)$$

$$\leq \mathbb{P}\left(\gamma(G_k) \geq d^{\zeta/4}\right) + \mathbb{P}\left(|I_3(k)| \geq d^{\zeta/4}\right).$$

The second term is superpolynomially small by Lemma C.5; the first term is superpolynomially small by (6) and (C.6).

$$\left| \mathbb{E}[F_k - F_k^{\beta} \,|\, \mathcal{F}_k] \right| = \left| \mathbb{E}[(F_k - F_k^{\beta})\mathbf{1}_{\{|F_k|>\beta\}} \,|\, \mathcal{F}_k] \right|$$

$$\leq \mathbb{E}[(F_k - F_k^{\beta})^2 \,|\, \mathcal{F}_k]^{\frac{1}{2}} \cdot \mathbb{E}[\mathbf{1}_{\{|F_k|>\beta\}}^2 \,|\, \mathcal{F}_k]^{\frac{1}{2}}$$

$$\leq 4\, \mathbb{E}[F_k^2 \,|\, \mathcal{F}_k]^{\frac{1}{2}} \cdot \mathbb{E}[\mathbf{1}_{\{|F_k|>\beta\}} \,|\, \mathcal{F}_k]^{\frac{1}{2}}$$

$$\leq 4d^{-1}\, \mathbb{E}[\gamma(G_k)^4 \,|\, \mathcal{F}_k]^{\frac{1}{4}} \cdot \mathbb{E}[I_3(k)^4 \,|\, \mathcal{F}_k]^{\frac{1}{4}} \cdot \mathbb{E}[\mathbf{1}_{\{|F_k|>\beta\}} \,|\, \mathcal{F}_k]^{\frac{1}{2}}.$$

As before, the first and second expectations are bounded by constants, and the last expectation is just the probability that $|F_k| > \beta$, which we have already shown is superpolynomially small. So with overwhelming probability, we have

$$|\mathcal{M}_k - \mathcal{M}_k^{\beta}| = \left| \sum_{i=0}^{k} (F_k - \mathbb{E}[F_k \,|\, \mathcal{F}_k]) - (F_k^{\beta} - \mathbb{E}[F_k^{\beta} \,|\, \mathcal{F}_k]) \right| \leq d^{-1/2+\zeta}$$

(any power of $d$ would have worked). Combining the error term and the projected martingale, we find that, with overwhelming probability,

$$|\mathcal{M}_k| \leq d^{-1/2+\zeta}.$$

We can now take the maximum over $k$ from 0 to $Td$ using a union bound; this does not affect the overwhelming probability statement. $\qquad \square$

**Proposition C.6** (Hessian martingale). *Let $f$ and $S$ be defined as in Assumption 2 and (37). Then, for any $z \in \Omega$, $\zeta > 0$ and $T > 0$, with overwhelming probability,*

$$\sup_{0 \leq t \leq T \wedge \vartheta} \left\| \mathcal{M}_{\lfloor td \rfloor}^{\mathrm{Hess}}(S(\cdot, z)) \right\| \leq d^{-1/2+\zeta}.$$

*Proof.* The proof here is basically identical to the previous one. Again, set $F_k = \gamma(G_k)^2 I_1(k)/d$ and $F_k^{\beta} = \mathrm{Proj}_{\beta}(F_k)$, with their associated martingales being $\mathcal{M}_k = F_k - \mathbb{E}[F_k \,|\, \mathcal{F}_k]$ and $\mathcal{M}_k^{\beta} = F_k^{\beta} - \mathbb{E}[F_k^{\beta} \,|\, \mathcal{F}_k]$. As before, Azuma's inequality, with $\beta = d^{-1+\zeta/2}$, gives us

$$\mathbb{P}(\mathcal{M}_k^{\beta} \geq d^{-1/2+\zeta}) \leq 2\exp\left(-\frac{d^{\zeta}}{8T}\right).$$

The error term is also quite similar:

$$|\mathcal{M}_k - \mathcal{M}_k^{\beta}| \leq \sum_{i=0}^{k} |F_k - F_k^{\beta}| + |\mathbb{E}[F_k - F_k^{\beta} \,|\, \mathcal{F}_k]|.$$

We have

$$\mathbb{P}(F_k - F_k^{\beta} \neq 0) \leq \mathbb{P}(\gamma(G_k)^2 \leq d^{\zeta/4}) + \mathbb{P}(|I_2(k)| \leq d^{\zeta/4}),$$

both of which are superpolynomially small by (6) and Lemma C.5. For the expectation, we have

$$|\mathbb{E}[F_k - F_k^{\beta} \,|\, \mathcal{F}_k]| \leq 4d^{-1}\, \mathbb{E}[\gamma(G_k)^8 \,|\, \mathcal{F}_k]^{\frac{1}{4}} \cdot \mathbb{E}[I_1(k)^4 \,|\, \mathcal{F}_k]^{\frac{1}{4}} \cdot \mathbb{E}[\mathbf{1}_{\{|F_k|>\beta\}} \,|\, \mathcal{F}_k]^{\frac{1}{2}};$$

this product is superpolynomially small by (6), Lemma C.6, and Lemma C.5. Overall, we have, with overwhelming probability,

$$|\mathcal{M}_k| \leq d^{-1/2+\zeta}.$$

Taking the supremum, we obtain the desired result. $\qquad \square$

**Proposition C.7** (Integral error term). *Let $f$ and $S$ be defined as in Assumption 2 and (37). Then, for $z \in \Omega$,*

$$|\xi_{td}(S(\cdot, z))| \leq d^{-1/2}.$$

*Proof.* We have, as above,

$$|\xi_{td}| = \left| \int_{(\lfloor td \rfloor - 1)/d}^{t} d \cdot \Delta\varphi(X_{sd}) \,\mathrm{d}s \right|$$

$$\leq \max_{0 \leq j \leq \lceil td \rceil} \{|\Delta\varphi(X_j)|\},$$

which is bounded by $d^{-1/2}$ w.o.p. by the boundedness of $I_1$, $I_2$, $I_3$, and $\gamma(B_k)$. $\qquad \square$

### C.2.2 General bounds

In this section, we make use of the subgaussian norm $\| \cdot \|_{\psi_2}$ of a random variable (see [56] for details.) When it exists, this norm is defined as

$$\|X\|_{\psi_2} \asymp \inf \left\{ V > 0 : \forall t > 0, \ \mathbb{P}(|X| > t) \leq 2e^{-t^2/V^2} \right\}. \tag{38}$$

In particular, Gaussian random variables have a well-defined subgaussian norm.

**Lemma C.3** ([15], Lemma 5.3). *There exist constants $c, C > 0$ such that*

$$c\|W\|^2 \leq \|S(W,z)\|_\Omega \leq C\|W\|^2, \quad \|\nabla_X S(W,z)\|_\Omega \leq C\|W\|, \quad \text{and} \quad \|\nabla_X^2 S(W,z)\|_\Omega \leq C.$$

**Lemma C.4** (Preliminary bounds). *With $f$ and $\Delta_k$ defined as above, for $\varepsilon > 0$ and $\lambda \geq 0$, we have*

$$f'(r_k) \leq d^\varepsilon \quad \text{w.o.p. and} \quad \mathbb{E}[|f'(r_k)|^\lambda \,|\, \mathcal{F}_k] \leq C(\lambda), \tag{39}$$

$$\frac{\|\Delta_k\|^2}{d} \leq d^\varepsilon \quad \text{w.o.p. and} \quad \mathbb{E}\left[ \left( \frac{\|\Delta_k\|^2}{d} \right)^\lambda \,|\, \mathcal{F}_k \right] \leq C(\lambda). \tag{40}$$

*Proof of* (39) *in Lemma C.4.* By [15, Lemma 3.4], if function $f$ is $\alpha$-pseudo-Lipschitz with Lipschitz constant $L(f)$ (as in (2)) and the noise $\epsilon$ is independent of $a$, then

$$|f'(r)| \leq C(\alpha)(L(f))(1 + |r| + |\epsilon|)^{\max\{1,\alpha\}}.$$

Then

$$
\begin{aligned}
|f'(r_k)| &\leq C(\alpha)(L(f))(1 + |r_k| + |\epsilon|)^{\max\{1,\alpha\}} \\
&\leq C(\alpha)(L(f))(1 + |X_k^\top a_{k+1}| + |\epsilon|)^{\max\{1,\alpha\}}.
\end{aligned} \tag{41}
$$

Now, since $a_{k+1}$ is Gaussian, we can write $a_{k+1} = \sqrt{K} v_k$, for a standard normal $v_k$. Then we see that $X_k^\top a_{k+1} = X_k^\top \sqrt{K} v_k$ is a single-variable Gaussian, with variance $|X_k^\top K X_k| \leq \|X_k\|^2 \cdot \|K\|_{\text{op}}$ (bounded independently of $d$ because of the stopping time on $X_k$). Similarly, $\epsilon$ is Gaussian and independent of $a_{k+1}$, so the expression (41) is bounded w.o.p. by $d^\varepsilon$, and

$$\mathbb{E}\left[ \left( C(\alpha)(L(f))(1 + |X_k^\top a_{k+1}| + |\epsilon|)^{\max\{1,\alpha\}} \right)^\lambda \,|\, \mathcal{F}_k \right] \leq C(\lambda)$$

for some constant $C(\lambda)$. □

*Proof of* (40) *in Lemma C.4.* We can write $a_{k+1} = \sqrt{K} v_k$, where $v_k$ is a standard $d$-dimensional normal vector. Then, by Hanson-Wright, we have

$$
\begin{aligned}
\mathbb{P}\left( |\|a_{k+1}\|^2 - \mathbb{E}[\|a_{k+1}\|^2 \,|\, \mathcal{F}_k]| \geq d \right) &= \mathbb{P}\left( |v_k^\top K v_k - \mathbb{E}[v_k^\top K v_k \,|\, \mathcal{F}_k]| \geq d \right) \\
&\leq 2 \exp\left( -\frac{cd^2}{\|K\|_F^2 + \|K\|_{\text{op}} d} \right) \\
&\leq 2 \exp\left( -\frac{cd^2}{d(\|K\|_{\text{op}} + \|K\|_{\text{op}}^2)} \right) \\
&\leq 2 \exp\left( -Cd \right).
\end{aligned}
$$

Now, note that $\mathbb{E}[v_k^\top K v_k \,|\, \mathcal{F}_k] = \text{Tr}(K) \leq d\|K\|_{\text{op}}$. Together, we get that $\|a_{k+1}\|^2 \leq d^{1+\epsilon}$ with overwhelming probability. Then

$$\frac{\|\Delta_k\|^2}{d} = \frac{\|f'(r_k) a_{k+1}\|^2}{d} = \frac{\|a_{k+1}\|^2 f'(r_k)^2}{d},$$

which is bounded by $d^{2\varepsilon}$ w.o.p. Now for the expectation:

$$
\begin{aligned}
\mathbb{E}\left[ \left( \frac{\|\Delta_k\|^2}{d} \right)^\lambda \,|\, \mathcal{F}_k \right] &\leq \mathbb{E}\left[ \left( \frac{\|\sqrt{K} v_k\|^2}{d} \right)^{2\lambda} \,|\, \mathcal{F}_k \right]^{\frac{1}{2}} \cdot \mathbb{E}\left[ f'(r_k)^{4\lambda} \,|\, \mathcal{F}_k \right]^{\frac{1}{2}} \\
&\leq \mathbb{E}\left[ \left( \frac{\|K\|_{\text{op}} \cdot \|v_k\|^2}{d} \right)^{2\lambda} \,|\, \mathcal{F}_k \right]^{\frac{1}{2}} \cdot \mathbb{E}\left[ f'(r_k)^{4\lambda} \,|\, \mathcal{F}_k \right]^{\frac{1}{2}} \tag{42}
\end{aligned}
$$

For the first term, we have

$$\mathbb{E}\left[\left(\frac{\|K\|_{\mathrm{op}} \cdot \|v_k\|^2}{d}\right)^{2\lambda} \mid \mathcal{F}_k\right] = \|K\|_{\mathrm{op}}^{2\lambda} \cdot \mathbb{E}\left[\left(\frac{\|v_k\|^2}{d}\right)^{2\lambda} \mid \mathcal{F}_k\right]$$

$$\leq \|K\|_{\mathrm{op}}^{2\lambda} \cdot \frac{1}{d}\sum_{i=0}^{d-1}\mathbb{E}\left[\left(\|v_k^i\|^2\right)^{2\lambda} \mid \mathcal{F}_k\right] \qquad \text{(Jensen's inequality)}$$

$$= \|K\|_{\mathrm{op}}^{2\lambda} \cdot \mathbb{E}\left[\|v_k^0\|^{4\lambda} \mid \mathcal{F}_k\right], \qquad \text{(i.i.d. assumption)}$$

where we are using the notation $v_k^i$ to refer to the $i$th component of the vector $v_k$. Now, since $v_k^0$ is just a standard Gaussian, all of its moments are bounded. The second term in (42) is bounded by a constant by (39), as desired. $\qquad\square$

**Lemma C.5** (Gradient and Hessian bounds). *Setting*

$$I_1(k) \stackrel{def}{=} \Delta_k^\top \nabla^2\varphi(X_k)\Delta_k/d, \quad I_2(k) \stackrel{def}{=} \mathrm{Tr}(\nabla^2\varphi(X_k)K)\mathbb{E}\left[f'(r_k)^2 \mid \mathcal{F}_k\right]/d,$$

$$I_3(k) \stackrel{def}{=} \nabla\varphi(X_k)^\top\Delta_k,$$

*for any $\varepsilon > 0$ and $\lambda \geq 0$, we have*

$$|I_1(k)| \leq d^\varepsilon \quad \text{w.o.p. and} \quad \mathbb{E}\left[|I_1(k)|^\lambda \mid \mathcal{F}_k\right] \leq C(\lambda), \tag{43}$$
$$|I_2(k)| \leq C, \tag{44}$$
$$|I_3(k)| \leq d^\varepsilon \quad \text{w.o.p. and} \quad \mathbb{E}\left[|I_3(k)|^\lambda \mid \mathcal{F}_k\right] \leq C(\lambda). \tag{45}$$

*Proof of* (43) *in Lemma C.5.* Using the fact that $\|\nabla^2\varphi(X_k)\|_{\mathrm{op}} \leq \|S(W_k, \cdot)\|_\Omega$,

$$\frac{|\Delta_k^\top \nabla^2\varphi(X_k)\Delta_k|}{d} \leq \frac{\|S(W_k, \cdot)\|_\Omega \|\Delta_k\|^2}{d}$$

$$\leq \frac{C\|W_k\|^2 \|\Delta_k\|^2}{d}. \qquad \text{(Lemma C.3)}$$

Now, $\|W_k\|$ is bounded by the stopping time. From Lemma C.4, $\frac{\|\Delta_k\|^2}{d}$ is bounded by $d^\varepsilon$ w.o.p., and every moment of this expression is bounded independent of $d$, as desired. $\qquad\square$

*Proof of* (44) *in Lemma C.5.* We have

$$\frac{\left|\mathrm{Tr}(\nabla^2\varphi(X_k)K)\mathbb{E}\left[f'(r_k)^2 \mid \mathcal{F}_k\right]\right|}{d} \leq \frac{d\|\nabla^2\varphi(X_k)K\|_{\mathrm{op}} \cdot \mathbb{E}[f'(r_k)^2 \mid \mathcal{F}_k]}{d}$$

$$\leq \|\nabla^2\varphi(X_k)\|_{\mathrm{op}} \cdot \|K\|_{\mathrm{op}} \cdot \mathbb{E}[f'(r_k)^2 \mid \mathcal{F}_k]$$

$$\leq CM^2\,\mathbb{E}[f'(r_k)^2 \mid \mathcal{F}_k]. \qquad \text{(Lemma C.3)}$$

From Lemma C.4, $\mathbb{E}[f'(r_k)^2 \mid \mathcal{F}_k]$ is bounded by a constant independent of $d$, as desired. $\qquad\square$

*Proof of* (45) *in Lemma C.5.* We have

$$|\nabla\varphi(X_k)^\top\Delta_k| \leq |\nabla\varphi(X_k)^\top a_{k+1}| \cdot |f'(r_k)|.$$

By Lemma C.3, $\|\nabla\varphi(X_k)\| \leq C\|W_k\| \leq CM$ (since we are working under a stopping time), and so $\nabla\varphi(X_k)^\top a_{k+1}$ is subgaussian (and thus bounded by $d^\varepsilon$ w.o.p.). By (39), $f'(r_k)$ is bounded by $d^\varepsilon$ w.o.p., and so their product is bounded by $d^{2\varepsilon}$ w.o.p., as desired. Now for the expectation:

$$\mathbb{E}\left[|\nabla\varphi(X_k)^\top\Delta_k| \mid \mathcal{F}_k\right] \leq \mathbb{E}\left[|\nabla\varphi(X_k)^\top a_{k+1}| \cdot |f'(r_k)| \mid \mathcal{F}_k\right]$$

$$\leq \mathbb{E}\left[|\nabla\varphi(X_k)^\top a_{k+1}|^2 \mid \mathcal{F}_k\right]^{\frac{1}{2}} \cdot \mathbb{E}\left[f'(r_k)^2 \mid \mathcal{F}_k\right]^{\frac{1}{2}}$$

The first term is bounded by a constant independent of $d$, since subgaussian moments are bounded. The second term is bounded by Lemma C.4, completing the proof. $\qquad\square$

**Lemma C.6** (Infinity norm bounds). *For $G_k$, $N_k$, $Q_k$ as defined in 1.2, we have, for any $\varepsilon, \lambda > 0$, there exists $C > 0$ such that,*

$$\|G_k\|_\infty \leq d^\varepsilon \quad \text{w.o.p. and} \quad \mathbb{E}[\|G_k\|_\infty^\lambda \mid \mathcal{F}_k] \leq d^\varepsilon \quad \text{w.o.p.,} \tag{46}$$

$$\|N_k\|_\infty \leq C, \quad \|Q_k\|_\infty \leq C, \quad \|\mathcal{G}_k\|_\infty \leq C. \tag{47}$$

*Proof.* The first line, (46), follows from (40). For the first inequality, $\|G_k\|_\infty = \max_{0 \leq j \leq k} \frac{\|\Delta_j\|^2}{d}$, which are all bounded by $d^\varepsilon$ with overwhelming probability. A union bound tells us that the maximum is also bounded by $d^\varepsilon$ w.o.p.. For the second inequality,

$$\mathbb{E}\left[|G_k|_\infty^\lambda \mid \mathcal{F}_k\right] \leq \mathbb{E}\left[\left(\frac{\|\Delta_k\|^2}{d}\right)^\lambda \mid \mathcal{F}_k\right] + \mathbb{E}\left[\max_{0 \leq j \leq k-1}\left(\frac{\|\Delta_j\|^2}{d}\right)^\lambda \mid \mathcal{F}_k\right]$$

$$\leq \mathbb{E}\left[\left(\frac{\|\Delta_k\|^2}{d}\right)^\lambda \mid \mathcal{F}_k\right] + \max_{0 \leq j \leq k-1}\left(\frac{\|\Delta_j\|^2}{d}\right)^\lambda$$

$$\leq d^\varepsilon, \tag{w.o.p.}$$

as desired. The second line is more straightforward:

$$\|N_k\|_\infty = \max_{0 \leq j \leq k} \|(W_j^+)^\top W_j^+\|.$$

Now, $\|X^\star\|$ and $\|X_0\|$ are bounded independent of $d$, and $\|X_j\|$ is bounded by $cM$ (because of the stopping time we are using.) Thus the maximum over $j$ of their inner products are bounded by a constant. The same thing holds for $\|Q_k\|_\infty$:

$$\|Q_k\|_\infty = \max_{0 \leq j \leq k} \mathcal{R}(X_j)$$

$$= \max_{0 \leq j \leq k} h(W_j^\top K W_j).$$

Since the derivative of $h$ is pseudo-Lipschitz, $h$ is continuous, and thus bounded for bounded arguments. And indeed, the argument to $h$ is bounded:

$$\|W_j^\top K W_j\| \leq \|W_j\|^2 \|K\|_{\text{op}},$$

both of which are bounded independent of $d$. Finally, a similar argument applies to $\mathcal{G}_k$:

$$\|\mathcal{G}_k\|_\infty = \max_{0 \leq j \leq k} \mathbb{E}\left[\frac{\|\Delta_j\|^2}{d} \mid \mathcal{F}_j\right] \leq \max_{0 \leq j \leq k} C = C$$

by Lemma C.4. $\qquad\square$

We now prove a concentration result that closely follows [15, Proposition 5.6].

**Lemma C.7** ([15], Lemma 5.2). *Suppose $v \in \mathbb{R}^d$ is distributed $\mathcal{N}(0, I_d)$ and $U \in R^{d \times 2}$ has orthonormal columns. Then*

$$v \mid U^\top v \sim v - U(U^\top v) + UU^\top v, \tag{48}$$

*where $v - U\left(U^T v\right) \sim N\left(0, I_d - UU^T\right)$ and $UU^T v \sim N\left(0, UU^T\right)$ with $v - U\left(U^T v\right)$ independent of $UU^T v$.*

**Lemma C.8.** *For a matrix $H = H_k$ with bounded operator norm, or $\|H\|_{\text{op}} < C$ and $\mathbb{E}[H_k \mid \mathcal{F}_k] = H_k$, set $q(a) = a^\top H a$. Then*

$$\left|\mathbb{E}[q(a_{k+1})f'(r_k)^2 \mid \mathcal{F}_k] - \text{Tr}(KH)\, \mathbb{E}[f'(r_k)^2 \mid \mathcal{F}_k]\right| \leq C(H).$$

*Note that the $H$ used here is not the same as the matrix used in the integro-differential equation.*

*Proof.* Many of the computations in this proof are taken directly from [15], but we repeat them here for completeness. We have $\mathcal{F}_k = \sigma(\{W_i\}_{i=0}^k)$; set $\hat{\mathcal{F}}_k = \sigma(\{W_i\}_{i=0}^k, \{r_i\}_{i=0}^k)$. A simple calculation shows that

$$\mathbb{E}[q(a_{k+1})f'(r_k)^2 \mid \hat{\mathcal{F}}_k] = \mathbb{E}[q(a_{k+1} - \mathbb{E}[a_{k+1} \mid \hat{\mathcal{F}}_k]) \mid \hat{\mathcal{F}}_k]\, \mathbb{E}_\epsilon[f'(r_k)^2]$$

$$+ q(\mathbb{E}[a_{k+1} \mid \hat{\mathcal{F}}_k])\, \mathbb{E}_\epsilon[f'(r_k)^2]. \tag{49}$$

To compute the conditional mean $\mathbb{E}[a_{k+1} \,|\, \hat{\mathcal{F}}_k]$ and covariance $(a_{k+1} - \mathbb{E}[a_{k+1} \,|\, \hat{\mathcal{F}}_k])(a_{k+1} - \mathbb{E}[a_{k+1} \,|\, \hat{\mathcal{F}}_k])^\top$, we use Lemma C.7. By Assumption 1, we can write $a_{k+1} = \sqrt{K} v_k$, for $v_k \sim \mathcal{N}(0, I_d)$.

Now we perform a QR-decomposition on $\sqrt{K} W_k \overset{\text{def}}{=} Q_k R_k$ where $Q_k \in \mathbb{R}^{d \times 2}$ with orthonormal columns and $R_k \in \mathbb{R}^{2 \times 2}$ is upper triangular (and invertible). Set $\Pi_k \overset{\text{def}}{=} Q_k Q_k^T$. In distribution,

$$a_{k+1} \,|\, a_{k+1}^\top W_k \overset{\text{d}}{=} \sqrt{K} v_k \,|\, R_k^T Q_k^T v_k.$$

As $R_k$ is invertible, by Lemma C.7,

$$a_{k+1} \,|\, a_{k+1}^\top W_k \overset{\text{d}}{=} \sqrt{K} v_k \,|\, Q_k^T v_k \overset{\text{d}}{=} \sqrt{K}\big(v_k - \Pi_k v_k\big) + \sqrt{K} \Pi_k v_k. \tag{50}$$

We note that $(I_d - \Pi_k) v_k \sim N(0, I_d - \Pi_k)$ and $\Pi_k v_k \sim N(0, \Pi_k)$ with $(I_d - \Pi_k) v_k$ independent of $\Pi_k v_k$. From this, we have that

$$\mathbb{E}\left[a_{k+1} \,|\, \hat{\mathcal{F}}_k\right] = \sqrt{K} \Pi_k v_k, \quad \text{where } v_k \sim N(0, I_d). \tag{51}$$

Moreover the conditional covariance of $a_{k+1}$ is precisely

$$\left(\mathbb{E}\left[(a_{k+1} - \mathbb{E}\left[a_{k+1} \,|\, \hat{\mathcal{F}}_k\right])(a_{k+1} - \mathbb{E}\left[a_{k+1} \,|\, \hat{\mathcal{F}}_k\right])^\top \,|\, \hat{\mathcal{F}}_k\right]\right) \tag{52}$$
$$= \sqrt{K}(I_d - \Pi_k)\sqrt{K}, \quad \text{where } \Pi_k = Q_k Q_k^T.$$

Next, using that $\mathbb{E}[H_k \,|\, \mathcal{F}_k] = H_k$, we expand (49) to get the leading order behavior

$$\mathbb{E}\left[q(a_{k+1}) f'(r_k)^2 \,|\, \hat{\mathcal{F}}_k\right] = \operatorname{Tr}(HK) \, \mathbb{E}_\epsilon[f'(r_k)^2]$$
$$- \operatorname{Tr}(H\sqrt{K}\Pi_k\sqrt{K}) \, \mathbb{E}_\epsilon[f'(r_k)^2] \tag{53}$$
$$+ q(\sqrt{K}\Pi_k v_k) \, \mathbb{E}_\epsilon[f'(r_k)^2].$$

Taking the expectation with respect to $\mathcal{F}_k$, we obtain

$$\mathbb{E}\left[q(a_{k+1}) f'(r_k)^2 \,|\, \mathcal{F}_k\right] - \operatorname{Tr}(HK) \, \mathbb{E}[f'(r_k)^2 \,|\, \mathcal{F}_k] = \mathbb{E}[\mathcal{E}_k \,|\, \mathcal{F}_k], \tag{54}$$

where the error $\mathcal{E}_k$ is defined as

$$\mathcal{E}_k = -\operatorname{Tr}(H\sqrt{K}\Pi_k\sqrt{K}) \, \mathbb{E}_\epsilon[f'(r_k)^2] \tag{55}$$
$$+ q(\sqrt{K}\Pi_k v_k) \, \mathbb{E}_\epsilon[f'(r_k)^2]. \tag{56}$$

The proof now turns to bounding the expectation of this error quantity.

$$|\operatorname{Tr}(H\sqrt{K}\Pi_k\sqrt{K}) \, \mathbb{E}[f'(r_k)^2 \,|\, \mathcal{F}_k]| = |\operatorname{Tr}(H\sqrt{K}\Pi_k\sqrt{K})| \cdot \mathbb{E}[f'(r_k)^2 \,|\, \mathcal{F}_k]$$
$$\leq \|H\|_{\text{op}}\|K\|_{\text{op}}|\operatorname{Tr}(\Pi_k)| \cdot \mathbb{E}[f'(r_k)^2 \,|\, \mathcal{F}_k]$$
$$\leq \|H\|_{\text{op}}\|K\|_{\text{op}} \cdot \operatorname{rank}(Q_k) \, \mathbb{E}[f'(r_k)^2 \,|\, \mathcal{F}_k]$$
$$\leq 2\|H\|_{\text{op}}\|K\|_{\text{op}} \, \mathbb{E}[f'(r_k)^2 \,|\, \mathcal{F}_k].$$

By (39), the expectation is bounded by a constant, so this term is overall bounded by a constant. We move on to the next term in the error:

$$q(\sqrt{K}\Pi_k v_k) f'(r_k)^2 \leq \|H\|_{\text{op}}\|K\|_{\text{op}}\|\Pi_k v_k\|^2 f'(r_k)^2.$$

Taking expectations and using Cauchy Schwarz, we obtain

$$\mathbb{E}[q(\sqrt{K}\Pi_k v_k) f'(r_k)^2 \,|\, \mathcal{F}_k] \leq \|H\|_{\text{op}}\|K\|_{\text{op}} \cdot \sqrt{\mathbb{E}[\|\Pi_k v_k\|^4 \,|\, \mathcal{F}_k]} \cdot \sqrt{\mathbb{E}[f'(r_k)^4 \,|\, \mathcal{F}_k]}.$$

The first expectation is $\mathbb{E}[\|\Pi_k v_k\|^2 \,|\, \mathcal{F}_k] = \|\Pi_k\|_{\text{F}}^4 = 8$, and the second is bounded by (39) as before. We thus conclude that $\mathbb{E}[\mathcal{E}_k \,|\, \mathcal{F}_k]$ is bounded by a constant depending on $\|H\|_{\text{op}}$, completing the proof. $\qquad\square$

**Lemma C.9.** *There is a constant $C$ such that*

$$|\gamma(\mathcal{G}_k) - \gamma(B_k)| \leq C d^{-1}.$$

*Proof.* Using the Lipschitz condition on the stepsize, we have

$$
\begin{aligned}
|\gamma(\mathcal{G}_k) - \gamma(B_k)| & \\
&\leq \|\mathcal{G}_k - \mathrm{Tr}(K)I(B_k)/d\|_\infty \times (1 + 2\|N_k\|_\infty^\alpha + \|\mathcal{G}_k\|_\infty^\alpha + \|\mathrm{Tr}(K)I(B_k)/d\|_\infty^\alpha + 2\|Q_k\|_\infty^\alpha) \\
&\leq C\|\mathcal{G}_k - \mathrm{Tr}(K)I(B_k)/d\|_\infty \qquad\qquad\qquad\qquad\qquad\qquad\qquad\qquad\qquad \text{(Lemma C.6)} \\
&\leq Cd^{-1} \max_{0 \leq j \leq k} \left\|\mathbb{E}[a_{j+1}^\top a_{j+1} f'(r_j)^2 \mid \mathcal{F}_j] - \mathrm{Tr}(K)\,\mathbb{E}[f'(r_j)^2 \mid \mathcal{F}_j]\right\| \\
&\leq Cd^{-1}, \qquad\qquad\qquad\qquad\qquad\qquad\qquad\qquad\qquad\qquad\qquad\qquad\qquad \text{(Lemma C.8)}
\end{aligned}
$$

as desired. □

### C.3 Specific learning rates

In this section, we confirm that AdaGrad-Norm satisfies Assumption 6. In the notation of Assumption 6, we have, for AdaGrad-Norm,

$$
\gamma(td, f, g, q) = \frac{\eta}{\sqrt{b^2 + \int_0^\infty g(s)\,\mathrm{d}s}}.
$$

Note that this reduces to the discrete stepsize if we plug in $g = G_k$:

$$
\begin{aligned}
\gamma(td, f, G_k(d \times \cdot), q) &= \frac{\eta}{\sqrt{b^2 + \int_0^\infty G_k(ds)\,\mathrm{d}s}} \\
&= \frac{\eta}{\sqrt{b^2 + \int_0^\infty \left(1_{\{ds \leq k\}} \frac{1}{d} \sum_{i=0}^k \|\nabla_X \Psi(X_i; a_{i+1}, \epsilon_{i+1})\|^2 1_{[i,i+1)}(ds)\right)\,\mathrm{d}s}} \\
&= \frac{\eta}{\sqrt{b^2 + \int_0^\infty \left(1_{\{u \leq k\}} \frac{1}{d^2} \sum_{i=0}^k \|\nabla_X \Psi(X_i; a_{i+1}, \epsilon_{i+1})\|^2 1_{[i,i+1)}(u)\right)\,\mathrm{d}u}} \\
&= \frac{\eta}{\sqrt{b^2 + \frac{1}{d^2} \sum_{i=0}^k \|\nabla_X \Psi(X_i; a_{i+1}, \epsilon_{i+1})\|^2}},
\end{aligned}
$$

which is exactly the discrete version of the AdaGrad-Norm stepsize.

**Proposition C.8** (Lipschitz). *For functions $f, g, q$ such that $f(ds) = g(ds) = q(ds) = 0$ for $s > t$, the AdaGrad stepsize $\gamma$ is Lipschitz. That is,*

$$
|\gamma(td, f(d \times \cdot), g(d \times \cdot), q(d \times \cdot)) - \gamma(td, \hat{f}(d \times \cdot), \hat{g}(d \times \cdot), \hat{q}(d \times \cdot))| \leq C(t, \gamma)(\|g - \hat{g}\|_\infty).
$$

**Remark C.2.** *This is a stronger condition than the $\alpha$-pseudo Lipschitz one in Assumption 6.*

*Proof.* To show this, we look at the derivative of the AdaGrad stepsize function. Setting $F(x) = \frac{\eta}{\sqrt{b^2 + x}}$, we have

$$
|F'(x)| = \frac{\eta}{2(b^2 + x)^{3/2}} \leq \frac{\eta}{2b^3}
$$

for $x \in [0, \infty)$. We thus have

$$|\gamma(td, f(d \times \cdot), g(d \times \cdot), q(d \times \cdot)) - \gamma(td, \hat{f}(d \times \cdot), \hat{g}(d \times \cdot), \hat{q}(d \times \cdot))|$$

$$= \left| \frac{\eta}{\sqrt{b^2 + \int_0^\infty g(ds) \, ds}} - \frac{\eta}{\sqrt{b^2 + \int_0^\infty \hat{g}(ds) \, ds}} \right|$$

$$= \left| F\left( \int_0^\infty g(ds) \, ds \right) - F\left( \int_0^\infty \hat{g}(ds) \, ds \right) \right|$$

$$\leq \frac{\eta}{2b^3} \left| \int_0^\infty g(ds) \, ds - \int_0^\infty \hat{g}(ds) \, ds \right|$$

$$\leq \frac{\eta}{2b^3} \left| \int_0^t g(ds) \, ds - \int_0^t \hat{g}(ds) \, ds \right|$$

$$\leq \frac{\eta}{2b^3} \left( t \cdot \|g - \hat{g}\|_\infty \right)$$

$$\leq \frac{\eta t}{2b^3} \cdot \|g - \hat{g}\|_\infty,$$

where we were able to replace the $\infty$ with a $t$ because $g(ds) = 0$ for $s > t$. We have thus obtained a Lipschitz constant $\frac{\eta t}{2b^3}$ depending only on $t$. $\qquad\square$

Next we show that the AdaGrad-Norm is bounded.

**Proposition C.9** (Boundedness). *Suppose $\gamma$ is AdaGrad-Norm. Then* (6), *as part of Assumption 6, holds.*

*Proof.* This is immediate:

$$\gamma(td, f, g, q) = \frac{\eta}{\sqrt{b^2 + \int_0^t g(s) \, ds}} \leq \frac{\eta}{b}.$$

$\qquad\square$

It remains to show that AdaGrad-Norm satisfies (5) in Assumption 6.

**Proposition C.10** (Concentration). *Suppose $\gamma$ is AdaGrad-Norm, with $G_k$ and $\mathcal{G}_k$ being defined as before. Then Equation* (5), *as part of Assumption 6, holds:*

$$\mathbb{E}[|\gamma(G_k) - \gamma(\mathcal{G}_k)| \,|\, \mathcal{F}_k] \leq C d^{-\delta} (1 + \|f\|_\infty^\alpha + \|q\|_\infty^\alpha).$$

*Proof.* Looking to remove the square roots, we have

$$|\gamma(G_k) - \gamma(\mathcal{G}_k)| \leq |\gamma(G_k)^2 - \gamma(\mathcal{G}_k)^2|^{\frac{1}{2}}.$$

For AdaGrad-Norm, we have

$$\left| \gamma(G_k)^2 - \gamma(\mathcal{G}_k)^2 \right| = \eta^2 \left| \frac{1}{b^2 + \frac{1}{d^2} \sum_{j=0}^k \|\Delta_j\|^2} - \frac{1}{b^2 + \frac{1}{d^2} \sum_{j=0}^k \mathbb{E}\left[ \|\Delta_j\|^2 \,|\, \mathcal{F}_j \right]} \right|$$

$$\leq \frac{\eta^2}{d^2 b^4} \cdot \left| \sum_{j=0}^k (\mathbb{E}\left[ \|\Delta_j\|^2 \,|\, \mathcal{F}_j \right] - \|\Delta_j\|^2) \right|. \tag{57}$$

We now bound the sum above. Set $F_i = \|\Delta_i\|^2/d$, $F_i^\beta = \text{Proj}_\beta(F_i)$, $\Delta \mathcal{M}_i = F_i - \mathbb{E}[F_i \,|\, \mathcal{F}_i]$, and $\Delta \mathcal{M}_i^\beta = F_i^\beta - \mathbb{E}[F_i^\beta \,|\, \mathcal{F}_i]$. Then $|\Delta \mathcal{M}_i^\beta| \in [-2\beta, 2\beta]$, so Azuma's inequality gives us

$$\mathbb{P}\left( |\mathcal{M}_k^\beta| \geq t \right) \leq 2 \exp\left( -\frac{-t^2}{2 \sum_{i=0}^k (2\beta)^2} \right),$$

$$\mathbb{P}\left( |\mathcal{M}_k^\beta| \geq d^{1/2+\varepsilon} \right) \leq 2 \exp\left( -\frac{-d^{1+2\varepsilon}}{2Td(2d^{\varepsilon/2})^2} \right) = \exp\left( -\frac{d^\varepsilon}{8T} \right).$$

where we set $\beta = d^{\varepsilon/2}$. This is close to the bound we want: the error is

$$|\mathcal{M}_k - \mathcal{M}_k^\beta| \leq \sum_{i=0}^{k} |F_i - F_i^\beta| + |\mathbb{E}[F_i - F_i^\beta \mid \mathcal{F}_i]|.$$

We have

$$\mathbb{P}(F_i - F_i^\beta \neq 0) = \mathbb{P}(|F_i| > \beta) = \mathbb{P}\left(\frac{\|\Delta_i\|^2}{d} > d^{\varepsilon/2}\right),$$

which superpolynomially small by (40). The expectation is similar:

$$\begin{aligned}
|\mathbb{E}[F_i - F_i^\beta \mid \mathcal{F}_i]| &= |\mathbb{E}[(F_i - F_i^\beta)\mathbf{1}_{\{|F_i|>\beta\}} \mid \mathcal{F}_i]| \\
&\leq \mathbb{E}[|F_i - F_i^\beta|^2 \mid \mathcal{F}_i]^{\frac{1}{2}} \cdot \mathbb{E}[\mathbf{1}_{\{|F_i|>\beta\}} \mid \mathcal{F}_i]^{\frac{1}{2}} \\
&\leq 4\,\mathbb{E}[|F_i|^2 \mid \mathcal{F}_i]^{\frac{1}{2}} \cdot \mathbb{E}[\mathbf{1}_{\{|F_i|>\beta\}} \mid \mathcal{F}_i]^{\frac{1}{2}}.
\end{aligned}$$

The first expectation is bounded by a constant independent of $d$ by (40), and the second expectation is superpolynomially small by the same argument as above. We then have

$$|\mathcal{M}_k - \mathcal{M}_k^\beta| \leq d^{1/2+\varepsilon}$$

with overwhelming probability (note that this would be true for any power of $d$, by the definition of superpolynomially small.) We thus conclude that

$$|\mathcal{M}_k| \leq d^{1/2+\varepsilon}$$

with overwhelming probability. Multiplying by $d$, we find that

$$\left|\sum_{j=0}^{k} (\mathbb{E}\left[\|\Delta_j\|^2 \mid \mathcal{F}_j\right] - \|\Delta_j\|^2)\right| \leq d^{3/2+\varepsilon} \quad \text{w.o.p.}$$

Plugging this back into (57), we find that

$$\begin{aligned}
\left|\gamma(G_k)^2 - \gamma(\mathcal{G}_k)^2\right| &\leq \frac{\eta^2}{d^2 b^4} d^{3/2+\varepsilon} \\
&\leq C d^{-1/2+\varepsilon}
\end{aligned}$$

with overwhelming probability, and so, taking the square root,

$$|\gamma(G_k) - \gamma(\mathcal{G}_k)| \leq C d^{-1/4+\varepsilon/2} \quad \text{w.o.p,}$$

which is less than $d^{-1/4+\varepsilon}$ as $d$ grows (we replaced the constant with an extra factor of $d^{\varepsilon/2}$.) Controlling the expectation via the boundedness of $\gamma$, we find that with $\delta = 1/8$,

$$\mathbb{E}[|\gamma(G_k) - \gamma(\mathcal{G}_k)| \mid \mathcal{F}_k] \leq d^{-\delta} \quad \text{w.o.p.,}$$

as desired. $\qquad\square$

## D  Proofs for AdaGrad-Norm analysis

In this section we provide proofs of the propositions related to AdaGrad-Norm in the least squares setting as well as the more general strongly convex setting. Statements of the propositions for least squares examples are found in Section 4.

### D.1  Strongly convex setting

In order to derive the limiting learning rate in this case, we need the following assumption and some standard definitions of strong convexity.

**Assumption 7** (Risk and loss minimizer). *Suppose that*

$$X^\star \in arg\,min_X\left\{\mathcal{R}(X) = \mathbb{E}_{a,\epsilon}[f(\langle X, a\rangle, \langle X^\star, a\rangle), \epsilon]\right\}$$

*exists and has norm bounded independent of $d$. Then one has,*

$$\langle X^\star, a\rangle \in arg\,min_x\{f(x, \langle X^\star, a\rangle, \epsilon)\}, \qquad \text{for almost surely } a \sim \mathcal{N}(0, K) \text{ and } \epsilon.$$

While at first, this assumption seems quite strong, in fact, in a typical student-teacher setup when label noise is 0 (i.e., $\epsilon = 0$), where the targets have the same model as the outputs, the assumption is satisfied. Our goal here is not to be exhaustive, but simply to illustrate that our framework admits a nontrivial and useful analysis and which gives nontrivial conclusions for the optimization theory of these problems.

**Definition D.1** ($\hat{L}$-smoothness of outer function $f$). *A function $f : \mathbb{R}^3 \to \mathbb{R}$ that is $C^1$-smooth (in the first variable) is called $\hat{L}(f)$-smooth if the following quadratic upper bound holds for any $x, \hat{x}, y, z \in \mathbb{R}$*

$$f(\hat{x}, y, z) \le f(x, y, z) + \langle f'(x, y, z), \hat{x} - x \rangle + \frac{\hat{L}(f)}{2}|\hat{x} - x|^2. \tag{58}$$

Note that if $f' = \frac{\partial}{\partial x}f(x, y, z)$ is $\hat{L}(f)$-Lipschitz, i.e., $|f'(x, y, z) - f'(\hat{x}, y, z)| \le \hat{L}(f)|x - \hat{x}|$, then the inequality (58) holds with constant $\hat{L}$. Suppose $x^\star \in \arg\min_x\{f(x, y, z)\}$ exists. An immediate consequence of (58) is that

$$\frac{1}{2\hat{L}(f)}|f'(x, y, z)|^2 \le f(x, y, z) - f(x^\star, y, z) \le \frac{\hat{L}(f)}{2}|x - x^\star|^2. \tag{59}$$

**Definition D.2** (Restricted Secant Inequality). *A function $f : \mathbb{R}^3 \to \mathbb{R}$ that is $C^1$-smooth (in the first variable) satisfies the $(\mu, \theta)$–restricted secant inequality (RSI) if, for any $x \in \mathbb{R}$ and $x^\star \in \arg\min_x\{f(x)\}$,*

$$\langle x - x^\star, f'(x) \rangle \ge \begin{cases} \mu|x - x^\star|^2, & \text{if } \max\{|x^\star|^2, |x - x^\star|^2\} \le \theta, \\ 0, & \text{otherwise.} \end{cases}$$

*If $f$ satisfies the above for $\theta = \infty$, then we say $f$ satisfies the $\mu$–RSI.*

**Proposition D.1.** *Let the outer function $f : \mathbb{R}^3 \to \mathbb{R}$ be a $\hat{L}(f)$-smooth function satisfying the RSI condition with $\hat{\mu}(f)$ with respect to $x \in \mathbb{R}$. Suppose $X^\star \in \arg\min_X\{\mathcal{R}(X)\}$ exists bounded, independent of $d$ and Assumption 7 holds and that $\gamma_0 = \frac{\eta}{b} = \frac{2\hat{\mu}(f)}{(\hat{L}(f))^2\frac{1}{d}\text{Tr}(K)}\zeta$, for some $\zeta \in (0, 1)$, and that $\int_0^\infty \mathcal{R}(s)\gamma_s\,\mathrm{d}s < \infty$ with $\gamma_s$ as in Table 2 (AdaGrad-Norm, general formula), then*

$$\gamma_\infty \ge \frac{\gamma_0\eta^2}{1 + \frac{\zeta}{1-\zeta}\mathscr{D}^2(0)}.$$

*Proof.* Given the Eq. (87) for the distance to optimality, with $(x, x^\star) \sim \mathcal{N}(0, \mathscr{B})$,

$$\frac{\mathrm{d}}{\mathrm{d}t}\mathscr{D}^2(t) = -2\gamma_t\,\mathbb{E}_{a,\epsilon}[\langle x - x^\star, f'(x, x^\star)\rangle] + \frac{\gamma_t^2}{d}\text{Tr}(K)\mathbb{E}_{a,\epsilon}[(f'(x, x^\star))^2]$$

By the RSI (with constant $\hat{\mu}(f)$) condition on $f$, we have that

$$\mathbb{E}_{a,\epsilon}\big[\langle x - x^\star, f'(x, x^\star)\rangle\big] \ge \hat{\mu}(f)\mathbb{E}_{a,\epsilon}[(x - x^\star)^2] = 2\hat{\mu}(f)\mathcal{R}(t), \tag{60}$$

where $x = \langle X, a \rangle$ and $x^\star = \langle X^\star, a \rangle$ and we note that $x$ has $t$-dependence due to the $t$-dependence in $\mathscr{B}$. By $\hat{L}(f)$-smoothness,

$$\frac{1}{2\hat{L}(f)}(f'(x))^2 \le \frac{\hat{L}(f)}{2}(x - x^\star)^2.$$

This implies that

$$\frac{1}{2(\hat{L}(f))^2}\mathbb{E}_{a,\epsilon}\big[(f'(x, x^\star)^2\big] \le \frac{1}{2}\mathbb{E}_{a,\epsilon}\big[(x - x^\star)^2\big] = \mathcal{R}(t). \tag{61}$$

Thus by (60) and (61), we have that

$$\frac{\mathrm{d}}{\mathrm{d}t}\mathscr{D}^2(t) \le -\gamma_t\left(4\hat{\mu}(f) - 2(\hat{L}(f))^2\frac{1}{d}\text{Tr}(K)\gamma_t\right)\mathcal{R}(t)$$

Which then yield:

$$\mathscr{D}^2(t) \le \mathscr{D}^2(0) - 2\left(2\hat{\mu}(f) - (\hat{L}(f))^2\frac{1}{d}\text{Tr}(K)\gamma_0\right)\int_0^t \mathcal{R}(s)\gamma_s\,\mathrm{d}s.$$

Changing variables $u = \Gamma(t) = \int_0^t \gamma_s \, ds$, we have that $\int_0^\infty \mathscr{R}(t)\gamma_t \, dt = \int_0^\infty r(u) \, du = \|r\|_1$. Rearranging the term in the above equation and taking $t \to \infty$. We obtain: $\|r\|_1 \leq \frac{\mathscr{D}^2(0)}{\left(2\hat{\mu}(f)-(\hat{L}(f))^2\frac{1}{d}\operatorname{Tr}(K)\gamma_0\right)}$, given that $\frac{2\hat{\mu}(f)}{(\hat{L}(f))^2\frac{1}{d}\operatorname{Tr}(K)} > \gamma_0$. Using Lemma D.1, with $i(v) = I(\mathscr{B}(\Gamma^{-1}(v))) = \mathbb{E}_{a,\epsilon}\left[(f'(x,x^\star)^2\right]$ instead of the risk

$$\gamma_\infty = \frac{\eta^2}{\frac{b}{\eta} + \frac{1}{2d}\operatorname{Tr}(K)\int_0^\infty i(v)\,dv} \geq \frac{\eta^2}{\frac{b}{\eta} + \frac{1}{d}\operatorname{Tr}(K)(\hat{L}(f))^2\int_0^\infty r(v)\,dv} \tag{62}$$

$$\geq \frac{\eta^2}{\frac{b}{\eta} + \frac{1}{d}\operatorname{Tr}(K)\frac{(\hat{L}(f))^2\mathscr{D}^2(0)}{\left(2\hat{\mu}(f)-(\hat{L}(f))^2\frac{1}{d}\operatorname{Tr}(K)\gamma_0\right)}} = \frac{\eta^2}{\frac{b}{\eta} + \frac{\frac{1}{d}\operatorname{Tr}(K)(\hat{L}(f))^2}{2\hat{\mu}(f)(1-\zeta)}\mathscr{D}^2(0)}.$$

where the first inequality is by Eq. 61, and the last transition is by taking the initial learning rate to be $\gamma_0 = \frac{2\hat{\mu}(f)}{(\hat{L}(f))^2\frac{1}{d}\operatorname{Tr}(K)}\zeta$, for $\zeta \in (0,1)$. $\qquad\square$

**Lemma D.1.** *Given $\gamma_t$ as in Table 2 (AdaGrad-Norm), defining $g(u) = \gamma(\Gamma^{-1}(u))$, with $\Gamma(t) = \int_0^t \gamma_s \, ds$, then $g(u) = \frac{\eta^2}{\frac{b}{\eta} + \frac{1}{2d}\operatorname{Tr}(K)\int_0^u i(v)\,dv}$ with $i(v) = I(\mathscr{B}(\Gamma^{-1}(v)))$.*

*Proof.* Taking the square of both sides of the $\gamma_t$ equation in Table 2 (AdaGrad-Norm), changing variables to $u = \Gamma(t)$ and rearranging the terms:

$$b^2 + \frac{\operatorname{Tr}(K)}{d}\int_0^u \frac{i(v)}{g(v)}\,dv = \frac{\eta^2}{g(u)^2}, \tag{63}$$

such that $i(v) = I(\mathscr{B}(\Gamma^{-1}(v)))$. Taking derivative with respect to $u$, rearranging terms and integrating leads to the desired result. $\qquad\square$

## D.2 Least squares setting

To study the effect of the structured covariance matrix and cases in which the problem is not strongly convex, we will focus on the linear least square problem. In this setting, the continuum limit of the risk for the AdaGrad-Norm algorithm has the form of a convolutional integral Volterra equation,

$$\mathscr{R}(t) = F(\Gamma(t)) + \int_0^t \gamma_s^2 \mathcal{K}(\Gamma(t) - \Gamma(s))\mathscr{R}(s)\,ds \tag{64}$$

where $\Gamma(t) := \int_0^t \gamma_s \, ds$ with,

$$F(x) \stackrel{\text{def}}{=} \frac{1}{2d}\sum_{i=1}^d \lambda_i \mathscr{D}_i^2(0)e^{-2\lambda_i x}, \tag{65}$$

$$\mathcal{K}(x) \stackrel{\text{def}}{=} \frac{1}{d}\sum_{i=1}^d \lambda_i^2 e^{-2\lambda_i x}. \tag{66}$$

In the following we consider three cases, a strongly convex risk in which the spectrum of the eigenvalues is bounded from below (section D.2.1). A case in which the spectrum is not bounded from below as $d \to \infty$, but the number of eigenvalues below some fixed threshold is $o(d)$ (section D.2.2). Finally, power law spectrum supported on $[0,1]$ with $d \to \infty$ (section D.2.3).

### D.2.1 Proofs for case of fixed $d$

*Proof of Proposition 4.2.* Define the composite functions $r(u) = \mathscr{R}(\Gamma^{-1}(u))$, and $g(u) = \gamma(\Gamma^{-1}(u))$. Integrating the formula for the risk:

$$\int_0^t r(u)\,\mathrm{d}u = \int_0^t F(u)\,\mathrm{d}u + \int_0^t \int_0^{\Gamma^{-1}(u)} \gamma_s^2 \mathcal{K}(u - \Gamma(s))\mathscr{R}(s)\,\mathrm{d}s\,\mathrm{d}u$$

$$= \int_0^t F(u)\,\mathrm{d}u + \int_0^t \int_0^u \mathcal{K}(u-x)r(x)g(x)\,\mathrm{d}x\,\mathrm{d}u$$

$$\leq \int_0^t F(u)\,\mathrm{d}u + \gamma_0 \int_0^t r(x) \int_x^t \mathcal{K}(u-x)\,\mathrm{d}u\,\mathrm{d}x$$

Taking $t \to \infty$, we get

$$\|r\|_1 \leq \|F\|_1 + \gamma_0 \|\mathcal{K}\|_1 \|r\|_1.$$

Using $\|\mathcal{K}\|_1 = \int_0^\infty \mathcal{K}(x)\,\mathrm{d}x < \gamma_0^{-1}$, and noting that by Eq. (66), and Eq. (65), we have that $\|F\|_1 = \frac{1}{4}\mathscr{D}^2(0)$, and $\|\mathcal{K}\|_1 = \frac{1}{2d}\operatorname{Tr}(K)$,

$$\|r\|_1 \leq \frac{\|F\|_1}{1 - \gamma_0\|\mathcal{K}\|_1} = \frac{\frac{1}{4}\mathscr{D}^2(0)}{1 - \frac{\gamma_0}{2d}\operatorname{Tr}(K)}.$$

On the hand following Lemma D.3, $\frac{1}{4}\mathscr{D}^2(0)(1 + \frac{\gamma_0}{2d}\operatorname{Tr}(K)) \leq \|r\|_1$. Therefore, $\|r\|_1 \asymp \frac{1}{4}\mathscr{D}^2(0)$.

Next, rewriting the $\gamma_t$ equation in Table 2 (AdaGrad-Norm for least squares) in terms of $g(u)$ (Lemma D.1), we obtain

$$g(u) = \frac{\eta^2}{\frac{b}{\eta} + \frac{1}{d}\operatorname{Tr}(K)\int_0^u r(x)\,\mathrm{d}x} \tag{67}$$

Taking $u \to \infty$, and using $\|r\|_1 \asymp \frac{1}{4}\mathscr{D}^2(0)$,

$$\gamma_\infty = g(\infty) = \frac{\eta^2}{\frac{b}{\eta} + \frac{1}{d}\operatorname{Tr}(K)\|r\|_1} \asymp \frac{\eta^2}{\frac{b}{\eta} + \frac{1}{4d}\operatorname{Tr}(K)\mathscr{D}^2(0)}. \tag{68}$$

This then completes the proof. $\qquad\square$

**Remark D.1.** *We note that, on the Least square problem $\hat{L}(f) = \hat{\mu}(f) = 1$, therefore, the bound in Proposition D.1 yields $\frac{\eta^2}{\frac{b}{\eta} + \frac{1}{2(1-\varsigma)}\frac{1}{d}\operatorname{Tr}(K)\mathscr{D}^2(0)}$.*

*Proof of Proposition 4.1.* Using the equation for the distance to optimality (Eq. 8), we can derive an equation for the integral of the risk (with no target noise) which we denote by $g(t) = \int_0^t \mathscr{R}(s)\,\mathrm{d}s$:

$$g''(t) = -\gamma_t \sum_i \lambda_i^2 \mathscr{D}_i^2(t) + \gamma_t^2 \frac{\operatorname{Tr}(K^2)}{d} g'(t). \tag{69}$$

For $K = I_d$, this equation simplifies,

$$g''(t) = -2\gamma_t g'(t) + \gamma_t^2 \frac{\operatorname{Tr}(K^2)}{d} g'(t). \tag{70}$$

Plugging in the equation for the AdaGrad-Norm learning rate (Table 2) leads to the desired result. We note that by using the equation for the learning rate, one can also derive a close equation for the learning rate itself. $\qquad\square$

### D.2.2 Vanishingly few eigenvalues near 0 as $d \to \infty$

We now consider the case where, as $d \to \infty$, there are eigenvalues of $K$ arbitrarily close to 0. In Proposition 4.2 we saw a constant lower bound on $\gamma_t$ when $d$ is fixed (and thus there are finitely many eigenvalues within any fixed distance of 0). This can be extended to the case where we have some $C > 0$ such that the number of eigenvalues of $K$ below $C$ is $o(d)$ (see Proposition 4.3).

*Proof of Proposition 4.3.* Following the structure of the loss, after some time the risk starts to decrease, and therefore $\mathscr{R}(t) \leq R_0$ for and $t \geq 0$. Using these observations, we obtain a preliminary lower bound of $\gamma_t > C_1 t^{-1/2}$ (for $t > 0$), which enables us to deduce that $\mathscr{R}(t)$ is integrable and finally obtain a constant lower bound for $\gamma_t$. The details of this are below.

For $t \geq 0$ and some $C_1 > 0$,

$$\gamma_t = \frac{\eta}{\sqrt{b^2 + \frac{2}{d}\operatorname{Tr}(K)\int_0^t \mathscr{R}(s)ds}} \geq \frac{\eta}{\sqrt{b^2 + \frac{2}{d}\operatorname{Tr}(K)R_0 t}} \geq C_1 t^{-1/2}. \tag{71}$$

Next, to show that the risk is integrable, we divide the matrix $K$ into two parts $K_+$, and $K_-$, such that the eigenvalues of $K_+$ are greater than some $\alpha_s > 0$ and the eigenvalues of $K_-$ are smaller than $\alpha_s$ where $\alpha_s$ is a decreasing function of $s$ to be determined later. We then have that, following Eq. (8), and the definition of the risk $\mathscr{R}(t) = \frac{1}{2d}\sum_{i=1}^d \lambda_i \mathscr{D}_i^2(t)$,

$$\mathscr{R}(t) = \mathscr{R}(0) - \frac{1}{d}\sum_{i=1}^d \lambda_i^2 \int_0^t \gamma_s \mathscr{D}_i(s)ds + \frac{1}{d}\int_0^t \gamma_s^2 \operatorname{Tr}(K^2) \cdot \mathscr{R}(s)ds \tag{72}$$

$$\leq \mathscr{R}(0) - \int_0^t \gamma_s(2\alpha_s - \gamma_s \frac{1}{d}\operatorname{Tr}(K^2)) \cdot \mathscr{R}(s)ds + 2\int_0^t \gamma_s \mathscr{R}_2(s)\,ds$$

with $\mathscr{R}_2(s) = \frac{1}{2d}\sum_{i:\lambda_i \leq \alpha_s} \lambda_i \mathscr{D}_i^2(s)$. Next, choosing $\alpha_s = \gamma_s \frac{1}{d}\operatorname{Tr}(K^2)$, we show that the last term is of order $o_d(1)$. By Lemma D.2 $\forall i$, $\mathscr{D}_i^2(t) \leq \max\left(\gamma_{t_1}\mathscr{R}(t_1), \mathscr{D}_i^2(0)\right) = c_0$ where the bound $c_0$ comes from the assumption $\langle X^\star, \omega_i \rangle = O(d^{-1/2})$ and the initialization $X_0 = 0$. Therefore,

$$2\int_0^t \gamma_s \mathscr{R}_2(s)\,ds \leq \frac{1}{d^2}\operatorname{Tr}(K^2)c_0 \int_0^t \gamma_s N_s\,ds. \tag{73}$$

where $N_s = \sum_{i=1}^d \mathbf{1}_{\lambda_i \leq \gamma_s \frac{1}{d}\operatorname{Tr}(K^2)}$. This implies that, if $\gamma_s N_s = o(d)$, then $2\int_0^t \gamma_s \mathscr{R}_2(s)\,ds = o_d(1)$, provided that $d$ is taken to be large before $t$.

We then have that up to $o_d(1)$ constant,

$$\mathscr{R}(t) \leq \mathscr{R}(0) - \frac{1}{d}\operatorname{Tr}(K^2)\int_0^t \gamma_s^2 \cdot \mathscr{R}(s)ds. \tag{74}$$

Using Gronwall's inequality,

$$\mathscr{R}(t) \leq \mathscr{R}(0)e^{-\frac{1}{d}\operatorname{Tr}(K^2)\int_0^t \gamma_s^2\,ds} \leq \mathscr{R}(0)e^{-\frac{1}{d}\operatorname{Tr}(K^2)C_1^2 t} \tag{75}$$

where in the last transition we used the lower bound on the learning rate derived in Eq. (71). Thus, the risk is integrable, i.e. there is some $C_3$ such that

$$\int_0^t \mathscr{R}(s)\,ds \leq \frac{\mathscr{R}(0)}{\frac{1}{d}\operatorname{Tr}(K^2)C_1^2}$$

for all $t > 0$. Finally, we plug this into the formula for $\gamma_t$ and conclude that, for all $t > 0$,

$$\gamma_t \geq \frac{\eta}{\sqrt{b^2 + \frac{\frac{1}{d}\operatorname{Tr}(K)\mathscr{R}(0)}{\frac{1}{d}\operatorname{Tr}(K^2)C_1^2}}}. \tag{76}$$

$\square$

**Lemma D.2.** *Assume that the risk is bounded and attains its maximum at time $t_1$. Then, for each $i$, we have $\mathscr{D}_i^2(t) \leq \max(\gamma_{t_1}\mathscr{R}(t_1), \mathscr{D}_i^2(0))$ for all $t \geq 0$.*

*Proof.* Case 1: Suppose that $\mathscr{D}_i^2(0) \leq \gamma_0 \mathscr{R}(0)$. Then, by equation (8), $\frac{d}{dt}\mathscr{D}_i^2(0) \geq 0$. However, since $\mathscr{D}_i^2(t), \mathscr{R}(t)$ are continuous, this equation implies that $\mathscr{D}_i^2(t) \leq \gamma_t \mathscr{R}(t)$ for all $t$ and thus $\mathscr{D}_i^2(t) \leq \gamma_{t_1}\mathscr{R}(t_1)$ for all $t$.

Case 2: Suppose that $\mathscr{D}_i^2(0) > \gamma_0 \mathscr{R}(0)$. Then, by equation (8), $\frac{d}{dt}\mathscr{D}_i^2(0) < 0$. If $\frac{d}{dt}\mathscr{D}_i^2(t) < 0$ for all $t$, then $\mathscr{D}_i^2(t) \leq \mathscr{D}_i^2(0)$ for all $t$. If at some point $\frac{d}{dt}\mathscr{D}_i^2(t) > 0$, this implies $\mathscr{D}_i^2(t) \leq \gamma_t \mathscr{R}(t)$ and we are in Case 1. $\square$

In the next section, we consider cases in which the risk is not integrable, an example of such case is when the spectrum of $K$ is supported on the interval $[0, 1]$ or has power-law behavior near 0.

### D.2.3 Power law behavior at $d \to \infty$

**Non-asymptotic bound for the Convolutional Volterra**    In this section, we use the convolutional Volterra structure of the risk (Eq. (64)) to derive non-asymptotic bounds on the risk, which will be useful in Section D.2.3 to derive the asymptotic behavior of the risk and the learning rate under power law assumption on the spectrum of the covariance matrix and the discrepancy from the target at initialization.

**Lemma D.3.** *Let* $\Gamma(t) := \int_0^t \gamma_s \, \mathrm{d}s$ *and let*

$$\mathcal{R}(t) = F(\Gamma(t)) + \int_0^t \gamma_s^2 \mathcal{K}(\Gamma(t) - \Gamma(s))\mathcal{R}(s) \, \mathrm{d}s$$

*where* $\gamma_t, \mathcal{K}$ *are monotonically decreasing, with* $\|\mathcal{K}\|_1 < \infty$. *Then all t,*

$$\mathcal{R}(t) \geq F(\Gamma(t)) + \int_0^t \gamma_s^2 \mathcal{K}(\Gamma(t) - \Gamma(s))F(\Gamma(s)) \, \mathrm{d}s$$

*If in addition, there exist* $\epsilon > 0$ *and* $T > 0$ *such that, for all* $t > T$,

$$\int_0^t \mathcal{K}(s)\mathcal{K}(t - s) \, \mathrm{d}s \leq 2(1 + \epsilon)\|\mathcal{K}\|_1 \mathcal{K}(t) \quad and \quad 2\|\mathcal{K}\|_1(1 + \epsilon)\gamma_0 < 1$$

*then for all t*

$$\mathcal{R}(t) \leq F(\Gamma(t)) + C \int_0^t \gamma_s^2 \mathcal{K}(\Gamma(t) - \Gamma(s))F(\Gamma(s)) \, \mathrm{d}s$$

*for*

$$C = \left(\frac{\mathcal{K}(0)}{\mathcal{K}(T)(2\epsilon + 1)} + 2\right) \frac{1}{1 - 2\gamma(0)\|\mathcal{K}\|_1(1 + \epsilon)}.$$

*Proof.* The lower bound holds trivially, using $\mathcal{R}(s) \geq F(\Gamma(s))$. For the upper bound, we start with the following change of variables:

$$\mathcal{R}(t) = F(\Gamma(t)) + \int_0^{\Gamma(t)} g(u)\mathcal{K}(\Gamma(t) - u))\mathcal{R}(u) \, \mathrm{d}u,$$

with $g(u) = \gamma_{\Gamma^{-1}(u)}$. Let us define the convolution map

$$\mathcal{G}(f)(\Gamma) = \mathcal{K} * (gf)(\Gamma) = \int_0^\Gamma \mathcal{K}(\Gamma - u)g(u)f(u) \, \mathrm{d}u.$$

Next we show that this map is contracting and in particular,

$$\mathcal{G}^2(f) = \mathcal{G}(\mathcal{G}(f))(t) = \int_0^t \mathcal{K}(t - s)\mathcal{G}(f)(s)g(s) \, \mathrm{d}s \tag{77}$$

$$= \int_0^t \mathcal{K}(t - s) \int_0^s \mathcal{K}(s - u)g(u)f(u) \, \mathrm{d}u g(s) \, \mathrm{d}s$$

$$= \int_0^t \left(\int_u^t \mathcal{K}(t - s)\mathcal{K}(s - u)g(s) \, \mathrm{d}s\right) g(u)f(u) \, \mathrm{d}u$$

$$\leq \int_0^t \mathcal{K}^{*2}(t - u)g(u)^2 f(u) \, \mathrm{d}u$$

where the third transition is since $u < s < t$. The last transition is by change of variables and the assumption that $\gamma_t$ is a monotone decreasing function. Consecutive application of the convolution map will then yield by induction,

$$\mathcal{G}^j(f)(t) \leq \int_0^t \mathcal{K}^{*(j)}(t - u)g(u)^j f(u) \, \mathrm{d}u.$$

Therefore, expanding the loss and using the above upper bound, and denote by $q = 2(1 + \varepsilon)\|\mathcal{K}\|_1 \gamma_0$ such that $q < 1$,

$$\mathscr{R}(t) = F(t) + \sum_{j=1}^{\infty} \mathcal{G}^j(F)(t) \tag{78}$$

$$\leq F(t) + \sum_{j=1}^{\infty} \int_0^t \mathcal{K}^{*(j)}(t-u)g(u)^j F(u)\,\mathrm{d}u$$

$$\leq F(t) + \left(\sum_{j=0}^{\infty} (2\|\mathcal{K}\|_1 \gamma_0 (1+\varepsilon))^j - 1\right) C_1 \int_0^t \mathcal{K}(t-u)g(u)F(u)\,\mathrm{d}u$$

$$\leq F(t) + \frac{q}{1-q} C_1 (\mathcal{K} * (gF))(t) \tag{79}$$

where the third transition is by Lemma D.4, with $C_1 = \frac{\mathcal{K}(0)}{\mathcal{K}(T)(2\epsilon+1)} + 1$, which then completes the proof. $\qquad\square$

**Lemma D.4** (Lemma IV.4.7 in [3]). *Suppose $\mathcal{K}$ is monotonically decreasing, with $\|\mathcal{K}\|_1 < \infty$, and that there exists $T > 0$ such that $\forall t \geq T$, and $\epsilon \geq 0$,*

$$\int_0^t \mathcal{K}(s)\mathcal{K}(t-s)\,\mathrm{d}s \leq 2(1+\epsilon)\|\mathcal{K}\|_1 \mathcal{K}(t). \tag{80}$$

*Then,*

$$\sup_{t\geq 0} \frac{\mathcal{K}^{*n}(t)}{\mathcal{K}(t)} \leq (2\|\mathcal{K}\|_1 (1+\epsilon))^{n-1} \left(\frac{\mathcal{K}(0)}{\mathcal{K}(T)(2\epsilon+1)} + 1\right) \tag{81}$$

*Proof.* Define $\alpha_n = \sup_{t\geq 0} \frac{\mathcal{K}^{*n}(t)}{\mathcal{K}(t)(2\|\mathcal{K}\|_1)^{n-1}}$, trivially $\alpha_1 = 1$. Consider the $n+1$ convolution,

$$\frac{\mathcal{K}^{*(n+1)}(t)}{\mathcal{K}(t)(2\|\mathcal{K}\|_1)^n} = \frac{1}{\mathcal{K}(t)} \int_0^t \frac{\mathcal{K}(s)\mathcal{K}^{*n}(t-s)}{(2\|\mathcal{K}\|_1)^n}\,\mathrm{d}s \tag{82}$$

By the assumption of the Lemma, we know that there exists some $T > 0$ such that for $\forall t \geq T$

$$\int_0^t \frac{\mathcal{K}(s)\mathcal{K}(t-s)}{2\|\mathcal{K}\|_1}\,\mathrm{d}s \leq (1+\epsilon)\mathcal{K}(t). \tag{83}$$

Therefore, if $t \geq T$, we have

$$\frac{1}{\mathcal{K}(t)} \int_0^t \frac{\mathcal{K}(s)\mathcal{K}^{*n}(t-s)}{(2\|\mathcal{K}\|_1)^n}\,\mathrm{d}s \tag{84}$$

$$= \int_0^t \frac{\mathcal{K}(s)\mathcal{K}(t-s)}{2\|\mathcal{K}\|_1} \frac{\mathcal{K}^{*n}(t-s)}{\mathcal{K}(t-s)(2\|\mathcal{K}\|_1)^{n-1}}\,\mathrm{d}s \leq \alpha_n(1+\epsilon)$$

On the other hand, if $t < T$,

$$\frac{1}{\mathcal{K}(t)} \int_0^t \frac{\mathcal{K}(s)\mathcal{K}^{*n}(t-s)}{(2\|\mathcal{K}\|_1)^n}\,\mathrm{d}s \leq \frac{\mathcal{K}(0)}{\mathcal{K}(T)} \frac{\|\mathcal{K}^{*n}(t)\|_1}{(2\|\mathcal{K}\|_1)^n} \leq \frac{\mathcal{K}(0)}{\mathcal{K}(T)2^n} \tag{85}$$

Taking supremum in Eq. (82), and combining the results of Eq. (85), and Eq. (84), we obtain that,

$$\alpha_{n+1} \leq \frac{\mathcal{K}(0)}{\mathcal{K}(T)2^n} + \alpha_n(1+\epsilon)$$

Solving the above recursion equation,

$$\alpha_n \leq \frac{\mathcal{K}(0)}{\mathcal{K}(T)} \sum_{k=0}^{n-2} \frac{1}{2^{n-k-1}}(1+\epsilon)^k + (1+\epsilon)^{n-1} = \frac{\mathcal{K}(0)}{\mathcal{K}(T)2^{n-1}} \frac{1 - (2(1+\epsilon))^{n-1}}{1 - 2(1+\epsilon)} + (1+\epsilon)^{n-1}$$

$$\leq (1+\epsilon)^{n-1} \left(\frac{\mathcal{K}(0)}{\mathcal{K}(T)(2\epsilon+1)} + 1\right),$$

rearranging the terms we arrived at the required result. $\qquad\square$

**Asymptotic analysis of the risk** Here, we consider a family of models with $d \to \infty$, for which the following power law asymptotics assumption is satisfied:

**Assumption 8.** $F(x) \asymp x^{-\kappa_1}$ and $\mathcal{K}(x) \asymp x^{-\kappa_2}$ for $x \geq 1$ with $\kappa_1 \geq 0$, $\kappa_2 > 1$

Corollary D.1 apply Lemma D.3 in the setting for which $F$, and $\mathcal{K}$ has a power law behavior asymptotically. It shows that the risk will then be dominated by $F$ only. Corollary D.2 shows the behavior of the learning rate in this setting. Finally, Lemma D.5 shows that Assumption 8 is a consequence of a power law spectrum near zero on the eigenvalues of the covariance matrix and a power law assumption on the projected discrepancy at initialization.

**Corollary D.1.** *Suppose Assumption 8 is satisfied, then $\mathcal{R}(t) \asymp F(\Gamma(t))$.*

*Proof.* Define $g(u) = \gamma_{\Gamma^{-1}(u)}$ and $r(u) = \mathcal{R}(\Gamma^{-1}(u))$ and observe that $g(u)$ is a decreasing function. Then, from the upper bound in Lemma D.3, we have

$$
\begin{aligned}
r(u) &\leq F(u) + C \int_0^u g(v)\mathcal{K}(u-v)F(v)\,\mathrm{d}v \\
&= F(u) + C \left( \int_0^{u/2} g(v)\mathcal{K}(u-v)F(v)\,\mathrm{d}v + \int_{u/2}^u g(v)\mathcal{K}(u-v)F(v)\,\mathrm{d}v \right) \\
&\leq F(u) + C_1 g(0) \left( \left(\frac{u}{2}\right)^{-\kappa_2} \int_0^{u/2} F(v)\,\mathrm{d}v + \left(\frac{u}{2}\right)^{-\kappa_1} \int_{u/2}^u \mathcal{K}(u-v)\,\mathrm{d}v \right) \\
&\leq F(u) + C_2 (u^{-\kappa_2+1-\kappa_1} + u^{-\kappa_1}\|\mathcal{K}\|) \\
&= O(F(u)).
\end{aligned}
\tag{86}
$$

Combining this upper bound with the lower bound from Lemma D.3 and that $\kappa_2 > 1$, we conclude that $r(u) \asymp F(u)$ and $\mathcal{R}(t) \asymp F(\Gamma(t))$. $\square$

Next, we derive the asymptotics of $\gamma_t$. There are three different cases, depending on whether the risk is integrable, which translates to a threshold with respect to the parameter $\kappa_1$.

**Corollary D.2.** *Suppose Assumption 8 then the following asymptotics for the learning rate hold:*

- *For $\kappa_1 > 1$, there exists $\tilde{\gamma}$ such that $\gamma_t \geq \tilde{\gamma}$ and $\mathcal{R}(t) \asymp t^{-\kappa_1}$ for all $t \geq 0$.*

- *For $\kappa_1 < 1$, $\gamma_t \asymp t^{-(1-\kappa_1)/(2-\kappa_1)}$ and $\mathcal{R}(t) \asymp t^{-\frac{\kappa_1}{2-\kappa_1}}$ for all $t \geq 1$.*

- *For $\kappa_1 = 1$, $\gamma_t \asymp \frac{1}{\log(t+1)}$ and $\mathcal{R}(t) \asymp \left(\frac{t}{\log(t+1)}\right)^{-\kappa_1}$ for all $t \geq 1$.*

*Proof.* Using the notations $g(u)$ and $r(u)$ defined above along with the change of variable $u = \Gamma(t)$, we get $\int_0^t \mathcal{R}(s)\,\mathrm{d}s = \int_0^u \frac{r(v)}{g(v)}\,\mathrm{d}v$. Combining this with Corollary D.1 and the formula for $\gamma_t$ we get

$$
g(u) \asymp \frac{\eta}{\sqrt{b^2 + \frac{2}{d}\mathrm{Tr}(K)\int_0^u \frac{(1+v)^{-\kappa_1}}{g(v)}\,\mathrm{d}v}}.
$$

Let $I(u) = b^2 + \frac{2}{d}\mathrm{Tr}(K)\int_0^u \frac{(1+v)^{-\kappa_1}}{g(v)}\,\mathrm{d}v$ and observe that $g(u) \asymp \frac{1}{\sqrt{I(u)}}$ and $I'(u) = \frac{2}{d}\mathrm{Tr}(K)\frac{(1+u)^{-\kappa_1}}{g(u)}$. Thus, $I(u)$ satisfies $\frac{I'(u)}{\sqrt{I(u)}} \asymp (1+u)^{-\kappa_1}$ so we have

$$
\sqrt{I(u)} - \sqrt{I(0)} \asymp \int_0^u (1+v)^{-\kappa_1}\,\mathrm{d}v.
$$

In the case of $\kappa_1 > 1$, this implies $\sqrt{I(u)} \leq \sqrt{I(0)} + C\int(1+v)^{-\kappa_1}\,\mathrm{d}v$. This upper bound on $I(u)$ gives a corresponding lower bound on $g(u)$ and thus a lower bound on $\gamma_t$.

In the case of $\kappa_1 < 1$, we have $\sqrt{I(u)} - \sqrt{I(0)} \asymp (1+v)^{1-\kappa_1}$ so, for $u$ sufficiently large, $g(u) \asymp (1+u)^{\kappa_1-1}$. To recover the asymptotic for $\gamma_t$, we observe that $\frac{\mathrm{d}}{\mathrm{d}u}\Gamma^{-1}(u) = \frac{1}{g(u)} \asymp (1+u)^{1-\kappa_1}$. Integrating both sides and changing back to $t$ variables, we get $t \asymp (1+\Gamma(t))^{2-\kappa_1}$ (or equivalently

$1 + \Gamma(t) \asymp t^{1/(2-\kappa_1)}$). Finally, plugging this into the formula for $\gamma_t$ and applying Corollary D.1, we get

$$\gamma_t \asymp \frac{\eta}{\sqrt{b^2 + \frac{2}{d}\text{Tr}(K)\int_0^t F(\Gamma(s))\,\mathrm{d}s}} \asymp (1+t)^{-(1-\kappa_1)/(2-\kappa_1)}.$$

In the case of $\kappa_1 = 1$, we follow a similar procedure as for $\kappa_1 < 1$ to show that $t \asymp \Gamma(t)\log(\Gamma(t))$ for sufficiently large $t$. This implies $\Gamma(t) \asymp t/\log(t)$ which gives the desired result after integration. The decay rate of the risk is then immediate using Corollary D.1. $\qquad\square$

**Lemma D.5.** *Let $K$ have a spectrum that converges as $d \to \infty$ to the power law measure $\rho(\lambda) = C\lambda^{-\beta}\mathbf{1}_{(0,\lambda_{\max})}$, with $C^{-1} = \frac{\lambda_{\max}^{1-\beta}}{1-\beta}$ for some $\beta < 1$, and $\lambda_{\max} > 0$, and suppose that $\mathscr{D}_i^2(0) \sim \lambda_i^{-\delta}$, then $F(t) \asymp t^{-\kappa_1}$, and $\mathcal{K}(t) \asymp t^{-\kappa_2}$, with $\kappa_1 = 2-\beta-\delta$, and $\kappa_2 = 3-\beta$. In addition, $\mathcal{K}(t) \asymp t^{-\kappa_2}$, satisfies Eq. (80).*

*Proof.* Following the definition in Eq. (66), and Eq. (65)

$$F(x) = \frac{1-\beta}{2\lambda_{\max}^{1-\beta}}\int_0^{\lambda_{\max}} \lambda^{1-\beta-\delta}e^{-2\lambda x}\,\mathrm{d}\lambda$$

$$= \frac{1-\beta}{2\lambda_{\max}^{1-\beta}(2x)^{2-\beta-\delta}}\int_0^{2\lambda_{\max}x} y^{1-\beta-\delta}e^{-y}\,\mathrm{d}y = \frac{1-\beta}{\lambda_{\max}^{1-\beta}2^{3-\beta-\delta}}\frac{\gamma(2-\beta-\delta, 2\lambda_{\max}x)}{x^{2-\beta-\delta}}.$$

Similarly for $\mathcal{K}$,

$$\mathcal{K}(x) = \frac{1-\beta}{\lambda_{\max}^{1-\beta}}\int_0^{\lambda_{\max}} \lambda^{2-\beta}e^{-2\lambda x}\,\mathrm{d}\lambda = \frac{1-\beta}{\lambda_{\max}^{1-\beta}2^{3-\beta}}\frac{\gamma(3-\beta, 2\lambda_{\max}x)}{x^{3-\beta}}.$$

with $\gamma(s,z) = \int_0^z x^{s-1}e^{-x}\,\mathrm{d}x$ is the incomplete gamma function. For large $z$, $\gamma(s,z) \asymp \Gamma(s)$, the complete gamma function. We therefore obtain $\kappa_1 = 2-\beta-\delta$, and $\kappa_2 = 3-\beta$. Next, we show that $\mathcal{K}(x) \asymp x^{-\kappa_2}$ satisfies Eq. (80),

$$\int_0^t \mathcal{K}(s)\mathcal{K}(t-s)\,\mathrm{d}s \leq \int_0^{t/2}\mathcal{K}(t)\mathcal{K}(t-s)\,\mathrm{d}s + \int_{t/2}^t \mathcal{K}(t)\mathcal{K}(t-s)\,\mathrm{d}s$$

$$\leq \mathcal{K}(t/2)\left(\int_0^{t/2}\mathcal{K}(s)\,\mathrm{d}s + \int_{t/2}^t \mathcal{K}(t-s)\,\mathrm{d}s\right) \leq 2\mathcal{K}(t/2)\|\mathcal{K}\|_1$$

by the power-law assumption for $t > T$, $\mathcal{K}(t/2) \asymp \mathcal{K}(t)$ which then complete the proof. $\qquad\square$

*Proof of Proposition 4.4.* The proof is an immediate application of Corollary D.2 with, $\kappa_1 = 2-\beta-\delta$ as implied by Lemma D.5. $\qquad\square$

**Remark D.2.** *This includes the case $\beta = 0$, which is the uniform measure on $[0, \lambda_{\max}]$.*

# E  Polyak Stepsize

The distance to optimality of SGD is measured say by $D^2(X) = \|X - X^\star\|^2$. Let us consider the deterministic equivalent for the distance to optimality $\mathscr{D}^2(t)$ in (11). Fixing $T > 0$ and any $\varepsilon \in (0, 1/2)$, we have by Theorem 2.1 (see also corollary B.1 which show concentration for large class of statistics) that $\sup_{0 \leq t \leq T}\|\|X_{\lfloor td\rfloor} - X^\star\|^2 - \mathscr{D}^2(t)\| \leq d^{-\varepsilon}$, w.o.p. In this way, if we want to guarantee that the distance to optimality of SGD decreases, we need $\mathrm{d}\mathscr{D}^2(t) < 0$ with the maximum decrease being $\min_{\gamma_t}\mathrm{d}\mathscr{D}^2(t)$.

As it turns out, the evolution of $\mathscr{D}^2$ is particular simple, as it solves the differential equation (derived from the ODE in (9))

$$\frac{\mathrm{d}}{\mathrm{d}t}\mathscr{D}^2(t) = -2\gamma_t A(\mathscr{B}(t)) + \frac{\gamma_t^2}{d}\text{Tr}(K)I(\mathscr{B}(t)), \quad \begin{cases} A(\mathscr{B}) = \mathbb{E}_{a,\epsilon}[\langle x - x^\star, f'(x \oplus x^\star)\rangle], \\ I(\mathscr{B}) = \mathbb{E}_{a,\epsilon}[f'(x \oplus x^\star)^2], \quad \text{where} \quad (87) \\ (x \oplus x^\star) \sim N(0, \mathscr{B}). \end{cases}$$

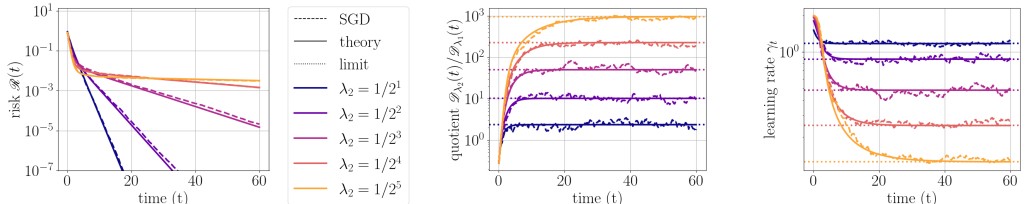

Figure 5: **Convergence in Exact Line Search** on a noiseless least squares problem. The plot on the left illustrates the convergence of the risk function, while the center and right plots depict the convergence of the quotient $\frac{\mathscr{D}_{\lambda_2}(t)}{\mathscr{D}_{\lambda_1}(t)}$ and the learning rate $\gamma_t$, respectively. Further details and formulas for the limiting behavior can be found in the Appendix F.2. See Appendix H for simulation details.

The distance to optimality threshold, $\bar\gamma_t^{\mathscr{D}}$, occurs precisely when $\mathrm{d}\mathscr{D}^2 < 0$. This choice of $\gamma$ makes the ODE for the distance to optimality stable. By translating the relevant deterministic quantities in $\bar\gamma_t^{\mathscr{D}}$ back to SGD quantities, we get

$$\bar{\mathfrak{g}}_k^{\mathscr{D}} \stackrel{\text{def}}{=} \frac{2\langle X_k - X^\star, \nabla\mathcal{R}(X_k)\rangle}{\frac{\mathrm{Tr}(K)}{d}\mathbb{E}_{a,\epsilon}[f'(\langle X_k, a\rangle; \langle X^\star, a\rangle, \epsilon)^2]} \quad \text{with the deterministic equiv. } \bar\gamma_t^{\mathscr{D}} = \frac{2A(\mathscr{B}(t))}{\frac{\mathrm{Tr}(K)}{d}I(\mathscr{B}(t))}.$$
(88)

A greedy learning rate that maximizes the decrease at each iteration is simply given by $\mathfrak{g}_t^{\text{Polyak}} \in \arg\min \mathrm{d}\mathscr{D}^2(t)$. This has a closed form and we call this *Polyak stepsize*[11]. Again translating this back to SGD, we have

$$\text{Polyak learning rate} \quad \mathfrak{g}_k^{\text{Polyak}} = \tfrac{1}{2}\bar{\mathfrak{g}}_k^{\mathscr{D}} \quad \text{and} \quad \text{deterministic equivalent} \quad \gamma_t^{\text{Polyak}} = \tfrac{1}{2}\bar\gamma_t^{\mathscr{D}}. \quad (89)$$

In this context, the Polyak learning rate is impractical because we do not known $X^\star$. In spite of this, we can learn some things about this learning rate as it is the natural extension of Polyak learning rate to SGD.

The quantities $A(\mathscr{B})$ and $I(\mathscr{B})$ in (88) and (89) only depend on the low-dimensional function $f$ and thus do not carry any covariance $K$ or $d$ dependence. Moreover, under additional assumptions on the function such as (strong) convexity, we can bound from below $A(\mathscr{B})/I(\mathscr{B})$. Thus, in terms covariance $K$ and $d$, the Polyak stepsize $\mathfrak{g}_k^{\text{Polyak}} \asymp \frac{1}{\mathrm{Tr}(K)/(d)} = \frac{1}{\text{avg. eig of } K}$.

In the case of least squares (see (7)), we get

$$\mathfrak{g}_k^{\text{Polyak}} = \frac{2\mathcal{R}(X_k) - \omega^2}{\frac{2\mathrm{Tr}(K)}{d}\mathcal{R}(X_k)} \quad \text{and on a noiseless least squares,} \quad \mathfrak{g}_k^{\text{Polyak}} = \frac{1}{\frac{\mathrm{Tr}(K)}{d}}.$$

The latter gives the best fixed learning rate for a noiseless target on a LS problem (as noted in [35, 43]).

# F Line Search

## F.1 General Line Search

Naturally, one can ask a similar question as in Polyak in the context of line search (i.e., decreasing risk at each iteration of SGD). First, by the structure of the risk (Assumption 3 and 4),

$$\|\nabla\mathcal{R}(X)\|^2 = m(W^T K^2 W) \quad \text{and} \quad \mathrm{Tr}(\nabla^2\mathcal{R}(X)K) = v(K). \quad (90)$$

Therefore using (9), we have that the deterministic equivalent for $\|\nabla\mathcal{R}(X)\|^2$ is $\mathscr{M}(t) = \frac{1}{2}\sum_{i=1}^d m(\mathscr{V}_i(t)\lambda_i^2)$. In this case, the deterministic equivalent for the risk $\mathscr{R}$ satisfies the following ODE

$$\mathrm{d}\mathscr{R} = -\gamma_t\mathscr{M}(t)\,\mathrm{d}t + \frac{\gamma_t^2}{d}v(K)I(\mathscr{B}(t)). \quad (91)$$

---

[11]This is the idea of Polyak stepsize when the problem is deterministic.

From this, we get an immediate learning rate (stability) threshold for the risk, that is, $\bar{\mathfrak{g}}_k^{\mathscr{R}}$ is the largest learning rate for which SGD is guaranteed to decrease at each iteration, i.e., when the deterministic equivalent of $\mathscr{R}$ satisfies $\mathrm{d}\mathscr{R} < 0$ or equivalently after translating relevant terms into SGD quantities

$$\text{risk threshold} \quad \bar{\mathfrak{g}}_k^{\mathscr{R}} = \frac{\|\nabla\mathcal{R}(X_k)\|^2}{\frac{\mathrm{Tr}(K\nabla^2\mathcal{R}(X_k))}{d} I(W_k^T K W_k)} \quad \text{and deterministic equiv} \quad \bar{\gamma}_t^{\mathscr{R}} = \frac{\mathscr{M}(t)}{\frac{v(K)}{d} I(\mathscr{B}(t))}.$$
(92)

The greediest approach, which we call *exact line search*, would choose the learning rate such that $\gamma_t^{\text{line}} \in \arg\min_\gamma \mathrm{d}\mathscr{R}$. In this case, we get

$$\mathfrak{g}_k^{\text{line}} = \tfrac{1}{2}\mathfrak{g}_k^{\mathscr{R}} \quad \text{and} \quad \text{deterministic equiv} \quad \gamma_t^{\text{line}} = \tfrac{1}{2}\gamma_t^{\mathscr{R}}.$$

## F.2 Line Search on least squares

In this section, we provide a proof of Proposition 3.1, but, we show more than this including the exact limiting value for $\gamma_t$.

**Proposition F.1.** *Consider the noiseless ($\omega = 0$) least squares problem* (7) *. Then the learning rate is always lower bounded by*

$$\frac{\lambda_{\min}(K)}{\frac{1}{d}\mathrm{Tr}(K^2)} \leq \gamma_t^{\text{line}} \quad \text{for all } t \geq 0.$$

*Moreover, suppose $K$ has only two distinct eigenvalues $\lambda_1 > \lambda_2 > 0$, i.e., $K$ has $d/2$ eigenvalues equal to $\lambda_1$ eigenvalues and $d/2$ eigenvalues equal to $\lambda_2$. In this context, the exact limiting value of $\gamma_t^{\text{line}}$ is given by*

$$\lim_{k\to\infty} \gamma_t^{\text{line}} = \frac{2\left(\lambda_1^2 + \lambda_2^2 x\right)}{\left(\lambda_1 + \lambda_2 x\right)\left(\lambda_1^2 + \lambda_2^2\right)},$$
(93)

*where $x$ is the positive real root of the second-degree polynomial*

$$\mathcal{P}(x) = \lambda_1\lambda_2(x+1)(\lambda_2 x - \lambda_1) + (\lambda_2 - \lambda_1)^3 x.$$
(94)

*This leads to*

$$\frac{\lambda_{\min}(K)}{\frac{1}{d}\mathrm{Tr}(K^2)} \leq \lim_{t\to\infty} \gamma_t^{\text{line}} \leq \frac{2\lambda_{\min}(K)}{\frac{1}{d}\mathrm{Tr}(K^2)}.$$
(95)

*Proof.* We establish the inequality

$$\frac{\lambda_{\min}(K)}{\frac{1}{d}\mathrm{Tr}(K^2)} \leq \gamma_t^{\text{line}} \quad \text{for all } t \geq 0$$

by observing

$$\frac{1}{d}\sum_{i=1}^d \lambda_i^2 \mathscr{D}_i^2(t) \geq 2\lambda_{\min}(K)\frac{1}{2d}\sum_{i=1}^d \lambda_i \mathscr{D}_i^2(t) = 2\lambda_{\min}(K)\mathscr{R}(t).$$

Now let us consider $K \sim \tfrac{1}{2}\lambda_1 + \tfrac{1}{2}\lambda_2$ for $\lambda_1 > \lambda_2 > 0$.

We define $\mathscr{D}_\lambda(t) \overset{\text{def}}{=} \sum_{\lambda_i=\lambda}^d \mathscr{D}_i^2(t)$. Utilizing the ODEs in (9), we derive

$$\frac{\mathrm{d}}{\mathrm{d}t}\mathscr{D}_\lambda(t) = -2\gamma_t\lambda\mathscr{D}_\lambda(t) + 2\gamma_t^2\lambda \times |\{\lambda = \lambda_i\}_{i=1}^d| \times \mathscr{R}(t)$$

for each distinct eigenvalue $\lambda$ of $K$. Here $|\{\lambda = \lambda_i\}_{i=1}^d|$ is the number of eigenvalues of $K$ that are equal to $\lambda$. It immediately follows by our construction of $K$ that $|\{\lambda = \lambda_i\}_{i=1}^d| = \frac{d}{2}$. Thus, we establish the following system of ODEs

$$\begin{cases} \frac{\mathrm{d}}{\mathrm{d}t}\mathscr{D}_{\lambda_1}(t) = -2\gamma_t\lambda_1\mathscr{D}_{\lambda_1}(t) + d\gamma_t^2\lambda_1\mathscr{R}(t) \\ \frac{\mathrm{d}}{\mathrm{d}t}\mathscr{D}_{\lambda_2}(t) = -2\gamma_t\lambda_2\mathscr{D}_{\lambda_2}(t) + d\gamma_t^2\lambda_2\mathscr{R}(t) \end{cases}$$
(96)

where $\mathscr{R}(t) = \frac{1}{2d}\left(\lambda_1\mathscr{D}_{\lambda_1}(t) + \lambda_2\mathscr{D}_{\lambda_2}(t)\right)$ and $\gamma_t^{\text{line}} = \frac{2\left(\lambda_1^2\mathscr{D}_{\lambda_1}(t)+\lambda_2^2\mathscr{D}_{\lambda_2}(t)\right)}{\left(\lambda_1\mathscr{D}_{\lambda_1}(t)+\lambda_2\mathscr{D}_{\lambda_2}(t)\right)\left(\lambda_1^2+\lambda_2^2\right)}$.

Since $\mathscr{D}_{\lambda_2}(t) \geq 0$ and $\lambda_1 > \lambda_2 > 0$, we infer that $\mathscr{R}(t) = \frac{1}{2d}\left(\lambda_1\mathscr{D}_{\lambda_1}(t) + \lambda_2\mathscr{D}_{\lambda_2}(t)\right) \geq \frac{1}{2d}\lambda_1\mathscr{D}_{\lambda_1}(t) \geq 0$. The structure of the exact line search algorithm ensures $\lim_{t\to\infty}\mathscr{R}(t) = 0$, hence $\lim_{t\to\infty}\mathscr{D}_{\lambda_1}(t) = 0$. Similarly, we deduce $\lim_{t\to\infty}\mathscr{D}_{\lambda_2}(t) = 0$.

By applying L'Hôpital's rule and substituting the expressions for $\gamma_t^{\mathrm{line}}$ and $\mathscr{R}(t)$ in terms of $\mathscr{D}_{\lambda_1}(t)$ and $\mathscr{D}_{\lambda_2}(t)$, we derive

$$
\lim_{t\to\infty}\frac{\mathscr{D}_{\lambda_2}(t)}{\mathscr{D}_{\lambda_1}(t)} = \lim_{t\to\infty}\frac{\mathrm{d}\mathscr{D}_{\lambda_2}(t)}{\mathrm{d}\mathscr{D}_{\lambda_1}(t)}
$$

$$
= \lim_{t\to\infty}\frac{-2\gamma_t\lambda_2\,\mathscr{D}_{\lambda_2}(t) + d\gamma_t^2\lambda_2\mathscr{R}(t)}{-2\gamma_t\lambda_1\,\mathscr{D}_{\lambda_1}(t) + d\gamma_t^2\lambda_1\mathscr{R}(t)}
$$

$$
= \lim_{t\to\infty}\frac{-2\lambda_2\,\mathscr{D}_{\lambda_2}(t) + d\gamma_t\lambda_2\mathscr{R}(t)}{-2\lambda_1\,\mathscr{D}_{\lambda_1}(t) + d\gamma_t\lambda_1\mathscr{R}(t)}
$$

$$
= \lim_{t\to\infty}\frac{\gamma_t\frac{\lambda_1\lambda_2}{2}\mathscr{D}_{\lambda_1}(t) + \lambda_2\mathscr{D}_{\lambda_2}(t)\left(\gamma_t\frac{\lambda_2}{2} - 2\right)}{\gamma_t\frac{\lambda_1\lambda_2}{2}\mathscr{D}_{\lambda_2}(t) + \lambda_1\mathscr{D}_{\lambda_1}(t)\left(\gamma_t\frac{\lambda_1}{2} - 2\right)}
$$

$$
= \lim_{t\to\infty}\frac{\mathscr{D}_{\lambda_1}(t)^2\lambda_1^3\lambda_2 + \mathscr{D}_{\lambda_1}(t)\mathscr{D}_{\lambda_2}(t)(-\lambda_1\lambda_2^3 + \lambda_1^2\lambda_2^2 - 2\lambda_1^3\lambda_2) + \mathscr{D}_{\lambda_2}(t)^2(-\lambda_2^4 - 2\lambda_1^2\lambda_2^2)}{\mathscr{D}_{\lambda_1}(t)^2(-\lambda_1^4 - 2\lambda_1^2\lambda_2^2) + \mathscr{D}_{\lambda_1}(t)\mathscr{D}_{\lambda_2}(t)(-\lambda_1^3\lambda_2 + \lambda_1^2\lambda_2^2 - 2\lambda_1\lambda_2^3) + \mathscr{D}_{\lambda_2}(t)^2\lambda_1\lambda_2^3}
$$

$$
= \frac{\lambda_1^3\lambda_2 + \lim_{t\to\infty}\frac{\mathscr{D}_{\lambda_2}(t)}{\mathscr{D}_{\lambda_1}(t)}(-\lambda_1\lambda_2^3 + \lambda_1^2\lambda_2^2 - 2\lambda_1^3\lambda_2) + \left(\lim_{t\to\infty}\frac{\mathscr{D}_{\lambda_2}(t)}{\mathscr{D}_{\lambda_1}(t)}\right)^2(-\lambda_2^4 - 2\lambda_1^2\lambda_2^2)}{(-\lambda_1^4 - 2\lambda_1^2\lambda_2^2) + \lim_{t\to\infty}\frac{\mathscr{D}_{\lambda_2}(t)}{\mathscr{D}_{\lambda_1}(t)}(-\lambda_1^3\lambda_2 + \lambda_1^2\lambda_2^2 - 2\lambda_1\lambda_2^3) + \left(\lim_{t\to\infty}\frac{\mathscr{D}_{\lambda_2}(t)}{\mathscr{D}_{\lambda_1}(t)}\right)^2\lambda_1\lambda_2^3}.
$$

Therefore, $\lim_{t\to\infty}\frac{\mathscr{D}_{\lambda_2}(t)}{\mathscr{D}_{\lambda_1}(t)}$ is the positive real root of the second-degree polynomial

$$
\mathcal{P}(x) = \lambda_1\lambda_2(x+1)(\lambda_2 x - \lambda_1) + (\lambda_2 - \lambda_1)^3 x. \tag{97}
$$

Solving for $x > 0$, we derive the explicit formula

$$
\lim_{t\to\infty}\frac{\mathscr{D}_{\lambda_2}(t)}{\mathscr{D}_{\lambda_1}(t)}
$$
$$
= \frac{\lambda_1^3 - 2\lambda_1^2\lambda_2 + 2\lambda_1\lambda_2^2 - \lambda_2^3 + \sqrt{\lambda_1^6 - 4\lambda_1^5\lambda_2 + 8\lambda_1^4\lambda_2^2 - 6\lambda_1^3\lambda_2^3 + 8\lambda_1^2\lambda_2^4 - 4\lambda_1\lambda_2^5 + \lambda_2^6}}{2\lambda_1\lambda_2^2}. \tag{98}
$$

Given

$$
\gamma_t^{\mathrm{line}} = \frac{2\left(\lambda_1^2\mathscr{D}_{\lambda_1}(t) + \lambda_2^2\mathscr{D}_{\lambda_2}(t)\right)}{\left(\lambda_1\mathscr{D}_{\lambda_1}(t) + \lambda_2\mathscr{D}_{\lambda_2}(t)\right)\left(\lambda_1^2 + \lambda_2^2\right)} = \frac{2\left(\lambda_1^2 + \lambda_2^2\frac{\mathscr{D}_{\lambda_2}(t)}{\mathscr{D}_{\lambda_1}(t)}\right)}{\left(\lambda_1 + \lambda_2\frac{\mathscr{D}_{\lambda_2}(t)}{\mathscr{D}_{\lambda_1}(t)}\right)\left(\lambda_1^2 + \lambda_2^2\right)}, \tag{99}
$$

we have

$$
\lim_{t\to\infty}\gamma_t^{\mathrm{line}} = \frac{2\left(\lambda_1^2 + \lambda_2^2\lim_{t\to\infty}\frac{\mathscr{D}_{\lambda_2}(t)}{\mathscr{D}_{\lambda_1}(t)}\right)}{\left(\lambda_1 + \lambda_2\lim_{t\to\infty}\frac{\mathscr{D}_{\lambda_2}(t)}{\mathscr{D}_{\lambda_1}(t)}\right)\left(\lambda_1^2 + \lambda_2^2\right)}. \tag{100}
$$

By substituting (98), we get

$$
\lim_{t\to\infty}\gamma_t^{\mathrm{line}}
$$
$$
= \frac{\lambda_1^3 + 2\lambda_1^2\lambda_2 + 2\lambda_1\lambda_2^2 + \lambda_2^3 - \sqrt{\lambda_1^6 - 4\lambda_1^5\lambda_2 + 8\lambda_1^4\lambda_2^2 - 6\lambda_1^3\lambda_2^3 + 8\lambda_1^2\lambda_2^4 - 4\lambda_1\lambda_2^5 + \lambda_2^6}}{\left(\lambda_1^2 + \lambda_2^2\right)^2}. \tag{101}
$$

A direct calculation reveals that $\lambda_1 > \lambda_2 > 0$ implies $\lim_{t\to\infty}\gamma_t^{\mathrm{line}} \leq \frac{2\lambda_{\min}(K)}{\frac{1}{d}\mathrm{Tr}(K^2)}$. □

**Remark F.1.** *For the scenario where $K$ has an arbitrary number $n$ of distinct eigenvalues, equation (13) remains valid. The proof parallels the one outlined above. However, in this case, the expression for $\lim_{k\to\infty}\mathfrak{g}_k$ is given by*

$$
\lim_{k\to\infty}\mathfrak{g}_k = \frac{n\left(\lambda_1^2 + \lambda_2^2 x_1 + \cdots + \lambda_n^2 x_{n-1}\right)}{\left(\lambda_1 + \lambda_2 x_1 + \ldots \lambda_n x_{n-1}\right)\left(\lambda_1^2 + \cdots + \lambda_n^2\right)}, \tag{102}
$$

*where $x_1, \ldots, x_{n-1} > 0$ satisfy a more intricate coupled system of $n-1$ equations.*

# G  Examples

Any single index model with $\alpha$-pseudo Lipschitz ($\alpha \leq 1$) activation function is covered by our SGD+AL theory. In this section, we provide key learning problems within this family of models.

## G.1  Binary logistic regression

We consider a binary logistic regression problem with $\epsilon = 0$ where we are trying to classify two classes. We will follow a Student-Teacher model, in which there exists a true vector $X^\star$ to be the true direction such that possible labels are, $y = \frac{\exp(\langle X^\star, a \rangle)}{\exp(\langle X^\star, a \rangle)+1}$. or $1 - y$. In order to classify the data we minimize the KL-divergence between the label $y$ and our estimate defined by the above formula,

$$\mathcal{R}(X) = \mathbb{E}_a \left[ -\langle X, a \rangle \cdot \frac{\exp(\langle X^\star, a \rangle)}{\exp(\langle X^\star, a \rangle) + 1} + \log\left(\exp(\langle X, a \rangle) + 1\right) \right]. \tag{103}$$

To study the ODE dynamics of SGD in Eq. (9) one needs the deterministic risk $h(B)$, and $I(B) = \mathbb{E}_a[f'(\langle X, a \rangle, \langle X^\star, a \rangle)^2]$, with $B = W^T K W$. Following the computation in Appendix D example D.4 in [15] we obtain that

$$h(B) = -B_{21} \mathbb{E}_z \left[ \frac{\exp(\sqrt{B_{22}} \cdot z)}{(1 + \exp(\sqrt{B_{22}} \cdot z))^2} \right] + \mathbb{E}_w \left[ \log(\exp(w\sqrt{B_{11}}) + 1) \right], \tag{104}$$

where $z, w \sim \mathcal{N}(0, 1)$. The $I$ function can also be computed explicitly by solving the following Gaussian integral, where we define $g(x) \stackrel{\text{def}}{=} \frac{\exp(x)}{1+\exp(y)}$

$$I(B) = \frac{1}{2\pi\sqrt{\det(B)}} \int_{\mathbb{R}^2} (g(x) - g(y))^2 \exp\left( -\frac{1}{2} \begin{pmatrix} x \\ y \end{pmatrix}^T B^{-1} \begin{pmatrix} x \\ y \end{pmatrix} \right) \, dx \, dy. \tag{105}$$

We note that, the logistic regression is $(\mu, \theta)$–RSI with $\mu = \frac{1}{\ell e^{\sqrt{4\theta}}}$ see section 2.2 in [15]. Its Lipschitz constant is $\hat{L}(f) = 1$. Using Proposition D.1 one can derive a lower bound on the limiting learning of AdaGrad Norm.

For more details and more examples, see [15].

## G.2  CIFAR 5m

Finally, we include an example that uses real-world data, that is, the CIFAR 5m dataset [38]. Our theory does not explicitly deal with non-Gaussian distributions, but we find that the theoretical risk curves generalize cleanly to that case.

As we are now working with discrete data points rather than a distribution, the learning setup, while closely analogous to what was presented earlier, has some slight differences.

We start with a subset of the data consisting of $n$ grayscale images, each of which is $32 \times 32$ pixels, that is, $A \in \mathbb{R}^{n \times 1024}$. We fill a vector $b \in \mathbb{R}^n$ with the corresponding labels (0 for an image of a plane, 1 for an image of a car.) We then randomly choose a matrix $W \in \mathbb{R}^{1024 \times d}$ with i.i.d. Gaussian entries to generate the features $F = \text{relu}(AW)$. We want to use least squares to predict the label from the features, i.e., find

$$\arg\min_{X \in \mathbb{R}^d} \left\{ \mathcal{R}(X) := \frac{1}{2n} \|FX - b\|^2 = \frac{1}{2n} \sum_{i=1}^n (f_i \cdot X - b_i)^2 \right\}, \tag{106}$$

where $f_i$ is the $i$th row of $F$. The SGD we now consider is

$$X_{k+1} = X_k - \gamma_k \left( f_{i_{k+1}} \cdot X - b_{i_{k+1}} \right) f_{i_{k+1}}, \quad \{i_k\} \text{ iid Unif}(\{1, 2, \cdots, n\}), \tag{107}$$

where $\gamma_k$ is the usual AdaGrad-Norm stepsize, as in (15). Our empirical covariance matrix $K$ (remembering that $f_i$ is a row vector) is then

$$K = \mathbb{E}_{i \in [n], j \in [n]} \left[ f_i^\top f_j \right] = \frac{1}{n} F^\top F. \tag{108}$$

We now use (64), with the AdaGrad-Norm stepsize, to numerically simulate the SGD loss, which we then compare to the actual loss. Our theory matches empirical results very closely.

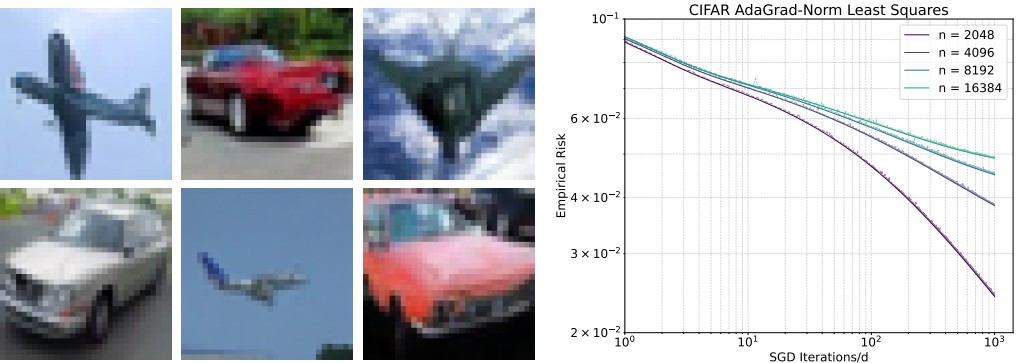

Figure 6: Predicting the training dynamics on a real dataset, CIFAR-5m [38], using multi-pass AdaGrad-Norm. This suggests the theory extends beyond Gaussian data and one-pass. Note that the curves look significantly different for different $n$; smaller values of $n$ lead to an overparametrized problem, allowing least squares to memorize datapoints, whereas for larger $n$, least squares must learn a general function mapping images of cars and airplanes to their respective labels.

## H    Numerical simulation details

Here we provide more details for the figures that appear in the main paper.

**Figure 1:    Concentration learning rate and risk for AdaGrad-Norm** on a least squares problem with label noise $\omega = 1$ (left) and on a logistic regression problem with no label noise (right). For logistic, see Section G. 30 runs of AdaGrad-Norm with parameters $b = 1$ and $\eta = 1$ for each $d$; $X^\star \sim \mathcal{N}(0, I_d/d)$, $X_0 = 0$, and $K = I_d$. The shaded region represents a 90% confidence interval for the SGD runs. As the dimension increases, the risk and stepsize both concentrate around a deterministic limit (red). The deterministic limit is described by an ODE in Theorem 2.1. The initial loss increase in the least squares problem suggesting that the learning rate was initially too high, but AdaGrad-Norm naturally adapts and still the loss converges. Our ODEs predict this behavior.

**Figure 2:    Comparison for Exact Line Search and Polyak Stepsize** on a noiseless least squares problem. The left plot illustrates the convergence of the risk function, while the right plot depicts the convergence of the quotient $\gamma_t / \frac{\lambda_{\min}(K)}{\frac{1}{d} \operatorname{Tr}(K^2)}$ for Polyak stepsize and exact line search. Both ODE theory and SGD results are presented, showing a close agreement between the two approaches. The covariance matrix $K$ is generated such that the eigenvalues follow the expression $\lambda_i(K) = \sqrt{\frac{d}{\sum_{i=1}^{d} \left(\frac{i}{d+1}\right)^{-2/s}}} \cdot \left(\frac{i}{d+1}\right)^{-1/s}$, $i = 1, \ldots, d$, where $s > 2$ is a constant. As $s$ approaches 2, the spectrum becomes more spread out, resulting in larger values of $\frac{1}{d} \operatorname{Tr}(K^2)$. Larger values of $s$ correspond to smaller spreads in the spectrum. Additionally, $\operatorname{Tr}(K)/d = 1$ for all $s$. Both plots highlight the implication of equation (13) in high-dimensional settings, where a broader spectrum of $K$ results in $\frac{\lambda_{\min}(K)}{\frac{1}{d} \operatorname{Tr}(K^2)} \ll \frac{1}{\frac{1}{d} \operatorname{Tr}(K)}$, indicating slower risk convergence and poorer performance of exact line search (unmarked) as it deviates from the Polyak stepsize (circle markers). The gray shaded region demonstrates that equation (13) is satisfied.

**Figure 3:    Quantities effecting AdaGrad-Norm learning rate.** *(left):* The effect of adding noise to the targets ($\omega = 1.0$) to the risk (left axis) and learning rate (right axis). Ran AdaGrad-Norm($b = 1.0, \eta = 2.5$) on least squares problem with $d = 500$. $X_0, X^\star \sim \mathcal{N}(0, I_d/d)$. A single run of the SGD (solid line purple) matches exactly the prediction (ODE, teal). The shaded region represents 10 runs of SGD with 90% confidence interval. The learning rate decays at the exact predicted rate of $\frac{\eta}{\sqrt{b^2 + \frac{\operatorname{Tr} K \omega^2}{d} t}}$. Depicted is $\frac{\text{learning rate}}{\text{asymptotic}}$ so it approaches 1. *(center, right)*: Noiseless least squares setting ($\omega = 0$). *(center)*: Prop. 4.2 predicts the avg. eig of $K$ ($\operatorname{Tr}(K)/d$) as compared with $\lambda_{\max}$ effects the $\lim_{k \to \infty} \mathfrak{g}_k$. Indeed, this is true. We varied the $\kappa = \lambda_{\max}/\lambda_{\min}$ while keeping the $\operatorname{Tr}(K)/d$ and all other parameters fixed. All the learning rates behave identically verifying our

theory about the effect of $\text{Tr}(K)/d$ on learning rates. *(right)*: Varying the learning rate of AdaGrad norm by $\|X_0 - X^\star\|^2$; our predictions (dashed) match and we see the inverse relationship predicted by Prop. 4.2. See Appendix D for details. Additionally, we did the following.

- **Center plot**: AdaGrad with $b = 0.5$, $\eta = 2.5$ is run on the least squares problem with $d = 1000$ and $X_0, X^\star \sim \frac{1}{\sqrt{d}}\mathcal{N}(0, I)$. The covariance matrix $K$ is generated so that the eigenvalues are

$$\lambda_i(K) = \sqrt{\frac{d}{\sum_{i=1}^{d}\left(\frac{i}{d+1}\right)^{-2/s}} \cdot \left(\frac{i}{d+1}\right)^{-1/s}}, \quad i = 1, \ldots, d.$$

  The constant $s > 2$. When $s$ is near 2, the spectrum is more spread out, i.e., $\kappa = \frac{\lambda_{\max}}{\lambda_{\min}}$ is large. Larger values of $s$ mean smaller the spreads. Moreover $\text{Tr}(K)/d = 1$ for all $s$. In the simulations, we used $s \in \{2.1, 3.0, 3.5, 4.0, 5.5\}$ and recorded the condition number $\kappa$.

- **Right plot**: Ran AdaGrad with $b = 0.5$, $\eta = 2.5$ on the least squares problem with $d = 1000$. $X^\star = 0$ and $X_0 \sim \sqrt{\frac{p}{d}}\mathcal{N}(0, I)$ where $p \in \{1, 2, 4, 8, 16\}$. In this way, $\|X_0 - X^\star\|^2 = p$.

**Figure 4:** **Power law covariance in AdaGrad Norm** on a least squares problem. Generated covariance $K$ such that the density of eigenvalues are $(1 - \beta)\lambda^{-\beta}$ where $\beta = 0.2$ and set $X_0 = 0$. Choose $(X_i^\star)_{i=1}^d = (\lambda_i^{-\delta/2})_{i=1}^d$ where $\lambda_i$ is the $i$-th eigenvalue of $K$ and we vary $\delta \in (0, 1.8)$ so that $0 < \delta + \beta \leq 2$. Setting of Prop. 4.4.

**Figure 5:** **Convergence in Exact Line Search** on a noiseless least squares problem. The plot on the left illustrates the convergence of the risk function, while the center and right plots depict the convergence of the quotient $\frac{\mathscr{D}_{\lambda_2}(t)}{\mathscr{D}_{\lambda_1}(t)}$ and the learning rate $\gamma_t$, respectively. Predictions from ODE theory are compared with results obtained from SGD, demonstrating close agreement between the two approaches. Initialization was performed randomly, with $X_0 \sim \mathcal{N}(0, I_d/d)$ and $X^\star \sim \frac{1}{\sqrt{d}}\mathbf{1}$, where $d = 400$. The covariance matrix $K$ has two distinct eigenvalues $\lambda_1 = 1 > \lambda_2 > 0$, and was constructed by specifying the spectrum, with $\lambda_i$ sampled from a discrete uniform distribution $\mathcal{U}\{1, \lambda_2\}$ for $i = 1, \ldots, d = 400$, and setting $K = \text{diag}(\lambda_i : i = 1, \ldots, 400)$. Further details and formulas for the limiting behavior can be found in the Appendix F.2.

**Figure 6** **Convergence on CIFAR 5m [38].** We train a classifier to distinguish between images of airplanes and cars. Fix $d = 2000$. Then for multiple values of $n$, we run AdaGrad-Norm with initialization $X_0 = 0$, $b = 0.1$ and $\eta = 5$, randomly sampling a datapoint from $F$ at every step. Details of the setup can be found in Appendix G.2.

