# OpenReview forum: "The High Line: Exact Risk and Learning Rate Curves of Stochastic Adaptive Learning Rate Algorithms"
_NeurIPS.cc/2024/Conference — NeurIPS 2024 poster_

### Official Review · Reviewer_Qoif · 2024-07-03

**Soundness:** 3
**Presentation:** 3
**Contribution:** 4
**Rating:** 7
**Confidence:** 4

**Summary:**

The paper is a continuation of a broad line of work exploring deterministic limits of stochastic gradient descent. In particular, the author(s) follow one specific branch of the many, which derives a solution to a coupled integro-differential equation by passing on the complex space. The novelty lies mainly in the fact that they derive, for least squares, deterministic limits of dynamics that have adaptive step-sizes, for non identity covariance of the observed dataset. Industrially, this is important as the idealized scenario of fixed step-sizes is __very much__ idealized, and realistic datasets have structure. Thanks to the general expressions (under 6 assumptions that are claimed to be not super-restrictive), they can proceed to study the various differences between standard schedules. The limiting learning rate can be derived, as well as the scaling of the risk along time. This is done for Line Search, Polyak step-size, AdaGrad Norm. Interestingly, their bounds suggest that for covariances with largely separated eigenvalues (say, power law spectrum), there should be a change in the phenomenology (i.e. a phase transition). They verify this for a power law covariance, which exhibits three phases depending on the parameters of the problem (notably the starting distance from $X^{\star}$). All is accompanied by experiments corroborating the predictions.

**Strengths:**

The motivation in lines 17-31 is very well stated. As a reader, I would be inclined to continue parsing the paper.
- Proposition 4.3 states that with a null initialization $X_0 = 0$ and subject to a condition on the covariance (which roughly says that $\lambda_{\min}$ can be small but that there is a very small number of vanishing eigenvalues), if the ground truth signal is spread out, then the learning rate is never zero. For symmetric activations (say, phase retrieval), and constant learning rate, I think you cannot really start at zero because you have a symmetry, you need at least a tiny bit of overlap. From this result, it appears that you can start from the most useless guess and still get away with it, provably. Interesting.
- The higher arching aim of the work is very important, as knowing that the behavior of SGD is predictable is pivotal for understanding how our models work.
- The techniques are of independent interest and are a continuation of a line of work that is systematically, step by step, developing a global answer to these questions.
- The result that Polyak's step-size converges to the inverse of the average eigenvalue is yet another validation of a method originated from convex optimization holding in more generality. This is reassuring.

**Weaknesses:**

Weaknesses are questions and questions are weakenesses in some sense. Once the author(s) engage, I will reorder things that get answered and things that remain as "concerns".

- Proposition 4.4 supposes that $\mathscr{D}_i^2(0)$ could potentially have a different scaling at each site $i$. I understand that the assumption includes an uninformative initialization $\delta = 0$. However, when is it really that one can think of a setting in which you can have an initialization that scales like $\lambda_i^{-\delta}$ for non trivial $\delta$? The result is very aesthetic, but I did not think through of its realistic value. Anyways, it is very nice so I am __not__ criticizing the math. Maybe I am just interested in understanding if you have in mind something.
- In the Checklist, you state that you have been "__very careful__ \[...\] to include __all__ the assumptions needed". I agree with this statement. However, as a reader, I do not agree with this being the limitation of your work, as in some sense you claim over the whole text that your assumptions cover standard adaptive step-sizes (hence you hiddenly claim that this theory is all we need). I would be very happy to see a paragraph where you clearly state your limitations. Where does your theory fail? Where does the technique __not__ extend naturally? If you want to sell me your paper, I do not believe its value is in the assumptions. On a first aspect, the ideas and the techniques are of independent interest. Secondly, it would be sad if the bottleneck were the assumptions.
- Your equation (7) is found across some other works. One of the earliest appearances is (Yoshida and Okada, 2019). However, from a first check, I would guess that you are missing a term on the first object. To be fair, I quickly tried to re-derive it and I would guess that you need a further $H_{2, t}\mathscr{V}_{12, i}(t)$ in the first expression. Am I wrong? If so, please correct me. It looks like the equations are missing some symmetry of the dynamics.
- I am concerned about the scaling. My understanding of the literature is that scalings are very important. Your assumptions require that $\lVert K\rVert$ is bounded independently of the dimension and that $\lVert X_0\rVert$ is bounded, independently of the dimension. Moreover, your step-size is $\frac{1}{d}\mathfrak{g}_k$. Therefore, at the beginning you move by very little, I think by $\propto \frac{1}{d}$. What is done in some other references is very different. Let me take an example, and correct me if I am wrong.
1. You can derive heuristically deterministic dynamics for step-size $\mathfrak{g}_k$ (not normalized), starting from a standard Gaussian standard gaussian vector $X_0$, with data-points being i.i.d. Gaussians $a\sim \mathcal{N}\left(0,\frac{1}{d}I_d\right)$. Now, the step-size is not divided by dimension, the $X_0\sim \mathcal{N}(0, I_d)$ is not bounded in norm by something independent of $d$, and the $a$ matches your assumptions. In this case however, the signal is far stronger, as we are removing two normalizations you indeed have. Below my questions regarding this.
2. Can you clarify why it makes sense to choose your scaling? Apart from it allowing the proofs.
3. In particular, how is it not possible to allow for norms dependent on $d$, and in parallel, how is it that the signal is enough to be not stuck at the initialization?
4. Does this implicitly mean that you are exploring a slow regime? Say, a Gradient Flow? What happens if in your experiments you just change the scalings to something larger in magnitude?


#### Typos
__NOTE__: I am including for completeness typos in "weaknesses". Please do not count them as such.
- (line 560) In the reminder of $\Omega$, you are missing a $\max$.
- (equation 20) there is no $\Gamma(t)$ in the expression, right?
- (lines 1140-1141) the way of quoting "noise" and "variance" is not the correct one, TeX-wise.

**Questions:**

Please note that some of the questions below are very ingenuous.

- (lines 49-57) This is more of an impression. Why is $\lambda_{\min}(K) > C > 0$ for you a situation of strong anisotropy? The power-law case is anisotropic in that the minimum eigenvalue is not bounded below by a constant as $d\to\infty$. Put another way, I do not understand how the __minumum__ eigenvalue being bounded below from zero would describe the whole behavior of the covariance profile of the data-points $a$. I understand that this might be related to the fact that the optimal strategy uses the trace so you say something in the line of "if the minimum eigenvalue is very well distant from the average then there is anisotropy". At the very least, it appeared to me as a reader that you somehow __define__ strong anisotropy as this eigenvalue condition, while in fact it is a consequence of the algorithmic behavior that you can tune the eigenvalues in this "anisotropic" way and observe that "greed can be arbitrarily bad".
- The original introduction of the Polyak step-size is that of a step-size that uses the optimal value of the function, and the value of the current iterate. In your case, what you call Polyak step-size has a connection with the original method, which is related to your assumptions, as you find that eventually it gets to a $\gamma_{\infty}$ that takes care of the geometry of the function. From a pedagogical point of view, it would be nice to comment further on this.
- Theorem 2.1 states that the concentration is for any $\epsilon \in \left(0,\frac{1}{2}\right)$, line 48 says "for any inverse power of d", and Theorem B.1, Corollary B.1  cited below Theorem 3.1 in line 195 give a result for existance of an $\epsilon$ such that the approximate solution of the integro-differential equation is verified. Can you shed light on these differences? Since it is your main Theorem, it would be nice to have an explicit discussion of what goes on where. More in detail, why do you get from existance in the appendix to a whole interval of $\epsilon$ (that by the way gives a rather slow decay wrt to say $\epsilon > 1$), to claim in text "for any inverse power"? I have to be honest here and I did not spend time re-doing all the proofs.
- Related to the point above, in equation (27), your supremum is over $T\wedge \hat{\tau}_M$. In line 531 you state that you will not specify which stopping time it is when it is "clear from context", but for (27), given that you want to prove (26), my natural question is if stability is for $\hat{\tau}_M(S)$ or $\hat{\tau}_M(\mathscr{S})$, as you only mention one. At the very least, it is not clear from context here which stopping time it is in (27).
- Lemma C.1 requires as condition that $S(t, z)$ is an approximate solution over the $\xi$-mesh. How do you guarantee that this condition holds in your case? If ever, it is not evident from text that this is the case. If it is a direct consequence of arguments in reference 14, I would appreciate it being made explicit.
- Can you clarify the sentence of lines 239-240? Ignore the "sufficiently anisotropic" aspect which I already discussed above. The idea that smaller learning rates lead to under-performance is not entirely true. In the simplest setting if you over-shoot with your learning rate you will never converge. I believe here this is tightly linked to your Table 1, but there line search and Polyak have the same convergence rate for the risk. Where is the difference?
- In Figure 1, how many runs do you make? It would be nice to know.
- The same question above but for any other plot where you show error bars.
- (line 1125) You choose as scaling for the initialization $X_0\sim\mathcal{N}\left(0,\frac{I_d}{\sqrt{d}}\right)$  and the ground truth $X^{\star} = \frac{1}{\sqrt{d}}\mathbf{1}_d$. The norm of the latter is $1$, but the former has norm concentrating at $\sqrt{\sqrt{d}}$, and this violates your assumptions (not bounded by a constant independent of $d$). Is it a typo? Or is it that the experiment does not match the assumptions? If I look at the scaling of Figure 3, line 1099, you indeed sample $X_0\sim \mathcal{N}\left(0,\frac{I_d}{d}\right)$.
- Proposition 4.4 is nice. What happens when $\beta +\delta > 2$?

My soundness score is due to the many questions. I hope I will raise it after the discussion. The overall score follows the same principle.

**Limitations:**

None.

---

> ### Author Rebuttal · Authors · 2024-08-06
>
> We thank the reviewer for their positive review and careful reading of our paper, including the appendix. We address below all of the reviewer’s comments. *Due to space limitations, we will include an additional "Official Comment" to answer all the reviewer's questions.*
>
> **Practical application of power law scaling on $\\mathscr{D}\_i^2(0)$.** First, it is important to note that this power law scaling is not on the initialization but rather on the distance from the initialization to the ground truth (signal) $X^\\star$.  If one initializes at 0, which is reasonable to do in the least squares setting, then this is a power law scaling on the ground truth.  In settings where the data has power law covariance, one can also see power law scaling in the ground truth. For more details on this, see, for example, reference \[56\] mentioned in the paper, particularly their discussion of scaling laws in section 6\.
>
> **Limitations of the paper.** Added a “Limitations” paragraph. For the version going in the main paper, see “All Reviewers Rebuttal”.
>
> **Coupled ODEs in Eq (7).** Thank you for pointing out the potential issue with Equation (7). You are correct that there was an oversight in the first term of Eq(7).  We have revised the equation to include the missing term and corrected ODEs. Page 17 contains the correct ODEs for S and reflects the updated formulation. We appreciate your attention to detail and the opportunity to correct this error.
>
> **Scalings.** Thank you for the questions.  Please see the discussion for all reviewers, where we discuss other scalings beside $(1/d)$.
>
> You also asked about whether a learning rate on the scale of $1/d$ is large enough for the algorithm to really move away from initialization.  The key here is that we work in the regime where the number of iterations is proportional to dimension.  Thus, even with learning rate of size $1/d$, one sees significant movement after order $d$ iterations.  Indeed, for examples such as linear or logistic regression, this learning rate yields convergence of the risk in order $d$ iterations.
>
> You are correct that, in certain cases (e.g. phase retrieval with a cold start), the algorithm can become stuck at initialization and require more than order $d$ iterations to converge.  This is also captured in our result.  The theorem is not a statement about the convergence of the algorithm to an optimal solution, but rather about the convergence of the SGD dynamics to a deterministic curve.  Thus, our theorem can accurately predict the risk trajectory for order $d$ iterations, even when the risk does not converge to zero in that time.  We’ll add that to capture the escape of phase retrieval from the high-dimensional saddle $O(d \log d)$ iterations are needed, and moreover the high-dimensional limit of the risk curve will contain non-concentrating and non-vanishing effects of SGD noise (this is implicit in Tan and Vershynin. *Online Stochastic Gradient Descent with Arbitrary Initialization Solves Non-smooth, Non-convex Phase Retrieval. 2019*).
>
>
> Finally, you asked about our choice to bound the norm of $X$, independent of $d$.  Because we are interested in attaining a high-dimensional limit of the dynamics, we need to scale things in such a way that the risk is dimension-independent.  In our set-up the risk is always expressible as a (generally non-linear) function of the inner product $X^Ta$, so we need this inner product to be dimension-independent.  Our choice of scaling achieves this (although one could also, for example, simultaneously adjust the scaling of $X$ and $a$ to achieve an equivalent result).  There are, as you suggest, interesting problems in which the norm of $X$ may grow with $d$, but those are beyond the scope of this paper.
>
> **Typos.** Thank you for pointing out these things.  *We corrected them.*

---

> > ### Author Response · Authors · 2024-08-07
> > **(Response to questions)**
> >
> > **Answers to the reviewers’ questions**
> >
> > * **Strong anisotropy.** When we say “strong anisotropy” we are not referring to the lower bound on the eigenvalues.  Rather, we refer to context where, for fixed $\\lambda\_{\\min}$, we have $\\lambda\_{\\min}/(\\text{Tr}(K^2)/d)\\ll (\\text{Tr}(K)/d)^{-1}$ or in other words, $\\text{Tr}(K)/d\\ll\\text{Tr}(K^2)/d$.  This is in contrast to the isotropic case where $\\text{Tr}(K)/d=\\text{Tr}(K^2)/d$.  *We can make this more clear by writing $\\text{Tr}(K)/d\\ll\\text{Tr}(K^2)/d$ in parentheses after the phrase “strong anisotropy.”*
> >
> > * **Polyak stepsize.** Indeed, we are using the derivation of Polyak stepsize (see, e.g.,  \[1\]) as motivation to construct what should be the “idealized” stochastic version and not the $( R(X\_k)-R(X^\*) ) / ||\\nabla R(X\_k)||^2$. We will add commentary about the derivation of the Polyak step size in the paper and how it deviates from the classical Polyak stepsize. We appreciate the reviewer pointing this out.
> >
> >   \[1\] Elad Hazan and Sham Kakade. “Revisiting the Polyak Step Size”.
> >
> > * **Concentration as an inverse power of d.** The concentration for the distance from SGD statistics to their deterministic equivalent is indeed $d^{-\\varepsilon}$ for $0\<\\varepsilon\<1/2$, so the best bound we obtain for this concentration is slightly worse than $1/\\sqrt{d}$.  The comment in line 48 is that the concentration occurs with **probability** better than any inverse power of d.  In other words, the probability that our statistic will be less than $d^{-\\varepsilon}$ is at least $1-d^{-C}$ for arbitrarily large C.  In the theorem, we describe this more succinctly in the phrase “with overwhelming probability.”  However, the definition of overwhelming probability is not introduced until line 74, so we do not use it in the comment on line 48\.
> >
> > * **Stopping times.** This is indeed unclear, and thank you for pointing it out.  It is meant to be the minimum of the two stopping times, as in the preceding equation and in the Proposition that is cited from \[14\].  *We can clarify this in the updated version of the paper.*
> >
> > * **Verifying $S(t,z)$ is an approximate solution.** Indeed, this is a very important condition to verify, and doing so is the primary purpose of Section C.2.  The way in which this is utilized with regard to the mesh is briefly described in Section C.1.3 where we prove Proposition C.1 (pending the details that are in Section C.2).  *Based on your question, we can add a sentence at the location that you referenced summarizing the flow of the proof and where these details can be found.*

---

> ### Comment · Reviewer_Qoif · 2024-08-09
>
> Acknowledged, see full comment below. I believe you are missing the last three questions (one might be a typo therefore I ping you about this).

---

> > ### Author Response · Authors · 2024-08-09
> >
> > Oops! Yes, we forgot to add the responses to the last questions.  Sorry!
> >
> > * **Comparison of Line search and Polyak risk convergence rate.** As you have observed, Table 1 indicates that the convergence rates for the risk in line search and Polyak have the same formula as a function of the limiting learning rate, $\\gamma\_{\\infty}$.  The key difference is in the value of $\\gamma\_{\\infty}$ in these two cases.  For data that is strongly anisotropic in the sense described above, the value of $\\gamma\_{\\infty}$ can be much smaller for line search than for Polyak, thus yielding slower convergence.
> >
> > * **Number of runs for simulations.** The number of runs for the simulations, as well as other implementation details, are provided in Appendix H. Due to space constraints, we could not include these details in the main body of the paper.
> >
> > * **Scaling of initialization in line 1125\.** This is indeed a typo, and thank you for pointing it out.  *We corrected it.*
> >
> > * **What happens when $\\beta+\\delta\>2$.** This is a natural question and thank you for asking.  We do not address the case of $\\beta+\\delta\>2$ because it represents a setting in which the high-dimensional problem devolves to a finite-dimensional one where our scalings no longer really make sense.  This is perhaps easiest to see in our formulation of the power law set-up for the covariance, where we take the spectrum of $K$ to approach a continuous density function, supported on $(0,1)$, with unbounded density near 0\.  This density is only well-defined for $0\<\\beta\<1$.  Similarly, one can view the power-law set-up for $\\mathscr{D}^2\_i(0)$ giving the distribution of the projections of $(X\_0-X^{\\star})^2$ in the eigenvector directions.  This distribution approaches a well-defined, continuous density function when $0\<\\delta\<1$.  Thus, the limits that we consider cease to make sense when $\\beta+\\delta\>2$.
> >
> >   One can also see that the proofs break down when $\\beta+\\delta\>2$.  Our analysis of Adagrad-Norm in the power-law setting (see Section D.2.3) relies upon the fact that the deterministic limit of the risk can be expressed as a convolution Volterra equation (see (67)).  Under the assumption that the forcing function F has power law decay (see Assumption 8), we can derive the asymptotics of the learning rate and the risk for various regimes, as displayed in Table 1\.  Assumption 8 is satisfied by our power-law set-up, provided that $\\beta+\\delta\<2$ (see Lemma D.5).

---

> ### Comment · Reviewer_Qoif · 2024-08-09
> **General Response**
>
> Dear author(s),
> thank you for your general rebuttal and for the rebuttal concerning my specific comments. I understand all your points, and after the corrections/specifications you put, I will raise my score. Some quick points:
> - strong anisotropy and verifying the approximate solution: you comments in italics are necessary in my opinion.
> - Polyak comment: yes, please.
> - scalings comment: see one of the questions missing (at the very end) in my long list to close this aspect.
> - all other typos: thank you for fixing them.
>
>
>  last comment: This is a very deep paper, maybe a journal deserves it.
>
> My score is higher. I hope the other reviewers will raise their very low grades. Some scores, in their very description, are too harsh with respect to the actual review.
>
> My score reflects my opinion on the exact description provided by Neurips: moderate-to-high impact. I stand with the idea that this paper is definitely not a 3.
>
> Good luck!!

---

> > ### Author Response · Authors · 2024-08-09
> > **Regarding gradient flow, different scalings**
> >
> > Dear reviewer,
> >
> > We want to address the question you re-raised, which we didn't completely answer:
> >
> > > I am concerned about the scaling. My understanding of the literature is that scalings are very important. Your assumptions require that is bounded independently of the dimension and that is bounded, independently of the dimension. Moreover, your step-size is ...
> >
> > > What is done in some other references is very different. Let me take an example, and correct me if I am wrong.
> >
> > Let's take the example you posed.  The steps of SGD look like
> >
> > $$ X_{k+1} = X_k - \eta a_{k+1} f'\left( \langle a_{k+1}, X_k \rangle \right) $$
> >
> > Now you put $a_{k+1}$ to be scaled to have norm $O(1)$ and $X_{0}$ to have norm $O(\sqrt{d}))$.  This is not our setup, but it is possible to change variables to put it in our setup.
> >
> > Let $Y_{k} = X_{k}/\sqrt{d}$ and let $b = a \sqrt{d}$.  Now in these variables, the equation above becomes
> >
> > $$ Y_{k+1} = Y_k - \frac{\eta}{d} b_{k+1}  f'\left( \langle b_{k+1}, Y_k \rangle \right) $$
> >
> > So indeed, in the setup that you proposed, it would be correct to take $\eta  = O(1)$, and it can be equivalently represented in our setup with stepsize scaling like $O(1/d)$.  What's important in both cases is the scale of the inner product $\langle b_{k+1}, Y_k \rangle$ is $O(1)$.
> >
> > So our setup is not gradient flow, and the (not-degenerate) high-dimensional limit for the scaling you propose is the same as ours.
> >
> > By the way, we'd be happy to look at any literature involving the scalings that you propose, so that we can make a better comparison with existing literature.
> >
> > Please let us know if this answers all your questions regarding this setup.
> > Thanks!!

---

### Official Review · Reviewer_A5Ho · 2024-07-08

**Soundness:** 4
**Presentation:** 2
**Contribution:** 2
**Rating:** 7
**Confidence:** 2

**Summary:**

Analyzes the behavior of AdaGrad-Norm, Line-search and Polyak step size using ODEs.

**Strengths:**

This paper provides an interesting characterization of the behavior of several optimization methods using ODE tools, which are not as heavily used in optimization theory as they should be. This work seems technically sound, although I didn't review the (long) Appendix.

I am in favor of acceptance, although with reservations. The methods studied are interesting to theoreticians, but they are rarely if ever used in practice, which limits the practical appeal of this work. Especially the idealized special cases for the line search and Polyak step size. The focus on quadratic problems is also a major limitation, especially since the analyzed methods are not typically used on quadratic problems.

There is definitely potential for followup work to build on this work.

**Weaknesses:**

- Paper spends a lot of time setting up a general framework, which it then applies to specific settings. The amount of notation introduced seems over the top for a conference paper. For instance, the results for least-squares can be presented without establishing the full general framework, which could be left to the Appendix. Section 1.2 and 2 are just .... a lot to get through.
- Paper ends very abruptly.
- Some plot fonts are too small to read when printed.
- Further discussion of the implications and potential applications would be good.

**Questions:**

See above.

**Limitations:**

See above

---

> ### Author Rebuttal · Authors · 2024-08-06
>
> We thank the reviewer for their support of our paper. We answer some of the reviewer’s reservations below.
>
> **“The methods studied are interesting to theoreticians, but they are rarely if ever used in practice, which limits the practical appeal of this work. Especially the idealized special cases for the line search and Polyak step size. “**
>
> We agree with the reviewer that the algorithms studied are idealized and/or not widely used in practice. However, traditional analysis of adaptive stochastic learning rate algorithms (e.g., idealized line search/Polyak, etc.) all give the same rates as the ones used in practice. The traditional theory does not distinguish between “good” practical algorithms and “bad” practical algorithms. This suggests we need a new type of analysis. Part of our goal was to provide such a new framework. While we only analyze “idealized algorithms”, we now understand problems for which these “idealized algorithms” do not perform well. This hopefully illuminates and begins closing the gap between practice and theory.
>
> In future work, we want to derive the dynamics of more practical algorithms. We strongly believe this framework might explain theoretically why certain often used algorithms in practice perform better.
>
> **Focus on quadratic problems is a major limitation, especially since the analyzed methods are not typically used on quadratic problems.**
>
> We agree with the reviewer that “quadratic problems” are idealized, and we want in future work to extend the analysis of these algorithms to more complex losses. On the other hand, many of the algorithms were designed by approximating the “complex” loss with a quadratic. Moreover, it is surprising that we do not even know how these algorithms perform on these quadratics. In our opinion, we should understand this simple loss in detail first.
>
> This said, our analysis extends beyond quadratics to include generalized linear models such as logistic regression and more. Moreover, we do provide some analysis for other losses beyond quadratics. For strongly convex problems (not necessarily quadratic) we find a bound on the limiting learning rate, see Section D1. Also, Figure 1 shows our predictions matching the iterates of SGD with AdaGrad Norm for binary logistic regression.
>
> **Response to Weaknesses**
>
> * **Too much set-up in the main paper.** Thank you for your suggestions. In the updated version of the paper, we plan to lighten the notation and move the least square results upfront; see our reply to all reviewers.
> * **Plot fonts too small.** Thank you for pointing that out, in the next version of the paper we will update the figures and make the font larger. See the attached PDF with the revised figures.
> * **Paper ends abruptly.** Added a “Conclusion” paragraph. See comment to “All Reviewers” for the specific paragraph.
> * **Discussion of implications and potential applications.** Added a “Limitations” paragraph. The conclusion paragraph contains potential applications. See comment to “All Reviewers” for the specific paragraph.

---

### Official Review · Reviewer_hf9X · 2024-07-14

**Soundness:** 3
**Presentation:** 1
**Contribution:** 3
**Rating:** 5
**Confidence:** 3

**Summary:**

### Update after rebuttal
After reading the feedback carefully, I updated my score for the paper.

### Original review

In this work, the authors study SGD and its adaptive variants in the setting of noisy generalized linear models. Assuming the covariance matrix of the data is bounded by a dimension-independent constant and a few more similar assumptions, the authors establish that the expected risk of SGD and the expectation of adaptive learning rates both converge to deterministic curves described by an ODE.

I found the paper rather hard to follow and I couldn't understand the significance of the obtained results. The theory requires a lot of simplifications: the prediction model is linear; the learning rates that are studied are "idealized", so they are not the same as those used in practice; Adagrad-norm isn't really a method popular in practice; the ODE solution is not trivial in the case of least squares, as equation (11) is implicit, so it requires further assumptions to make a statement about the solution. I did not understand what insight we can draw from all of this.

**Strengths:**

1. The theory reveals that line search leads to slower convergence that Polyak stepsize.
2. In some simplified cases, there exist exact expressions that describe the dynamic of the studied methods.
3. The theory is also supported by numerical simulations.

**Weaknesses:**

1. The theory for line search and Polyak stepsize is only for the idealized version.
2. The insight that line search methods do not work particularly well is not exactly new, see for instance J Bolte, E Pauwels "Curiosities and counterexamples in smooth convex optimization"
3. The complexity of the theory and the discussion is such that it is very difficult to draw any insight, and it seems the theory wouldn't be extended by others because of how convoluted it is.

### Minor issues
Abbreviation "SGD" is used before (line 23) it is introduced (line 30)
"We fully expect"  -> "We expect"
"worst case convergence guarantees" -> "worst-case convergence guarantees"
"strongly-convex" -> "strongly convex"

**Questions:**

1. What's the intuition for why $h$ exists and why would it be well-behaved? I can see a discussion in Appendix B, but it doesn't provide any high-level intuition.
2. Why do you say in Appendix B that $R(X)$ is "an expectation of a Gaussian vector"?
3. The authors wrote "We shall remove this condition by approximation in what follows", where is it done?
4. The authors say they need "mild assumptions on the learning rate", but why are they mild?

**Limitations:**

The authors could do a better job when discussing the limitations of their work. There is no explicit limitations section and in the ckecklist, the authors state that they already did everything needed by clearly outlining the required assumptions. However, I think it'd benefit the paper if the authors made an explicit statement on how some aspects of the theory are too restrictive.

---

> ### Author Rebuttal · Authors · 2024-08-06
>
> The reviewer's comments required more space than 6000 characters to answer. *We will include an additional "Official Comment" with the answers to the reviewer's questions.*
>
> **Significance of results \- Main contributions**
>
> This work introduces a framework that goes beyond traditional analysis of stochastic algorithms to utilize the high-dimensionality in the problem. It allows for a finer analysis of stochastic learning rate algorithms (e.g., AdaGrad, line search, Adam, RMSprop, etc).
>
> We also emphasize that one of the main contributions of this work is an explicit deterministic expression for the dynamics of stochastic adaptive learning rate algorithms (Thm 2.1). That is, we can predict the behavior of the loss value and learning rate (both stochastic) at any iteration without ever running the stochastic algorithm (see Fig 1). We then analyze these dynamics to gain insights into the evolution of the stochastic learning rate at any iteration. These dynamics exactly agree when the problem is “high-dimensional” (number of parameters is large) and often numerically reproduce the dynamics on real data (see Fig. 6 on CIFAR-5m dataset).
>
> **Line search and Polyak only “idealized version”**
>
> It is true that the algorithms presented are idealized versions. However, we emphasize that a theoretical understanding of the behaviors of these idealized (stochastic) algorithms is not known \[Note: the reference given is for a line search on a deterministic objective and less is known about the behavior of stochastic line search\]. Adding more practical versions, which are often more complex, only adds complications to the already unknown behaviors. As such, a natural starting point for developing a theory for the exact dynamics of stochastic learning rates is to understand the behavior of these idealized versions.
>
> **Line search methods do not work well is not new**
>
> We first politely disagree with the reviewer that line searches do not work in practice. For deterministic optimization problems, line search methods are widely used and are the default in many well known algorithms (e.g., see scipy documentation for BFGS (Armijo condition)). While we appreciate the reference that showed an example where line searches fail on a deterministic optimization problem, line searches do provably work for a large class of deterministic optimization, including the deterministic versions of the problems analyzed in this paper. The theory for deterministic line searches is well developed, but for stochastic optimization problems the theory is still in its infancy.
>
> We do agree with the reviewer that on stochastic optimization problems, line search methods are not used in practice. In fact, one of the goals of this paper is to explain this\! We show why they perform badly due to anisotropic data; in contrast, a standard minimax-optimal analysis says they are as good as SGD (see e.g. \[50\]).  Moreover, we precisely quantify how different the convergence rates of a stochastic line search method will be from a tuned fixed learning rate SGD algorithm.
>
> The theoretical framework established in this paper (Theorem 2.1) provides the tools (albeit more complex than the “textbook analysis”) to do a finer analysis of stochastic line search methods.
>
> **Complexity of the theory is difficult**
>
> The standard tools in textbooks for theoretically analyzing stochastic algorithms often give too crude estimates on their behavior in practice. As an example, the standard analysis would say that you need to use a decreasing learning rate with SGD, but nobody does this in practice. The point of this paper is to provide another tool that allows for a finer analysis of these algorithms – traditional minimax optimality *isn’t capable of distinguishing these algorithms* – so something more complicated *is necessary*.
>
> **Complexity of theory makes it hard to derive insights. Theory will not be extended by others.**
>
> We politely disagree with the reviewer. The related work paragraph (Page 4 Lines 106-110) describes past works which use a high-dimensional framework for analyzing dynamics of stochastic algorithms. These past works usually assume isotropic Gaussian data ($N(0,I)$) and do not consider stochastic learning rates for their algorithm, but are part of a growing body of literature based on this style of analysis.
>
> We know that we are doing something quite different than the vast majority of optimization researchers, but we think there is a need for better ways to distinguish performance of stochastic algorithms.  For example, the more traditional analysis of line search in Vaswani et al.  \[50\] suggests SGD \+ line-search has no performance cost (their Theorems 1 and 4). We quantify the performance cost, as one incurs an extra factor of condition-number. This is a statement 99% of optimization-researchers should be able to understand, even if the methods are not to their taste, and one *needs* a more complicated setup to see it\!
>
> \[50\] Vaswani, Mishkin, Laradji, Schmidt, Gidel, Lacoste-Julien. Painless Stochastic Gradient: Interpolation, Line-Search, and Convergence Rates. 2019\.
>
> **Limitation paragraph.** Please see the response to "All Reviewers" for the exact paragraph we will add to the main document.

---

> > ### Author Response · Authors · 2024-08-07
> > **(Response to questions)**
> >
> > We thank the reviewer for finding typos, which have now been fixed. We answer below the reviewer's additional questions.
> >
> > **Intuition and existence of $h$ and why is $R(X)$ is “an expectation of a Gaussian vector”**
> >
> > The intuition for $h$ comes from the fact that $a^TX$ and $a^TX^{\\star}$ are correlated Gaussians with covariance given by $\[X|X^{\\star}\]^T K \[X|X^{\\star}\]$. Here $[X|X^{\\star}]$ means one concatenates $X$ and $X^{\\star}$.
> >
> >  Since $a \\sim N(0,K)$, for any fixed vector $X$, then $y \= a^TX \\sim N(0, X^TKX)$. Both $y \= a^T X$ and $y^{\\star} \= a^TX^{\\star}$ are Gaussians, but they are correlated normal since they both depend on the same $a$. A simple computation shows that the correlation between $y$ and $y^{\\star}$ is given by $\[X |X^{\\star}\]^T K \[X | X^{\\star}\]$ where $\[X|X^{\\star}\]$ is the concatenated vector. This is to say that $(y, y^{\\star})$ is Gaussian and $(y, y^{\\star}) \\sim N(0, \[X|X^{\\star}\]^T K \[X|X^{\\star}\]$. (See Section 1.1).
> >
> > Suppose $\epsilon =0$ as otherwise this will just add another integral. From Eq (1), using the density of multivariate Gaussian $(N(0, \[X|X^{\\star}\]^T K \[X|X^{\\star}\])$,
> > $$
> > R(X) \= \\int f(y,y^{\\star}, \\epsilon) e^{(y, y^{\\star})^T G (y, y^{\\star})} \\, dy dy^{\\star}
> > $$
> > where G is the 2 x 2 matrix, $\[X|X^{\\star}\]^T K \[X|X^{\\star}\]$. Integrating out $y$ and $y^{\\star}$,
> > $\\int f(y,y^{\\star}, \\epsilon) e^{(y, y^{\\star})^T G (y, y^{\\star})} \\, dy dy^{\\star}$ is a function of $\[X|X^{\\star}\]^T K \[X|X^{\\star}\]$.
> >
> > This function is $h$. Moreover, this also explains why $R(X)$ is an “expectation of a Gaussian vector” as $(y, y^{\\star}) \= (a^TX, a^T X^{\\star})$ is distributed as a multivariate Gaussian with covariance $\[X|X^{\\star}\]^T K \[X|X^{\\star}\]$ and you are taking the expectation over $(y, y^{\\star}, \epsilon)$.
> >
> > **“We shall remove this condition by approximation in what follows”**
> >
> > The reviewer is absolutely right: we did not end up removing the condition, although there is a straightforward way to do so, and we will correct this.  In short, one creates an $f\_\\epsilon$ which is an approximation to $f$ formed by convolving with an isotropic Gaussian of variance $\\epsilon$.  This is $C^2$ and has bounded second derivatives (as $f$ was smooth).  We then will take limits as $\\epsilon \\to 0$.
> >
> > We note that Lemma B.1 was intended as discussion, and it is not used in the main theorem.  Theorem B.1 generalizes to the more standard L-smooth f, at the cost of additional complexities in the formulation (discussed below the Lemma), and is intended to show why we didn’t simply use the more standard L-smooth f assumptions.
> >
> > **“Mild assumptions on the learning rate” Why mild?**
> >
> > The learning rate assumptions are mild since they encompass many adaptive learning rates used in practice; they do exclude learning rates which are non-concentrating.  Among non-concentrating stepsizes, one can further consider those which bias the gradient and those which do not bias the gradient.  Those which bias the gradient require a totally different theory (e.g. scaling the gradient to be norm-1 or things like gradient clipping).  Those which do not bias the gradient would have a similar theory as those which we consider, but with an additional variance term in the ODEs (e.g. RMS-prop with a very short window of averaging, with respect to dimension).  So we chose this case to keep the theory *as simple as possible* (which we know is aligned with the reviewer’s desires\!)

---

> > > ### Comment · Reviewer_hf9X · 2024-08-14
> > >
> > > I thank the authors for their feedback.
> > >
> > > > Line search
> > >
> > > I agree the practical limitation of line search is mostly seen in stochastic optimization. Orvieto et al. [40] had a counter-example for the convergence of stochastic Polyak stepsize due to a bias, and I think the same example should reveal a bias of stochastic line search. Your result, however, is quite different, and I agree your characterization of the rate is new.
> > >
> > > > the standard analysis would say that you need to use a decreasing learning rate with SGD, but nobody does this in practice
> > >
> > > I beg to disagree. Cosine annealing is commonly used with SGD, not just Adam. I agree, however, that nobody uses the standard $1/\sqrt{t}$ and $1/t$ schedules.
> > >
> > > > The point of this paper is to provide another tool that allows for a finer analysis of these algorithms
> > >
> > > I wish there was something in the middle: more precise than the standard theory and less convoluted than the one you offer in this submission. I do agree however that the one presented here has value when we require a finer analysis.
> > >
> > > > The related work paragraph (Page 4 Lines 106-110) describes past works which use a high-dimensional framework for analyzing dynamics of stochastic algorithms.
> > >
> > > True, I did underestimate the amount of work that has been done in this setting.
> > >
> > > > Intuition and existence of $h$
> > >
> > > To be honest, I am still struggling with understanding the existence and role of this function.
> > >
> > > > Mild assumptions on the learning rate
> > >
> > > I wouldn't use the term "mild" in this setting since it's hard to quantify how mild they are. My (optional) suggestion is to explicitly state the the assumption applies to several learning rates that are of interest.
> > >
> > > I'll update my score for the paper given the points above.

---

### Official Review · Reviewer_ipsV · 2024-07-17

**Soundness:** 4
**Presentation:** 4
**Contribution:** 3
**Rating:** 7
**Confidence:** 3

**Summary:**

This paper studies Stochastic Gradient Descent (SGD) training with adaptive step sizes. The setting is a generalization of single-index models. There is a ground truth vector X^*, and the model must find a vector X such that a loss L(X) = E_{a,epsilon}[f(<X,a>, <X^*,a>, epsilon)] is minimized. Here epsilon is a random additive error, and a is distributed according to N(0,K) for a potentially anisotropic covariance K.

This paper shows that (1) in the high-dimensional regime, the training dynamics converge to a deterministic limit given by an ODE. It studies these dynamics in two significant cases of adaptive step-size algorithms: (2) exact-line-search (i.e., greedily decreasing the risk optimally at each step), and (3) AdaGrad-Norm. For exact-line search, it is shown that this can be very suboptimal if the data covariance is highly anisotropic. For AdaGrad-Norm, it is shown that if the data covariance has lower-bounded eigenvalues the optimal step size (within a constant factor) is reached. However, for harder problems with worse conditioning of the data covariance, AdaGrad-Norm can be overly pessimistic.

**Strengths:**

This paper provides a fine-grained analysis of practical optimization algorithms, and also proves a general theorem on convergence to deterministic dynamics that I believe could be used to analyze dynamics for different adaptive-learning-rate algorithms in the future. There is also a strong match between theory and experiments. I believe it is an important step in the direction of understanding these practical optimization algorithms.

**Weaknesses:**

In terms of techniques, Theorem 2.1 on convergence to an ODE is based in large part on modifying a previous analysis of [Collins-Woodfin ’23] to the case of adaptive step sizes, so the techniques are not completely novel.

The data is also assumed to be Gaussian, although anisotropy is allowed.

**Questions:**

I am satisfied with the presentation in the paper, and do not have questions.

**Limitations:**

Yes

---

> ### Author Rebuttal · Authors · 2024-08-06
>
> We thank the reviewers for their comments. We will update the paper to emphasize the difference between our analysis and that of \[14\]. We state below the main differences between this work and \[14\]:
>
> * First, the learning rate is by itself a stochastic process.  In addition, since it carries all the history of the previous gradients, the iterates of SGD are no longer a Markovian process. We extend the proof of the previous paper to handle such situations by providing an additional concentration of the learning rate to its equivalent deterministic limit.
>
> * Second, there are differences in the analysis of the Volterra equation on the least square problem. Having the learning rate as a nonlinear function of the loss makes the analysis much harder, and we had to introduce new techniques to study this new generalized form of the Convolutional Volterra equation (see Appendix D).
>
> * Third, in this paper, we study in depth the effect of different covariance matrix structures. For the line search method, we analyze in detail the example of two distinct eigenvalues.  For AdaGrad-Norm, we study different power law covariance scalings and zero eigenvalues, for which we had to utilize some nontrivial asymptotic methods.
>
> The assumption of the data being Gaussian could be relaxed. Most of the concentration estimates we use to prove the deterministic equivalent are based on concentration inequalities, which are valid beyond Gaussian data. In addition, Figure 6 in the paper shows that we can predict multipass SGD dynamics on more realistic data (CIFAR-5m) which suggests that our theory extends beyond the Gaussian data and the streaming setting.
>
> \[14\] Elizabeth Collins-Woodfin, Courtney Paquette, Elliot Paquette, and Inbar Seroussi. Hitting the high-dimensional notes: An ODE for SGD learning dynamics on GLMs and multi-index models. arXiv preprint arXiv:2308.08977, 2023\.

---

> > ### Comment · Reviewer_ipsV · 2024-08-13
> > **Response**
> >
> > Thanks, I will keep my score.

---

### Official Review · Reviewer_hpAG · 2024-07-30

**Soundness:** 3
**Presentation:** 1
**Contribution:** 2
**Rating:** 5
**Confidence:** 3

**Summary:**

The paper focuses on analytical analysis of dynamics of a class of optimization algorithms with adaptive learning rate applied to linear models with Gaussian data. The class of algorithms includes AdaGrad-Norm, RMSprop-Norm, Polyak stepsize and line search, but doesn't include, for example, Adam, classical RMSprop and AdaGrad. Also, authors consider $\frac{1}{d}$ scaling of learning rate with dimension $d$ of the problem.

First, the authors establish convergence of the optimization trajectory to a deterministic ODE in the limit $d\to\infty$. Then, they focus on analysing the resulting dynamics for quadratic loss functions and specific algorithms
- Plolyak stepsize. Obtaining learning rate.
- Exact line search. Obtaining learning rate for two cases of data covariance $K$: isotropic and with two distinct eigenvalues
- AdaGrad-Norm. $O(t^{-\frac{1}{2}})$ learning rate asymptotic for noisy observations. Also, learning rate and risk asymptotic for isotropic and power-law data covariance $K$.

The theoretical results are validated with numerical experiments on synthetic data.

**Strengths:**

- The main result (convergence to deterministic continuous-time ODE) is valid for general loss functions. The precise characterization of the optimization trajectory, as opposed to upper/lower bounds, is typically harder to obtain for non-quadratic loss functions
- For AdaGrad Norm algorithm, the authors characterized the dynamics of risk and learning rate for several distinct types of covariance matrix spectrum.

However, both these points are achieved only partially (see *weaknesses* for details). Completing them would significantly improve the paper.

**Weaknesses:**

The main drawback of the manuscript is its writing. The paper mostly focuses on the introduction and setting, leaving too little space for actual results and their discussion. Out of 9 pages:
  - Five pages are reserved for introduction and setup.
  - Two pages are occupied by Section **2**, which mainly introduces notations and demonstrates them on the quadratic problem example. Indeed, it contains Theorem **2.1**, but that one is quite short
  - Only the last two pages contain the main results of the paper. This leaves very little space for discussing them. Some of the discussions are present in the (quite lengthy) *main contributions* part, but, for readability purposes, it would be nicer to have this discussion together with or after the results.
  - Any kind of concluding section is absent

Regarding the figures, only figure **1** is referenced in the text while figures **2**,**3**, and **4** are not referenced. This adds an extra effort to interpret them and put into right context.

**New developments from previous work**. As mentioned by the authors, the current paper follows the technical framework developed in [14], extending it to adaptive algorithms. Then, it would be good if the manuscript mentioned more clearly which results/notions were already obtained in [14] and which are new additions of the current paper.
- In particular, Theorem **2.1** of the current paper seems to be very close to Theorem 1.1 of [14] which can also handle non-constant learning rates. If two theorems are indeed close, it would be interesting to discuss new aspects introduced by adaptive algorithms.
- Assumptions **1-6**, and the notations of sections **1.1**, **1.2**, **2** look very close to that of [14]. If that is the case, it would be interesting to discuss more specifically their differences. Also in that case, sections **1.1**, **1.2**, **2** could be compressed by partially moving them in appendix, leaving more space for the results and their discussion.

**Results for non-quadratic losses**. Although the framework, as introduced in section **2** is designed for general non-quadratic problems, the specific results are given only for quadratic problems in the main paper. Some results for non-quadratic problems are given in appendix sections **D.1**, **E**, **F.1**, **G.1** (mainly based on results of [14]). In such form of these results, it is hard to extract interesting conclusions, or determine the contributions compared to the previous literature.

**Questions:**

**High dimensional vs. low dimensional problems**. Some covariance matrices $K$ satisfy assumption **1** but have $\frac{\mathrm{Tr} K^2}{d}\to0$ and therefore produce trivial ODE dynamics coinciding with full-batch Gradient Flow (as can be seen from eq. 11), for example
- rank 1 with non-trivial eigenvalue $\lambda=1$ (extreme case).
- with power-law spectrum $\lambda_i=i^{-\alpha}, \alpha>1$  (e.g. classical *capacity* and *source* conditions).

Indeed, for these two examples, the dynamics converge to full-batch Gradient Flow due to $\propto \frac{1}{d}$ scaling of learning rate in eq. 3, while learning rate independent of $d$ would be more natural.
- Could the techniques developed in this paper handle such problems?
- If no, is it possible to formulate a criterion on covariance matrix $K$ to distinguish non-trivial ODE dynamics from trivial ones (i.e. coinciding with full-batch gradient flow)? For general non-quadratic problems.

**Results for power-law spectrum** The last section provides variety of results in case of power-law distribution of eigenvalues (with exponent $\beta$), and target (with exponent $\delta$). In the non-vanishing learning rate phase $\beta+\delta<1$, how the obtained rate $\mathcal{R}(t)\sim t^{\beta+\delta-2}$ compares with other results for SGD under power-law spectrum (e.g. [1](https://arxiv.org/abs/2006.08212),[2](https://arxiv.org/abs/2102.03183),[3](https://arxiv.org/abs/2206.11124))?

**Learning rate phase transition**. For power-law spectrum, learning rate asymptotic exhibits transition at $\beta+\delta=1$, as demonstrated on fig. **4**. Interestingly, on this figure we can see that in both phases learning rate reduces similarly up until it reaches very small value ($\gamma\sim 10^{-4}$) at quite late optimization times ($t\sim10^{3}$). Can this dynamic transition (w.r.t. time $t$) be explained theoretically, for example, with some quantities characterizing transition scale with $d$?

**Limitations:**

The paper surely doesn't have any negative societal impact.

---

> ### Author Rebuttal · Authors · 2024-08-06
>
> We address below the questions of the reviewer.
>
> 1. **Isotropic covariance matrix $K$.** We emphasize that our results hold for *non-isotropic data covariance matrices $K$* (under mild assumption $||K||\_{op} \< C$, and the average eigenvalue of $K$ is bounded independent of $d$). We point to Reviewer ipsV who also noted the anisotropic assumption on $K$.
>
> 2. **Writing of the manuscript.** Please see some specific changes in the “All Reviewers” section of the response. We appreciate the reviewer’s suggestions.
>
> 3. **New developments from previous work \[14\].** See discussion to all reviewers.
>
> 4. **Beyond non-quadratic losses.** In Appendix D, we show that under a restricted secant inequality, similar conclusions to the quadratic case can be derived for AdaGrad-Norm.   Now the results we get are essentially the same as one gets for quadratics.  We weren’t so keen to generalize the whole paper to something like losses satisfying an RSI if, at the end of the day, we just get the same answers as for quadratics.  Identifying interesting and important non-quadratics with different phenomenology for adaptive algorithms is an interesting direction of future research (e.g. nonconvex problems like retrieval or problems exhibiting implicit bias). Using Theorem 2.1, the ODEs are already in place, and so there is a framework for finer analysis of adaptive algorithms which was missing from the literature. We believe these methods can eventually be used to provide better or more informed adaptive stepsize methods.
>
> 5. **High-dimensional vs low-dimensional problem extensions.** Thanks for these points – they are very good.  Please see the discussion in the rebuttal to all reviewers.  We’ll remark that the $\\text{Tr}(K^2) \= o(d)$ condition you observed is implied by $\\text{Tr}(K) \= o(d)$, so long as we keep bounded operator norm of $K$, and so it falls into the regime where SGD and the ODEs both need to be scaled differently.  We can add a discussion of this to the paper.
>
>    Part of the reason we wrote *this* paper is that we needed to demonstrate that it was actually possible to analyze these ODEs and derive meaningful conclusions about optimization algorithms.  Especially, we needed to analyze these ODES in the presence of anisotropy, which can heavily degrade the value of an adaptive stepsize algorithm (e.g. line searches work great in the isotropic case – and are essentially optimal – but they are bad in the presence of anisotropy.  In contrast, adagrad norm does much better, although still fails in some power law-type setups).  So half of this paper is analyzing various algorithms and half is formulating a precise theorem which says where the ODE comes from.
>
>    We hope this paper could be a starting point for us (or, gladly, other researchers\!) to refine the method further, for example:
> * Finding the optimal covariance conditions under which these ODEs hold and making a fully non-asymptotic comparison.
> * Understanding the extra correction terms to the ODEs in non-high-dimensional setups, such as the classical source/capacity conditions.  (The rank-1 problem you mentioned, in contrast, is not well-aligned with the goals of this type of analysis, which we think could be summarized as: how to give meaningful quantification of adaptive algorithms on anisotropic problems).
> * Digging deep into other adaptive stepsize strategies to see if they perform better in anisotropic setups.
> * Showing a version of Theorem 2.1 for non-Gaussian data (high-dimensionality also gives a path to showing universality of the ODEs).
>
>
>   It could be, for example, that Adagrad norm on classical capacity source setups could be analyzed in the way we did and have interesting conclusions.
>
>
> 6. **SGD power-law spectrum rate.** We thank the reviewer for the references. Under our set-up, constant stepsize SGD would be $\\mathcal{R}(t) \\asymp t^{\\beta \+\\delta \-2}$, and there is no transition at $\\beta+\\delta \=1$. We will add this to our paper. The references \[1\],\[2\],\[3\] suggested by the reviewer do consider similar assumptions on the spectrum and eigenvalues of the covariance matrix, but the papers referenced are not in the high-dimensional regime (diverging $\\text{Tr}(K)$). All \[1\],\[2\],\[3\] contain results which formally correspond to $\\beta \> 1$, so a quantitative comparison is not really possible (\[3\] discusses our regime here, but does not seem to have rates in this regime).  Qualitatively, they all demonstrate similar results as seen here (formally similar to how the loss undergoes a phase transition as the ‘source’ parameter varies in source/capacity setups). In the next version of the paper, we will add clarification regarding the scaling with constant stepsize together with comparison and citations of the relevant works.
>
> 7. **Learning rate phase transition.** This is a great question and worthy of future research. While we did not keep track of the exact finite dimensional $d$ effects, one could in principle use our analysis to theoretically quantify when the asymptotics for the learning rate and the risk take over.

---

> > ### Comment · Reviewer_hpAG · 2024-08-13
> > **Rebuttal update**
> >
> > Dear authors, thank you for your response, including the general rebuttal. Based on the rebuttal, I am willing to increase the score to 5. Let me clarify the reasoning behind it, so that the authors and AC have a possibility to calibrate the score. The positive things are
> >   - proposed changes in writing are good
> >   - the discussion of relation with [14] adds clarity and transparency
> >   - the comments regarding different scaling of learning rate with dimension $d$, as well as additional discussion of power-law rates, help to better identify the positioning and relation of the current paper with other approaches.
> >
> > What prevents me from raising the score further (e.g. 7) is that the paper still explores just a little bit of the behaviors of adaptive algorithms. The theorem 2.1 is a solid foundation for studying adaptive algorithms. However, it really pays off with its subsequent application to different adaptive optimization scenarios. The applications in the current manuscript (sections 3 and 4) still feel as kind of short and not properly explored. The same goes to the application to non-quadratic problems - authors comment that they are roughly the same as quadratic problems, which does not sound satisfying.

---

### Author Rebuttal · Authors · 2024-08-06

We thank the referees for their time and constructive comments that significantly helped us improve our paper. The reviewers questions were deep and so *we will add additional information as an "Official Comment."*

**Paper structural changes.**

In our next version of the paper, we will implement the following reorganization changes to improve the clarity and flow:

* Condense the assumptions, as much as possible, in the introduction. However, we do value stating upfront the formal assumptions in the main paper, as we do not want to hide anything from the reader. This will include, for instance, moving the discussion of the Assumptions, to the Appendix. We will also move the ODEs for the least squares problem earlier in the paper as recommended by Reviewer A5Ho.
* Add Conclusion paragraph *(see below in "Official Comment").*
* Add Limitation paragraph *(see below in "Official Comment")*.
* Increase the font size in the figures *(see attached PDF)* and added references to the figures in the inline text.

**New developments from previous work \[14\].**

This work’s Theorem 2.1 is inspired by Theorem 1.1 from \[14\]. We note that Theorem 1.1 in \[14\] does not apply to adaptive (stochastic) learning rates. Theorem 1.1 only holds for learning rates schedules which are *deterministic* such as decaying learning rates $1/t$. The new results (Thm. 2.1) allow for *stochastic adaptive learning rates.* This is important since many of the widely used algorithms in practice would not apply to results in \[14\], but do apply to our new result Theorem 2.1. This allows us to understand why these algorithms are used more in practice over say deterministic decaying step sizes.

Concretely, Theorem 2.1 (with stochastic adaptive learning rates) requires substantial work and a new proof that shows the stochastic learning rates and the loss are simultaneously concentrating around a deterministic function.

We will add a discussion about the differences between Theorem 1.1 in \[14\] and the new result Theorem 2.1 to the paper.

\[14\] Elizabeth Collins-Woodfin, Courtney Paquette, Elliot Paquette, and Inbar Seroussi. Hitting the high-dimensional notes: An ODE for SGD learning dynamics on GLMs and multi-index models. arXiv preprint arXiv:2308.08977, 2023

**Other scalings and lower rank K.**

We have adopted a scaling convention where stepsize scales like $1/d$.  As multiple reviewers noted, this may not be the only scaling in which one sees a nontrivial limit.  An important non-asymptotic measurement of $K$ is  $\\text{Tr}(K) / ||K||\_{op}$, sometimes called intrinsic dimension $\\text{Dim}(K)$.  We believe that a version of our main Theorem 2.1 generalizes to a non-asymptotic setting where the intrinsic dimension (as opposed to the ambient dimension $d$) is large, and this affects the step-size scalings that should be used in SGD.

1. Our theorem is really about the case that $\\text{Dim}(K)=\\Theta(d)$.  This is the case, e.g. when one has a dimension-independent condition number (a big part of our paper).  Here the $(1/d)$ scaling is where one sees non-trivial contribution to the dynamics from both the gradient flow term and the stochastic term (SGD noise).  If one were to choose a smaller scaling (e.g. $1/d^2$), the dynamics would devolve to gradient flow, whereas if one were to choose a more aggressive learning rate, the dynamics would devolve to pure stochastic noise (and the risk would generally tend to infinity).  With the $1/d$ scaling one sees the effect of both gradient flow and stochastic noise in the dynamics.  Put another way, we have chosen the largest learning rate scaling for which one has a nontrivial limit.
2. When the $\\text{Dim}(K) \\to \\infty$ but $\\text{Dim}(K) \= o(d)$, the scaling of $\\gamma$ should not be $1/d$.  It should rather be that the scaling is $1/\\text{Dim}(K)$ – for quadratic losses, one can check the same story as above holds here: larger learning rates lead to pure-noise and smaller learning rates lead to gradient flow approximations.  On the other hand, the ODEs that we derived *are still the same ODEs*; one just needs to rescale time and learning rate in the ODEs to produce the correct equations.  The method in our paper would definitely extend to *some* intermediate growth rates of $\\text{Dim}(K)$, but in our mind, the better theorem would be a fully non-asymptotic theorem – one which shows that when $\\text{Dim}(K)$ is large, the ODEs and the SGD curves are close as a function of $\\text{Dim}(K)$.  That’s a much bigger task, and it could clearly be a standalone future project.
3. When $\\text{Dim}(K)$ is bounded above (say in the case of classical source/capacity restrictions), the ODE approximation has to change substantially.  The losses don’t concentrate, and the mean behavior of the ODEs follow a discrete difference equation.  Now one can use tricks, like embedding the markov chain in continuous time so the mean risk follows an ODE.  But even doing this, the ODEs have additional terms (which simplify in high dimensions\!), and so they are strictly more complicated to analyze.  We expect that in some cases this is not hopeless (such as source/capacity type conditions), and the extra terms will not play a substantial role.

Both regimes 2 and 3 are interesting, and there is plenty of room for future work in both those directions (both with adaptive and non-adaptive stepsizes).  We did not directly pursue them in this paper, because we needed to be able to show that it was possible to derive meaningful conclusions about the limit system of ODEs in the high-dimensional case.  So we pursued the more modest goal of proving the ODE comparison in the maximal-dimension case.

---

> ### Author Response · Authors · 2024-08-07
> **Proposed conclusion/limitations paragraphs**
>
> **Conclusion paragraph (to be added):**  *This work studies stochastic adaptive optimization algorithms when data size and parameter size are large, allowing for nonconvex and nonlinear risk functions, as well as data with general covariance structure. The theory shows a concentration of the risk, the learning rate and other key functions to a deterministic limit, which is described by a set of ODEs. The theory is then used to derive the asymptotic behavior of the AdaGrad-Norm and idealized exact line search on strongly convex and least square problems, revealing the influence of the covariance matrix structure on the optimization.  A potential extension of this work would be to study other adaptive algorithms such as D-adaptation, DOG, and RMSprop which are covered by the theory. Studying the asymptotic behavior of the risk and the learning rate may improve our understanding of the performance and scalability of these algorithms on more realistic data. Another important application of the theory would be to analyze the ODEs presented here on nonconvex problems.*
>
> **Limitations (to be added):** *The current form of the theory is limited to Gaussian data, though many parts of the proof can be extended easily beyond Gaussian data. The main ODE comparison theorem is also only tuned for analyzing problem setups where the trace of the covariance is on the order of the ambient dimension; when the trace of the covariance is much smaller than ambient dimension, other stepsize scalings of SGD are needed. In addition, the analysis is limited to the streaming stochastic adaptive methods. We conjecture that a similar deterministic equivalent holds also for multi-pass algorithms at least for convex problems. This has already been shown in the least square problem for SGD with a fixed deterministic learning rate \[42,44\]. Lastly, numerical simulations on real datasets (e.g., CIFAR-5m) suggests that the predicted risk derived by our theory matches the empirical risk of multipass SGD beyond Gaussian data (see for example Figure 6).*

---

### Decision · Program_Chairs · 2024-09-25

**Decision:**

Accept (poster)

**Comment:**

This is an excellent paper that makes a number of strong theoretical and conceptual contributions to our understanding of the dynamics of adaptive stochastic optimization algorithms in high dimensions. Moreover, the mathematical techniques developed are likely to be of use to others. The authors have addressed the major concerns of the reviewers.

Given the depth of the paper and the additional areas for further exposition and exploration, I encourage the authors to consider the suggestion of one reviewer to write an extended, journal-length version of the paper that could do the material more justice.